# A Bias–Variance Tradeoff Perspective for Improving Test-Time Scaling

## Abstract

Parallel test-time scaling (PTTS) improves the reasoning performance of large language models (LLMs) by aggregating multiple candidate solutions at inference time. However, existing methods remains largely heuristic, lacking a principled framework that explains their behavior, clarifies their limitations, and guides systematic improvement. To bridge this notable gap, we introduce the first general framework for PTTS through a unifying probabilistic inference formulation, seamlessly encompassing prior disparate methods as special cases. This framework enables a novel bias-variance tradeoff perspective to reveal the intrinsic limitations of existing methods and serves as a principled foundation for developing new ones. Specifically, our framework reveals that existing verifier-based methods act as *high-variance* importance sampling (IS) estimators, yielding marginal gains under small scaling budgets, whereas generator-based methods act as *biased* variational inference (VI) estimators, yielding suppressed scalability even under large budgets. To formally characterize the tradeoff, we derive the first theoretical bias–variance formulation for PTTS, revealing that the relative variance upper bound is jointly governed by the generator and the verifier via their respective optimality gaps. To mitigate this tradeoff, we build upon our general framework and derive a theory-driven PTTS method named TSMC-TTS. Specifically, it instantiates our framework with twisted sequential Monte Carlo (TSMC) and performs EM-like optimization of the generator and verifier guided by the theoretical variance bound, thus achieving monotonic variance reduction without compromising unbiasedness. Furthermore, we introduce a new self-evolving TSMC mechanism, effectively alleviating both the reward sparsity and the computational overhead issues inherent in vanilla TSMC. Rigorous theoretical analysis and comprehensive empirical results demonstrate the efficacy of our proposed method.

## 1 Introduction

Recent progress in parallel test-time scaling (PTTS) (Brown et al., 2024) has demonstrated that aggregating multiple candidate solutions at inference time can significantly improve mathematical reasoning of large language models (LLMs). The most widely adopted aggregation strategy, majority voting (MV) (Wang et al., 2022), selects the answer with the highest frequency as the final answer. While simple and effective in some cases, vanilla MV suffers from a fundamental limitation we call *frequency–correctness mismatch*. The root cause is that the candidate distribution $p(\mathbf{x}_{1:T}|\mathbf{x}_0)$ reflects the model's generation preference rather than the correctness preference. Consequently, an answer that is frequently generated (high frequency) may not necessarily be correct. In contrast, the ideal candidate distribution should be reward-aligned , i.e., allocating higher probability mass to correct solutions, thereby ensuring frequency and correctness are well aligned.

Existing efforts to achieve better frequency-correctness alignment can be broadly categorized into two directions: (i) *Verifier-based methods* (Cobbe et al., 2021; Uesato et al., 2022; Lightman et al., 2024), which train an external verifier to score candidates, so as to adjust each candidate's voting weight when determining the final answer. (ii) *Generator-based methods* (Schulman et al., 2017; Shao et al., 2024b; Song et al., 2025), which refine the generator to produce better-quality candidates before applying voting. However, the field remains largely driven by heuristics, lacking a principled framework that explains their behavior, clarifies their limitations, and guides the systematic improvement of PTTS.

To bridge this notable gap, we propose a general framework that recasts PTTS as a probabilistic inference problem: estimating expected statistics (e.g., answer frequency) over an intractable reward-aligned distribution. Within this framework, existing methods can be unified as special cases, i.e., answer-frequency estimators derived from different approximate inference algorithms. This general framework offers several key insights: **(i) Mechanism clarification.** While existing methods are ostensibly distinct, they fundamentally seek to estimate expected answer frequency over a reward-aligned distribution, thus achieving better frequency-correctness alignment. **(ii) Limitation revelation via bias-variance tradeoff.** The framework reveals previously overlooked limitations tied to their underlying inference algorithms, notably the bias-variance tradeoff. Specifically, we reveal that verifier-based methods correspond to IS estimators, suffering from marginal gains under limited scaling budgets due to their inherent high-variance nature, whereas generator-based methods correspond to VI estimators, suffering from suppressed scalability even under larger budgets caused by their inherent biased nature. **(iii) Foundation for new methods.** Our general framework serves as a principled foundation that can be instantiated into new PTTS methods by leveraging more advanced approximate inference algorithms, thereby opening avenues for designing novel PTTS methods with well-grounded principles.

To formally characterize the tradeoff, we derive the first theoretical bias–variance formulation for PTTS, revealing that the relative variance upper bound is jointly governed by the generator and the verifier via their respective optimality gaps (see Theorem 5.2). To mitigate the bias–variance tradeoff, we build upon our general framework and derive a theory-driven PTTS method named TSMC-TTS. Specifically, TSMC-TTS is a principled instantiation of our general PTTS framework by adopting TSMC as the inference algorithm. The choice of TSMC (Zhao et al., 2024a) is motivated by its potential to yield an unbiased yet low-variance estimator. This instantiation with TSMC induces a step-wise LLM inference process where the generator incrementally extends partial solutions and the verifier allocates scaling budgets to the most promising ones. Guided by our theoretical insights (Theorem 5.2), we propose a EM-like generator-verifier optimization algorithm ensuring monotonic variance reduction without compromising unbiasedness, thus enabling significant gains under both low and high scaling budgets. This approach unifies the improvement of the generator and verifier in a principled manner, and does not require any step-wise human annotations as training data. Furthermore, we introduce a new self-evolving TSMC mechanism, effectively alleviating the reward sparsity and the computational overhead issues inherent in vanilla TSMC.

Overall, our contributions are summarized as follows:

- We introduce the first general framework for PTTS through a unifying probabilistic inference formulation, seamlessly encompassing previously disparate methods as special cases. This framework offers a new bias–variance tradeoff perspective to reveal the intrinsic limitations of existing methods and provides a principled foundation for developing new methods (see Sec. 4).
- We rethink existing methods through a fundamental bias–variance tradeoff lens: verifier-based methods act as high-variance IS estimators, yielding marginal gains under small scaling budgets, whereas generator-based methods act as biased VI estimators, yielding suppressed scalability even under large budgets (see Sec. 4.2).
- We present the first theoretical bias–variance formulation for PTTS, revealing that the relative variance upper bound is jointly governed by the generator and the verifier via their respective optimality gaps (see Theorem 5.2).
- Building upon our proposed framework, we propose a theory-driven PTTS method named TSMC-TTS, effectively mitigating the bias-variance tradeoff. Furthermore, we introduce a new self-evolving TSMC mechanism, effectively alleviating the reward sparsity and computational overhead issues inherent in vanilla TSMC (see Sec. 5).
- We provide in-depth theoretical analysis to derive the variance bound and prove the monotonic convergence property of our proposed method. Extensive experiments on several representative datasets further verify the effectiveness of our theory-driven method.

## 2 RELATED WORK

PTTS (Muennighoff et al., 2025; Snell et al., 2025) refers to generating and aggregating multiple candidate solutions at inference time. It typically involves two components, the verifier and generator, and existing work can be accordingly categorized into verifier-based and generator-based methods.

Verifier-based methods introduce an external verifier to score each candidate for guiding aggregation. Such verifiers are divided into Outcome Reward Models (ORMs) (Cobbe et al., 2021) and Process Reward Models (PRMs) (Uesato et al., 2022). ORMs directly provide sequence-wise scores, whereas PRMs compute such scores by combining step-wise scores. The final answer is selected via candidate aggregation: either the answer from the highest-score solution (best-of-N (Stiennon et al., 2020)) or the answer with the highest total score (weighted majority voting, WMV (Littlestone & Warmuth, 1994; Li et al., 2023)). Recently, Feng et al. (2025) and Puri et al. (2025) attempt to incorporate probabilistic inference techniques into PTTS, but they fail to address the intrinsic bias–variance tradeoff inherent in such techniques, leaving their verifier and generator design fundamentally ill-suited to handle this core limitation. The key distinctions from these works are summarized in Tab. 5 of App. B.2. In contrast to the above, generator-based methods directly optimize the generator to produce more likely correct candidates, typically by aligning it with a reward signal through reinforcement fine-tuning (RFT) (Schulman et al., 2017). Our work proposes a general probabilistic inference framework to unify various PTTS methods and offers a new bias-variance tradeoff perspective for improving PTTS.

## 3    PROBLEM SETUP

**PTTS setup.** PTTS involves generating multiple candidates during inference and then aggregating them into a final answer. Formally, let $q(\mathbf{x}_{1:T} \mid \mathbf{x}_0) = \prod_{t=1}^{T} q(\mathbf{x}_t \mid \mathbf{x}_{1:t-1}, \mathbf{x}_0)$ denote a generator that autoregressively produces candidates $\mathbf{x}_{1:T}$ of maximum step $T$ in response to a prompt $\mathbf{x}_0$. Typically, the generator $q(\mathbf{x}_{1:T}|\mathbf{x}_0)$ is initialized by a pre-trained language model $p_0(\mathbf{x}_{1:T}|\mathbf{x}_0)$. Let $\mathcal{R}(\mathbf{x}_{1:T}|\mathbf{x}_0)$ be a reward signal defined over complete sequences, indicating the desired property (e.g., $\mathcal{R}(\mathbf{x}_{1:T}|\mathbf{x}_0) \in \{0, 1\}$ for correctness in reasoning tasks). The goal of PTTS is to aggregate multiple candidates into a final answer that achieves a higher reward.

## 4    A GENERAL PROBABILISTIC FRAMEWORK FOR PTTS

**Target distribution.** We propose a general framework that recasts PTTS as a probabilistic inference problem: estimating expected statistics (e.g., answer frequency) over an intractable reward-aligned target distribution. Specifically, we consider a general formulation of the target distribution $\sigma(\mathbf{x}_{1:T}|\mathbf{x}_0)$, which is defined by modulating a base distribution $p$ with a potential function $\phi$:

$$\sigma\left(\mathbf{x}_{1:T} \mid \mathbf{x}_0\right) := \frac{1}{\mathcal{Z}_\sigma\left(\mathbf{x}_0\right)} p\left(\mathbf{x}_{1:T} \mid \mathbf{x}_0\right) \phi(\mathbf{x}_{1:T}|\mathbf{x}_0),$$

$$\text{where } \mathcal{Z}_\sigma\left(\mathbf{x}_0\right) = \sum_{\mathbf{x}_{1:T}} \tilde{\sigma}\left(\mathbf{x}_{1:T} \mid \mathbf{x}_0\right) = \sum_{\mathbf{x}_{1:T}} p\left(\mathbf{x}_{1:T} \mid \mathbf{x}_0\right) \phi(\mathbf{x}_{1:T}|\mathbf{x}_0). \quad (1)$$

Here, $\tilde{\sigma} = p \cdot \phi$ denotes the unnormalized density and $\mathcal{Z}_\sigma$ is the normalizing constant. The base distribution $p$ and potential function $\phi$ serve as flexible placeholders that can be instantiated by different TTS methods summarized in Tab. 1. This general form captures a broad family of $\phi$-aligned distributions by varying the choice of $\phi$. For notational simplicity, we omit the conditioning on $\mathbf{x}_0$ throughout the remainder of this paper.

**Expected answer frequency under $\sigma(\mathbf{x_{1:T}})$.** Given the target distribution $\sigma(\mathbf{x}_{1:T})$, the expected answer frequency $f_a$ is defined as the proportion of answer $a$ over all possible answers extracted from $\mathbf{x}_{1:T} \sim \sigma(\mathbf{x}_{1:T})$:

$$f_a = \mathbb{E}_{\sigma(\mathbf{x}_{1:T})}\left[\mathbb{I}\big(\text{Ans}(\mathbf{x}_{1:T}) = a\big)\right], \quad (2)$$

where $\text{Ans}(\mathbf{x}_{1:T})$ is the extracted answer and $\mathbb{I}(\cdot)$ is the indicator function. Since direct sampling from $\sigma(\mathbf{x}_{1:T})$ is intractable, approximate inference techniques such as IS and VI can be employed to estimate $f_a$.

### 4.1    UNIFY EXISTING PTTS METHODS AS SPECIAL CASES

Within our framework, existing methods can be unified as special cases, i.e., answer-frequency estimators under $\sigma(\mathbf{x}_{1:T})$ derived from different approximate inference algorithms. For ease of understanding, we refer readers to App. F for preliminaries on IS and VI estimators. Then, we formally establish this connection.

**Proposition 4.1** (**Verifier-based methods as IS estimators (Simplified; full version in App. C.1)**). *Let $q(\mathbf{x}_{1:T})$ be the generator and suppose the verifier assigns each candidate $\mathbf{x}_{1:T} \sim q(\mathbf{x}_{1:T})$ a sequence-wise score $W(\mathbf{x}_{1:T}) = \mathrm{Aggr}[w(\mathbf{x}_{1:t})]_{t=1}^{T}$ (e.g., min or $\prod$ over step-wise scores). Such verifiers are typically categorized into ORMs and PRMs. Verifier-based methods then select the final answer from $N$ candidates via score-weighted majority voting (Wu et al., 2025):*

$$a^* = \arg\max_a \sum_{i=1}^{N} W(\mathbf{x}_{1:T}^i)\mathbb{I}\big(\mathrm{Ans}(\mathbf{x}_{1:T}^i) = a\big), \quad \mathbf{x}_{1:T}^i \sim q(\mathbf{x}_{1:T}). \tag{3}$$

*Consider the reward-aligned distribution defined as $\sigma(\mathbf{x}_{1:T}) \propto q(\mathbf{x}_{1:T})\mathcal{R}(\mathbf{x}_{1:T})$ (cf. Eq. (1)). Then the expected answer frequency $f_a$ under $\sigma(\mathbf{x}_{1:T})$ can be estimated by IS as*

$$\hat{f}_a^{\mathrm{IS}} = \sum_{i=1}^{N} \frac{W(\mathbf{x}_{1:T}^i)}{\sum_{j=1}^{N} W(\mathbf{x}_{1:T}^j)} \mathbb{I}\big(\mathrm{Ans}(\mathbf{x}_{1:T}^i) = a\big) \approx f_a, \quad \mathbf{x}_{1:T}^i \sim q(\mathbf{x}_{1:T}). \tag{4}$$

*Derivation is provided in App. C.1. The approximation is justified by the Monte Carlo estimation of expectations and its asymptotically vanishing error (Owen, 2013). Here $q(\mathbf{x}_{1:T})$ acts as the proposal distribution and $W(\mathbf{x}_{1:T})$ as the unnormalized importance weight in IS. At inference time, we assume that $W(\mathbf{x}_{1:T})$ closely approximates the reward signal $\mathcal{R}(\mathbf{x}_{\infty:\mathcal{T}})$. Hence, verifier-based method in Eq. (3) is equivalent to selecting the answer with the highest estimated frequency $\hat{f}_a^{\mathrm{IS}}$ (the denominator $\sum_{j=1}^{N} W(\mathbf{x}_{1:T}^j)$ in Eq. (4) is constant and thus irrelevant to maximization):*

$$Eq.\,(3) \quad \Longleftrightarrow \quad \arg\max_a \hat{f}_a^{\mathrm{IS}}. \tag{5}$$

**Proposition 4.2** (**Generator-based methods as VI estimators (Simplified; full version in App. C.2)**). *Let $q_{\mathrm{RFT}}(\mathbf{x}_{1:T})$ denote the generator trained via reinforcement fine-tuning (RFT) to produce high-reward candidates by minimizing the policy-gradient loss $\mathcal{L}_{\mathrm{RFT}} = -\mathbb{E}_{\mathbf{x}_{1:T} \sim q}\big[\mathcal{R}(\mathbf{x}_{1:T})\big] + \beta\,\mathcal{D}_{\mathrm{KL}}\big(q(\mathbf{x}_{1:T}) \,\|\, p_0(\mathbf{x}_{1:T})\big)$ following (Jaques et al., 2020; Schulman et al., 2017; Zhang et al., 2025) (with $\beta > 0$ the KL-regularization coefficient). Generator-based methods then select the final answer from $N$ candidates via majority voting (Wang et al., 2022):*

$$a^* = \arg\max_a \sum_{i=1}^{N} \mathbb{I}\big(\mathrm{Ans}(\mathbf{x}_{1:T}^i) = a\big), \quad \mathbf{x}_{1:T}^i \sim q_{\mathrm{RFT}}(\mathbf{x}_{1:T}). \tag{6}$$

*Consider a reward-aligned distribution defined as $\sigma(\mathbf{x}_{1:T}) \propto p_0(\mathbf{x}_{1:T})\exp\big(\mathcal{R}(\mathbf{x}_{1:T})/\beta\big)$ (cf. Eq. (1)). Then the expected answer frequency $f_a$ under $\sigma(\mathbf{x}_{1:T})$ can be estimated by VI as*

$$\hat{f}_a^{\mathrm{VI}} = \frac{1}{N} \sum_{i=1}^{N} \mathbb{I}\big(\mathrm{Ans}(\mathbf{x}_{1:T}^i) = a\big) \approx f_a, \quad \mathbf{x}_{1:T}^i \sim q_{\mathrm{VI}}(\mathbf{x}_{1:T}). \tag{7}$$

*Derivation is provided in App. C.2. Here $q_{\mathrm{VI}}(\mathbf{x}_{1:T})$ acts as a tractable surrogate for $\sigma(\mathbf{x}_{1:T})$ by minimizing the VI loss $\mathcal{L}_{\mathrm{VI}} = \mathcal{D}_{\mathrm{KL}}\big(q(\mathbf{x}_{1:T}) \,\|\, \sigma(\mathbf{x}_{1:T})\big)$, where $q_{\mathrm{VI}}$ is constrained within the parametric family induced by the generator network. Based on the equivalence between RFT and VI objective, i.e., $q_{\mathrm{RFT}} \Longleftrightarrow q_{\mathrm{VI}}$ (see App. D.6 for proof), generator-based method in Eq. (6) is equivalent to selecting the answer the highest estimated frequency $\hat{f}_a^{\mathrm{VI}}$:*

$$Eq.\,(6) \quad \Longleftrightarrow \quad \arg\max_a \hat{f}_a^{\mathrm{VI}}. \tag{8}$$

*For clarity and ease of comparison, we summarize all these connections in Tab. 1.*

### 4.2 LIMITATIONS OF EXISTING PTTS METHODS

**Intrinsic bias–variance tradeoff.** Recasting existing methods into special cases of probabilistic inference exposes a fundamental limitation rooted in their underlying inference algorithms. It is well established that IS (corresponding to verifier-based method) yields an unbiased yet high-variance estimator (Robert et al., 1999), whereas VI (corresponding to generator-based method) yields a biased

Table 1: A general probabilistic inference framework for PTTS. Existing methods are unified as answer-frequency estimators under a reward-aligned target distribution $\sigma(\mathbf{x}_{1:T})$ via different inference algorithms. Descriptions of verifier-based ( ), generator-based ( ), and our method ( ) are provided in App. G.4.

| Method | Generation (generator) | Verification (score $w(\mathbf{x}_{1:t})$) | | Aggregation | Mapping to Probabilistic Inference: estimator under $\sigma(\mathbf{x}_{1:T})$ | | | |
| --- | --- | --- | --- | --- | --- | --- | --- | --- |
| | | $t < T$ | $t = T$ | | Target distribution | Inference Algorithm | Bias | Variance |
| MV | $p_0(\mathbf{x}_t\|\mathbf{x}_{1:t-1})$ | 1 | 1 | MV | $p_0(\mathbf{x}_{1:T})$ | – | – | – |
| ORM | $p_0(\mathbf{x}_t\|\mathbf{x}_{1:t-1})$ | 1 | $\mathrm{ORM}(\mathbf{x}_{1:T})$ | WMV | $p_0(\mathbf{x}_{1:T})\mathcal{R}(\mathbf{x}_{1:T})$ | IS | unbiased | high |
| PRM | $p_0(\mathbf{x}_t\|\mathbf{x}_{1:t-1})$ | $\mathrm{PRM}(\mathbf{x}_{1:t})$ | $\mathrm{PRM}(\mathbf{x}_{1:T})$ | WMV | $p_0(\mathbf{x}_{1:T})\mathcal{R}(\mathbf{x}_{1:T})$ | IS | unbiased | high |
| RFT | $\arg\min_q \mathcal{L}_{\mathrm{RFT}}$ | 1 | 1 | MV | $p_0(\mathbf{x}_{1:T})\exp\left(\frac{\mathcal{R}(\mathbf{x}_{1:T})}{\beta}\right)$ | VI | biased | low |
| TSMC-TTS | $q^*(\mathbf{x}_t\|\mathbf{x}_{1:t-1})$ | $\frac{\psi^*(\mathbf{x}_{1:t})}{\psi^*(\mathbf{x}_{1:t-1})}$ | $\frac{\psi^*(\mathbf{x}_{1:T})}{\psi^*(\mathbf{x}_{1:T-1})}$ | WMV | $q^*(\mathbf{x}_{1:T})\mathcal{R}(\mathbf{x}_{1:T})$ | TSMC | unbiased | low |

yet low-variance estimator (Yao et al., 2018). However, how this intrinsic bias–variance tradeoff manifests in practice in the context of PTTS remains unclear; we next explain its impact through the resulting failure modes.

**Verifier-based methods: marginal improvements under low solution budgets due to high variance.** Verifier-based methods inherit from IS the desirable property of unbiasedness: the estimated answer frequency recovers the true answer frequency under $\sigma(\mathbf{x}_{1:T})$ in expectation. However, when the number of candidates is limited, the variance of the estimator increases significantly (Chatterjee & Diaconis, 2018). Consequently, the frequency estimation exhibits low precision across individual runs, especially under low solution budgets. Empirical evidence in Fig. 1 demonstrates that, when only a small number of solutions (e.g., 40) are available, the verifier-based method (dark blue curve) yields only marginal gains over the baseline (light blue curve), thus severely constraining its practical effectiveness in scenarios with restricted solution budgets.

**Generator-based methods: suppressed scalability even with increasing solution budgets due to bias.** Generator-based methods inherit from VI, which yields lower-variance estimators of answer frequency but at the cost of introducing systematic bias. This property yields more stable estimations but risks anchoring to suboptimal or even incorrect estimations, even as the number of candidate solutions increases. This bias stems from the mode-seeking behavior of the reverse KL divergence adopted in RFT (Wang et al., 2024), which confines the candidate distribution to a limited region, potentially restricting exploration of correct answers (Li et al., 2025; Song et al., 2025). As shown in Fig. 1, as the number of candidate solutions increases, the performance of generator-based methods (pink curve) even drops below that of the baseline (light blue curve).

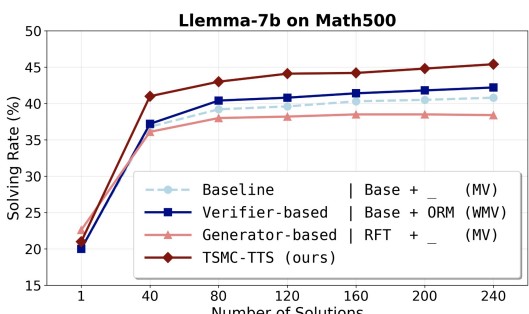

Figure 1: Parallel scaling performance in solving rate (%). "A + B (C)" indicates generator A + verifier B with aggregation strategy C.

## 5 METHODOLOGY

To address the bias–variance tradeoff, we propose TSMC-TTS, a principled instantiation of our framework that uses TSMC as the approximate inference algorithm, yielding an unbiased and low-variance estimator under $\sigma(\mathbf{x}_{1:T})$. We define $\sigma(\mathbf{x}_{1:T})$ as $\sigma(\mathbf{x}_{1:T}) \propto q(\mathbf{x}_{1:T})\mathcal{R}(\mathbf{x}_{1:T})$ (cf. Eq. (1)). We first outline the inference procedure for frequency estimation (Sec. 5.1), then derive a relative variance bound (Sec. 5.2), which in turn guides the optimization for variance reduction (Sec. 5.3).

## 5.1 INFERENCE PROCEDURE OF TSMC-TTS (ALG. 2 IN APP. A)

To estimate answer frequency under $\sigma(\mathbf{x}_{1:T})$, TSMC-TTS performs two inference-time stages: candidate collection and aggregation. Collection proceeds step by step, with each step $t$ involving three operations.

**1. Generation.** TSMC-TTS maintains a set of $N$ partial solutions $\{\mathbf{x}_{1:t-1}^i\}_{i=1}^N$. Each next-step extension $\mathbf{x}_t^i$ is sampled from the generator $q(\mathbf{x}_t^i \mid \mathbf{x}_{1:t-1}^i)$ at index $i$. For reasoning tasks, a step is defined as generating a variable-length string $\mathbf{x}_t$ representing one reasoning segment (e.g., text between two double-newline tokens).

**2. Verification.** To provide intermediate guidance, TSMC-TTS introduces a sequence of *learnable target distributions* $\{\pi_t(\mathbf{x}_{1:t})\}_{t=1}^T$. Each $\pi_t(\mathbf{x}_{1:t})$ is defined as the product of the generator distribution and a *twist function* $\psi(\mathbf{x}_{1:t})$:

$$\pi_t(\mathbf{x}_{1:t}) \propto q(\mathbf{x}_{1:t})\,\psi(\mathbf{x}_{1:t}). \tag{9}$$

We refer to $\{\pi_t(\mathbf{x}_{1:t})\}_{t=1}^T$ collectively as the *verifier*, emphasizing its role in assigning step-wise scores:

$$w(\mathbf{x}_{1:t}) = \frac{\tilde{\pi}_t(\mathbf{x}_{1:t})}{q(\mathbf{x}_t \mid \mathbf{x}_{1:t-1}) \cdot \tilde{\pi}_t(\mathbf{x}_{1:t-1})} = \frac{\psi(\mathbf{x}_{1:t})}{\psi(\mathbf{x}_{1:t-1})}, \tag{10}$$

where $\tilde{\pi}_t(\mathbf{x}_{1:t})$ denotes the unnormalized density corresponding to the normalized distribution $\pi_t(\mathbf{x}_{1:t})$. At the final step $T$, the sequence-level score is obtained by multiplying all step-wise scores:

$$W(\mathbf{x}_{1:T}) = \prod_{t=1}^T w(\mathbf{x}_{1:t}) = \psi(\mathbf{x}_{1:T}). \tag{11}$$

A key property of this construction is that if $\psi(\mathbf{x}_{1:T}) = \mathcal{R}(\mathbf{x}_{1:T})$, the resulting estimator over $\sigma(\mathbf{x}_{1:T})$ is unbiased regardless of the generator $q$ and intermediate verifier $\{\pi_t\}_{t=1}^{T-1}$ (see Theorem 5.1).

**3. Resampling.** When the generator $q$ diverges from the target $\pi_t$, most extensions $\{\mathbf{x}_t^i\}_{i=1}^N$ receive low scores, leading to particle degeneracy (Naesseth et al., 2019) and low sampling efficiency. TSMC-TTS instead resamples prefixes according to a categorical distribution over self-normalized step-wise scores $w(\mathbf{x}_{1:t})$:

$$\mathbf{x}_{1:t}^i \leftarrow \mathbf{x}_{1:t}^{i'}, \quad i' \sim \text{Cat}\left(\left\{\frac{w(\mathbf{x}_{1:t}^i)}{\sum_{j=1}^N w(\mathbf{x}_{1:t}^j)}\right\}_{i=1}^N\right), \quad i = 1, \ldots, N, \tag{12}$$

where the prefix $\mathbf{x}_{1:t}^i$ at index $i$ is replaced by the resampled prefix $\mathbf{x}_{1:t}^{i'}$ from index $i'$. This process clones high-score prefixes while pruning low-score ones, thereby allocating more rollout opportunities to promising solutions. The resampled prefix $\mathbf{x}_{1:t}^{i'}$ is then used to continue generation at index $i$.

**Frequency estimation via score-weighted MV.** Repeating operations 1–3 until $N$ complete sequences are collected yields a candidate set. TSMC-TTS then applies score-weighted MV (as in Proposition 4.1) to estimate the answer frequency and selects the answer with the highest estimated frequency as the final answer:

$$a^* = \arg\max_a \hat{f}_a = \arg\max_a \sum_{i=1}^N \frac{W(\mathbf{x}_{1:T}^i)}{\sum_{j=1}^N W(\mathbf{x}_{1:T}^j)} \mathbb{I}\left(\text{Ans}(\mathbf{x}_{1:T}^i) = a\right). \tag{13}$$

## 5.2 RELATIVE VARIANCE UPPER BOUND

The above procedure highlights that TSMC-TTS relies on two key components: the generator $q$ and the verifier $\{\pi_t\}_{t=1}^T$. In this section we theoretically analyze how their coupling affects the bias–variance tradeoff.

**Theorem 5.1** (Unbiasedness). *TSMC-TTS yields an unbiased estimator under target distribution* $\sigma(\mathbf{x}_{1:T})$, *regardless of the generator $q$ and intermediate verifier* $\{\pi_t\}_{t=1}^{T-1}$, *as long as* $\psi(\mathbf{x}_{1:T}) = \mathcal{R}(\mathbf{x}_{1:T})$.

Proof is in App. D.1. Theorem 5.1 shows that unbiasedness is invariant to the generator $q$ and intermediate verifier $\{\pi_t\}_{t=1}^{T-1}$, whereas the following bound reveals that their coupling critically determines the variance.

**Theorem 5.2** (**Relative variance upper bound**). *Let $\mathcal{V}(\pi, q)$ denote the relative variance of the TSMC-TTS estimator. Then $\log\big(\mathcal{V}(\pi, q) + 1\big) \leq \mathcal{B}(\pi, q)$, holds, where*

$$\mathcal{B}(\pi, q) := \sum_{t=1}^{T} \frac{C_t}{N} \Big[ \underbrace{\mathcal{D}_{\mathrm{KL}}\big(\sigma(\mathbf{x}_{1:t}) \,\|\, \pi_t(\mathbf{x}_{1:t})\big)}_{\mathcal{L}_{\mathrm{Ver}} \text{ verifier gap}} + \underbrace{\mathbb{E}_{q^\pi(\mathbf{x}_{1:t-1})}\big[\mathcal{D}_{\mathrm{KL}}\big(q^\pi(\mathbf{x}_t \mid \mathbf{x}_{1:t-1}) \,\|\, q(\mathbf{x}_t \mid \mathbf{x}_{1:t-1}))\big]}_{\mathcal{L}_{\mathrm{Gen}} \text{ generator gap}} \Big],$$

(14)

*and $C_t := \sup_{\mathbf{x}_{1:t}} w(\mathbf{x}_{1:t})$ is the essential supremum of the step-wise score at step $t$, and $N$ is the candidate size. Equality holds if the generator $q$ achieves its partial optimum $q = q^\pi$. Proof is provided in App. D.2.*

The variance bound decomposes into two terms, each tied to an optimality condition:

- *Verifier gap $\mathcal{L}_{\mathrm{Ver}}$*: the divergence between $\pi_t(\mathbf{x}_{1:t})$ and its *global optimum* $\sigma(\mathbf{x}_{1:t})$.
- *Generator gap $\mathcal{L}_{\mathrm{Gen}}$*: the divergence between $q(\mathbf{x}_t \mid \mathbf{x}_{1:t-1})$ and its *partial optimum* $q^\pi(\mathbf{x}_t \mid \mathbf{x}_{1:t-1})$.

We next formalize these optimality conditions, including the global optimum and the partial optimum.

**Proposition 5.3** (Global optimality, (Zhao et al., 2024b)). *The generator-verifier coupling achieves zero relative variance of the TSMC estimator (i.e., $\mathcal{V}(\pi^*, q^*) = 0$) if and only if:*

$$(i) \ \pi_t^*(\mathbf{x}_{1:t}) = \sigma(\mathbf{x}_{1:t}) \quad \forall 1 \leq t \leq T, \quad (ii) \ q^*(\mathbf{x}_t | \mathbf{x}_{1:t-1}) = \frac{\sigma(\mathbf{x}_{1:t})}{\sigma(\mathbf{x}_{1:t-1})} \quad \forall 1 \leq t \leq T.$$

(15)

**Proposition 5.4** (Partial optimality). *Given the fixed verifier $\{\pi_t\}_{t=1}^{T}$, the generator that minimizes the relative variance $\mathcal{V}(\pi, q)$ (i.e., $\frac{\partial \mathcal{V}(\pi, q)}{\partial q}\big|_{q=q^\pi} = 0$) is*

$$q^\pi(\mathbf{x}_t \mid \mathbf{x}_{1:t-1}) \propto \frac{\tilde{\pi}_t(\mathbf{x}_{1:t})}{\tilde{\pi}_{t-1}(\mathbf{x}_{1:t-1})}.$$

(16)

*In particular, when the verifier reaches its global optimum $\pi_t = \pi_t^*$ for all $1 \leq t \leq T$, the corresponding partially optimal generator $q^\pi$ matches exactly to the global optimum $q^*$. Proof is provided in App. D.3.*

**Theoretical insights.** The upper bound in Theorem 5.2 yields several key practical implications:

- *Surrogate for variance reduction.* Minimizing the variance $\mathcal{V}(\pi, q)$ is intractable due to non-differentiability. The bound becomes tight at the generator's partial optimum, providing a tractable surrogate objective.
- *Forward KL as variance control.* The bound explains why minimizing forward KL effectively reduces variance and provides a theoretical rationale for empirical findings on its advantages over reverse KL (Wang et al., 2024; Li et al., 2025; Song et al., 2025).
- *Coupling between generator and verifier.* Directly optimizing the generator and verifier toward their global optimum can be unstable due to the complex variance landscape. Our bound uncovers a synchronous coupling structure between the generator and verifier: as the verifier approaches its global optimum, the generator should adaptively align with the corresponding partial optimum, thereby ensuring stable variance reduction.

**Theoretical extensions beyond idealized assumptions.** To complement our main theoretical results, we provide three further theorems that extend our framework to more practical settings. Specifically, we analyze the impact of three sources of real-world imperfection: (i) approximate RFT optimization in App. C.3, (ii) imperfect verifiers in App. C.4, and (iii) imperfect reward models in App. C.5. These additional theorems establish explicit error decompositions under these practical settings and demonstrate that our conclusions remain robust under realistic deviations from ideal assumptions.

### 5.3 JOINT GENERATOR-VERIFIER OPTIMIZATION (ALG. 1 IN APP. A)

Building on the bound in Theorem 5.2, we propose an EM-like optimization algorithm that alternately updates the generator and verifier at each epoch $k$, ensuring monotonic variance reduction.

$$(\text{E-step}) \quad q^{(k+1)} \leftarrow \arg\min_q \mathcal{B}(\pi^{(k)}, q^{(k)}) = \arg\min_q \mathcal{L}_{\mathrm{Gen}}$$

$$(\text{M-step}) \quad \pi^{(k+1)} \leftarrow \arg\min_\pi \mathcal{B}(\pi^{(k)}, q^{(k+1)}) = \arg\min_\pi \mathcal{L}_{\mathrm{Ver}}$$

(17)

In the *E-step*, the generator is updated towards its partial optimum, thereby tightening the bound to the actual relative variance according to the equality condition in Theorem 5.2. In the *M-step*, the verifier is updated towards its global optimum, further reducing the bound.

A key property of the proposed EM-like optimization is its provable monotonic variance reduction:

**Property 5.5** (Monotonicity and convergence)**.** *At each update, the variance is monotonically reduced as:*

$$\mathcal{V}(\pi^{(k+1)}, q^{(k+1)}) \ \leq \ \mathcal{B}(\pi^{(k+1)}, q^{(k+1)}) \ \leq \ \mathcal{B}(\pi^{(k)}, q^{(k+1)}) = \ \mathcal{V}(\pi^{(k)}, q^{(k+1)}) \ \leq \ \mathcal{V}(\pi^{(k)}, q^{(k)}). \tag{18}$$

*Since $\mathcal{V}(\pi, q) \geq 0$, the sequence $\{\mathcal{V}(\pi^{(k)}, q^{(k)})\}_{k \geq 0}$ is bounded below, therefore converging to a finite limit. Proof is provided in App. D.4.*

**Gradient computation and analysis.** Since $\mathcal{L}_{\text{Gen}}$ and $\mathcal{L}_{\text{Ver}}$ defined in Eq. (14) involve expectations over intractable target distributions $\sigma(\mathbf{x}_{1:t})$ and $q^\pi(\mathbf{x}_{1:t})$, we efficiently estimate the expectation via IS or TSMC, using the generator $q(\mathbf{x}_{1:t})$ as the proposal distribution. The gradients are derived as

$$-\nabla_q \mathcal{L}_{\text{Gen}} = \sum_{t=1}^{T} \sum_{i=1}^{B} \left( \frac{\psi\left(\mathbf{x}_{1:t}^i\right)}{\sum_{j=1}^{B} \psi\left(\mathbf{x}_{1:t}^j\right)} \nabla_q \log q\left(\mathbf{x}_t^i \mid \mathbf{x}_{1:t-1}^i\right) \right). \tag{19}$$

$$-\nabla_\psi \mathcal{L}_{\text{Ver}} = \sum_{t=1}^{T} \sum_{i=1}^{B} \left( \frac{\mathcal{R}\left(\mathbf{x}_{1:T}^i\right)}{\sum_{j=1}^{B} \mathcal{R}\left(\mathbf{x}_{1:T}^j\right)} - \frac{\psi\left(\mathbf{x}_{1:t}^i\right)}{\sum_{j=1}^{B} \psi\left(\mathbf{x}_{1:t}^j\right)} \right) \nabla_\psi \log \psi(\mathbf{x}_{1:t}^i). \tag{20}$$

Derivations are provided in App. D.5. The generator update in Eq. (19) upweights the generation probability of prefixes $\mathbf{x}_{1:t}$ favored by the current verifier $\psi(\mathbf{x}_{1:t})$, while the verifier further be refined (Eq. (20)) to align step-wise scores with the final reward $\mathcal{R}(\mathbf{x}_{1:T})$. Moreover, since Eq. (14) minimizes the forward KL divergence $\mathcal{D}_{\text{KL}}(\sigma \,\|\, \pi)$, training essentially use the samples from the target distribution $\sigma$ (via IS or TSMC). This fundamentally differs from methods like PPO, which minimizes the reverse KL divergence $\mathcal{D}_{\text{KL}}(q \,\|\, \sigma)$ and thus solely uses samples from generator, without exploiting target samples.

**Self-evolving TSMC mechanism.** Unlike prior work (Zhao et al., 2024b) that fixes $\sigma(\mathbf{x}_{1:T}) \propto p_0(\mathbf{x}_{1:T})\mathcal{R}(\mathbf{x}_{1:T})$, we adopt a *self-evolving* distribution $\sigma(\mathbf{x}_{1:T}) \propto q(\mathbf{x}_{1:T})\mathcal{R}(\mathbf{x}_{1:T})$, where $q$ is iteratively refined via Eq. (19). This elegant design naturally synchronizes the evolution of $\sigma$ with our EM-like updated over $q$. This enables continual improvement while avoiding (i) the sparsity of high-reward samples caused by the fixed support of $p_0$ and (ii) the extra forward passes required by $w(\mathbf{x}_{1:t}) = \frac{p_0(\mathbf{x}_{1:t})}{q(\mathbf{x}_{1:t})} \frac{\psi(\mathbf{x}_{1:t})}{\psi(\mathbf{x}_{1:t-1})}$, reducing it to $w(\mathbf{x}_{1:t}) = \frac{\psi(\mathbf{x}_{1:t})}{\psi(\mathbf{x}_{1:t-1})}$.

# 6 EXPERIMENTS

**Datasets and models.** Following prior work, we evaluate TSMC-TTS on two widely used math datasets: GSM8K (Cobbe et al., 2021) and MATH (Hendrycks et al., 2021). We report results on the full GSM8K test set and on a 500-instance subset of MATH (MATH500) following Lightman et al. (2024). We test two backbones including Llemma-7B (Azerbayev et al., 2024) and DeepSeekMath-7B (Shao et al., 2024a), and each is first fine-tuned on PRM800K (Lightman et al., 2024) to learn the reasoning input–output format following Sun et al. (2024). In each setting, the backbone parameterizes both the generator and the verifier (i.e., twist function), with the verifier added by an additional linear head and sigmoid activation.

**Baselines.** We conduct a comprehensive comparison covering both verifier-based and generator-based baselines by varying the verifier, generator, and aggregation strategy. For verifier-based methods, we include external verifiers such as ORMs and PRMs. For generator-based methods, we adopt PPO as RFT following (Jaques et al., 2020; Zhang et al., 2025). We also consider hybrid methods, which simply combine generator-based and verifier-based approaches, for example by pairing an RFT-trained generator with an off-the-shelf ORM or PRM verifier. Full implementation details are provided in App. G.

**Implementation details.** For training, we perform two overall epochs. Within each epoch, the generator and the verifier are each updated for two sub-epochs. For inference, we perform up to five resampling steps. For sequences that terminate early, we assign a step-wise score of 1 for the remaining resampling steps. Additional details such as learning rate and batch size are provided in App. G.

Table 2: Problem-solving rates (%) on GSM8K and MATH500 using Llemma-7B and DeepSeekMath-7B. PRM is trained on PRM800K or MATH-SHEPHERD (Wang et al., 2023). The number of candidates is 240. † indicates results reproduced by us under the same training conditions.

| Generator | Verifier | Aggregation | Llemma-7B | | DeepSeekMath-7B | |
|---|---|---|---|---|---|---|
| | | | GSM8K | MATH500 | GSM8K | MATH500 |
| Base | – | Greedy | 38.3 | 20.0 | 73.1 | 33.0 |
| Base | – | MV | 71.2 | 40.8 | 86.9 | 54.4 |
| Base | ORM | WMV | 77.6 | 42.6 | 88.2 | 56.2 |
| Base | PRM (PRM800K) | WMV | 77.8 | 42.2 | 88.6 | 56.8 |
| Base | PRM (SHEPHERD) | WMV | 78.8 | 43.2 | 89.2 | 55.4 |
| Base | Feng et al. (2025)† | WMV | 79.6 | 45.2 | 90.4 | 58.8 |
| RFT | – | Greedy | 40.5 | 23.6 | 75.3 | 35.5 |
| RFT | – | MV | 69.6 | 38.4 | 84.8 | 52.2 |
| RFT | ORM | WMV | 75.2 | 39.3 | 86.4 | 53.3 |
| RFT | PRM (PRM800K) | WMV | 75.9 | 41.8 | 87.2 | 54.3 |
| RFT | PRM (SHEPHERD) | WMV | 76.2 | 42.9 | 87.9 | 53.8 |
| RFT | Feng et al. (2025)† | WMV | 78.2 | 43.8 | 88.3 | 57.2 |
| TSMC-TTS | TSMC-TTS | WMV | **82.2** | **47.2** | **92.6** | **61.8** |

## 6.1 MAIN RESULTS

The comparative results in Tab. 2 show that our method consistently outperforms all baselines across datasets and backbone settings. Notably, it surpasses the baselines using external verifiers by over 4%, as these baselines lack explicit variance-control mechanisms, producing unstable outputs that ultimately lower overall performance. Our method also shows significant performance gains (over 5%) over the baselines using RFT-based generators, since these baselines introduce potential bias that anchors solutions to a narrow region and thereby limits scalability (Song et al., 2025). Moreover, our method significantly outperforms these naive combinations between generator-based and verifier-based approaches (i.e., RFT+ORM and RFT+PRM), which aligns with our bias–variance analysis: such direct combinations still inherit the generator's training bias and therefore scale poorly with larger solution budgets. In contrast, our method is designed to effectively reduce variance without compromising unbiasedness, thus achieving consistent and significant performance gains.

## 6.2 ABLATION STUDIES

**Scalability evaluation.** Fig. 2 shows that our method outperforms all baselines across varying candidate budgets. Verifier-based baselines (dark blue curve) suffer from high variance, resulting in poor performance at low budgets. Generator-based baselines (pink curve) scale poorly and even underperform the vanilla baseline, a phenomenon also observed by concurrent work (Li et al., 2025; Song et al., 2025). Beyond empirical findings, we attribute this suppressed scalability to inherent biases that anchor solutions in potentially incorrect regions, from an intrinsic bias–variance tradeoff perspective.

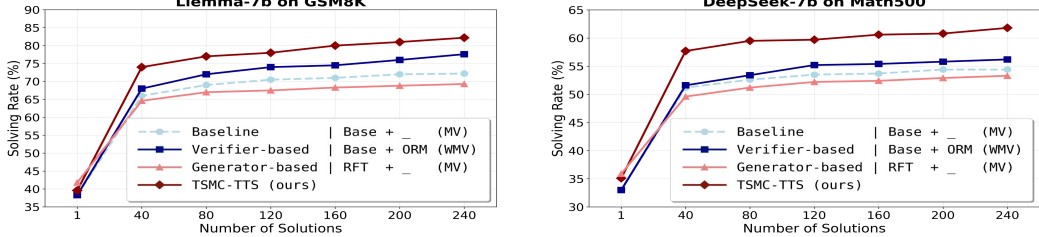

Figure 2: Parallel scaling performance on GSM8K and MATH500 using Llemma-7B and DeepSeekMath-7B.

**Effect of each component.** The ablation study in Tab. 3 shows both the generator and verifier updates contribute to performance gains. This observation is consistent with the theoretical bound in Theorem 5.2, suggesting that both components are critical in reducing variance. Due to space limit,

more experiments (e.g., resampling effect, complexity analysis, hyperparameter) and limitations are provided in Apps. B and E.

Table 3: Ablation study on MATH500 with Llemma-7b. "A + B (C)" indicates generator A + verifier B with aggregation strategy C.

| Method | Ablation | Solving Rate (%) |
|---|---|---|
| Base + _ (MV) | baseline | 23.6 |
| Base + TSMC (WMV) | + verifier (Eq. (20)) | 45.8 |
| TSMC + TSMC (WMV) | + generator (Eq. (19)) | 47.2 |

**Results on generalized reasoning datasets.** To further assess the effectiveness of our method, we extend our experiments to a broader set of reasoning datasets. In particular, we include evaluations on two generalized reasoning benchmarks: FinanceBench (Islam et al., 2023) (financial domain) and NumGLUE Task 2 Chemistry (Mishra et al., 2022) (scientific domain). We additionally evaluate our method on the Llama-3.1-8B-Instruct model. As shown in Tab. 4, our method consistently and substantially outperforms all baselines. These results demonstrate that our approach is not only highly effective for mathematical reasoning but also generalizes well to diverse domain-specific reasoning scenarios.

Table 4: Performance on generalized domain-specific reasoning tasks.

| Method | FinanceBench | NumGLUE Task 2 Chemistry |
|---|---|---|
| ORM | 67.67 | 79.69 |
| PRM | 68.33 | 80.92 |
| TSMC-TTS (ours) | 72.67 | 86.92 |

## 7 CONCLUSION

In this work, we establish the first general framework for PTTS through a unifying probabilistic inference formulation, seamlessly encompassing previously disparate methods as special cases. This framework reveals a fundamental bias–variance tradeoff and provides a principled foundation for developing new methods. Our framework reveals that verifier-based methods act as high-variance IS estimators, yielding marginal gains under small scaling budgets, whereas generator-based methods act as biased VI estimators, yielding suppressed scalability even under large budgets. We further derive the first theoretical bias–variance formulation for PTTS, showing that the relative variance bound is jointly governed by the optimality gaps of the generator and verifier. Building on our framework, we propose a theory-driven PTTS method named TSMC-TTS effectively mitigating the bias–variance tradeoff. We further introduce a self-evolving TSMC mechanism, effectively alleviating the reward sparsity and computational overhead issues inherent in vanilla TSMC. Rigorous theoretical analysis and comprehensive experiments substantiate the efficacy of the proposed method.

## ETHICS STATEMENT

This work introduces a method to improve the test-time scaling performance of large language models. The proposed approach may also pave the way for inference-time solutions to achieve personalized alignment, which is particularly important when the target distribution might be adversarially designed to encode socially consequential values. We affirm that this work does not involve human subjects, sensitive or personally identifiable data, or practices that could give rise to ethical concerns. No conflicts of interest are present.

## REPRODUCIBILITY STATEMENT

Detailed proofs and derivations for all propositions, theorems, and those formulations requiring formal justification are provided in Apps. C and D. Comprehensive implementation details for both our method and the baselines are included in Sec. 6 and App. G to ensure the reproducibility of our experiments.

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

# Appendix of A Bias–Variance Tradeoff Perspective for Improving Test-Time Scaling

## Contents

# A  ALGORITHMS

---

**Algorithm 1:** TSMC-TTS (Train)

---

**Input:** Generator $q$ and verifier (i.e., twisted function) $\psi$. Training prompt set $\mathcal{X}_0$. Reward function $\mathcal{R}(\cdot)$.

Initialize the generator $q^{(0)}$ and twisted function $\psi^{(0)}$.

$k \leftarrow 0$

**while** $k < E_{\text{TSMC}}$ **do**

    Collect $B$ rollouts for each prompt $\mathbf{x}_0 \in \mathcal{X}_0$: $\mathbf{x}_{1:T} \sim q^{(k)}(\cdot \mid \mathbf{x}_0)$ or via TSMC-TTS inference with

    $(q^{(k)}, \psi^{(k)})$

    **if** $k \neq 0$ **then**

        $q^{(k+1)} \leftarrow \arg\min_q \mathcal{L}_{\text{Gen}}(q^{(k)})$ by $E_{\text{Gen}}$-epoch SGD with gradient in Eq. (19)

    $\psi^{(k+1)} \leftarrow \arg\min_\psi \mathcal{L}_{\text{Ver}}(\psi^{(k)})$ by $E_{\text{Ver}}$-epoch SGD with gradient in Eq. (20)

    $k \leftarrow k + 1$

**Return** Trained generator $q^{(k)}$ and verifier $\psi^{(k)}$

---

Note that in Alg. 1, at the first epoch ($k = 0$), we deliberately skip updating the generator $q^{(0)}$ so as to avoid noisy gradients from Eq. (19) when the verifier $\psi^{(0)}$ is untrained.

---

**Algorithm 2:** TSMC-TTS (Inference)

---

**Input:** Trained generator $q$ and twisted function $\psi$. Parallel scaling budget $N$. Max step $T$.

`// Candidate generation and scoring`

**for** $t \leftarrow 1$ **to** $T$ **do**

    **for** $i \leftarrow 1$ **to** $N$ **do**

        $\mathbf{x}_t^i \sim q(\cdot \mid \mathbf{x}_{1:t-1}^i)$ `// generate extension from prefix`

        $\mathbf{x}_{1:t}^i \leftarrow \text{CONCAT}\left(\mathbf{x}_{1:t-1}^i, \mathbf{x}_t^i\right)$ `// concatenate extension with prefix`

        **if** $t < T$ **then**

            $w(\mathbf{x}_{1:t}^i) = \frac{\psi(\mathbf{x}_{1:t}^i)}{\psi(\mathbf{x}_{1:t-1}^i)}$ `// step-wise score in Eq. (10)`

        **else if** $t = T$ **then**

            $W(\mathbf{x}_{1:T}^i) = \psi(\mathbf{x}_{1:T}^i)$ `// sequence-wise score in Eq. (11)`

    **if** $t < T$ **then**

        **for** $i \leftarrow 1$ **to** $N$ **do**

            $\mathbf{x}_{1:t}^i \leftarrow \mathbf{x}_{1:t}^{i'}, \quad i' \sim \text{Cat}\left(\left\{\frac{w(\mathbf{x}_{1:t}^i)}{\sum_{j=1}^N w(\mathbf{x}_{1:t}^j)}\right\}_{i=1}^N\right)$ `// resampling in Eq. (12)`

`// Score-weighted majority voting in Eq. (13)`

$D \leftarrow \text{Dict}()$ `// empty dictionary to record the total scores of each answer`

**for** $i \leftarrow 1$ *to* $N$ **do**

    $a^i \leftarrow \text{Ans}(\mathbf{x}_{1:T}^i)$ `// extract the answer`

    `// Select the answer with the highest total scores`

    **if** $a^i \in W$ **then**

        $D[a^i] \leftarrow D[a^i] + W(\mathbf{x}_{1:T}^i)$

    **else**

        $D[a^i] \leftarrow W(\mathbf{x}_{1:T}^i)$

**Return** $\arg\max_a D[a]$

---

# B  MORE DISCUSSIONS

## B.1  LIMITATIONS

**Theorem (Bias decomposition into variational and optimization components).** Assume the true target distribution is $\sigma(\mathbf{x}_{1:T}) \propto p_0(\mathbf{x}_{1:T})R(\mathbf{x}_{1:T})$ with normalizing constant $Z_\sigma = \int p_0(\mathbf{x}_{1:T})R(\mathbf{x}_{1:T})d\mathbf{x}_{1:T}$. Let $\mathcal{Q}$ be a neural-network–parameterized family of generators and define the variational optimum

$$q_{\mathrm{VI}} = \arg\min_{q \in \mathcal{Q}} \mathrm{KL}(q(\mathbf{x}_{1:T})\|\sigma(\mathbf{x}_{1:T})).$$

Let $\tilde{q}$ denote the generator obtained by a practical optimizer (e.g., RFT fine-tuning) and assume its suboptimality is bounded by

$$\mathrm{KL}(\tilde{q}\|\sigma) \leq \mathrm{KL}(q_{\mathrm{VI}}\|\sigma) + \varepsilon_{\mathrm{opt}}.$$

Consider the VI normalizing-constant estimator $\hat{Z}_\sigma^{\mathrm{VI}}(q) = Z_\sigma \exp(-\mathrm{KL}(q\|\sigma))$. Then the estimation bias decomposes as

$$\left|Z_\sigma - \hat{Z}_\sigma^{\mathrm{VI}}(\tilde{q})\right| \leq Z_\sigma\big(1 - e^{-\mathrm{KL}(q_{\mathrm{VI}}\|\sigma)}\big) + Z_\sigma\big(1 - e^{-\varepsilon_{\mathrm{opt}}}\big),$$

where the first term corresponds to the variational bias (due to the restricted family $\mathcal{Q}$) and the second term corresponds to the optimization bias (due to not reaching $q_{\mathrm{VI}}$ exactly). In particular,

$$Z_\sigma\big(1 - e^{-\varepsilon_{\mathrm{opt}}}\big) = O(\varepsilon_{\mathrm{opt}}),$$

so the optimization bias scales linearly in the optimization suboptimality and does not grow with the sequence length $T$.

A limitation of our approach is the need to load both the generator and the verifier at inference time, which introduces additional computational overhead relative to standard parallel decoding. However, this extra cost is compensated by the substantial performance gains reported in our main experiments, while the detailed time- and memory-cost analysis is provided in App. E.3. Moreover, practical engineering strategies such as CPU offloading, model quantization, and distributed inference frameworks (e.g., pipeline or tensor parallelism) can further alleviate latency and balance the computational load for practical deployment.

Crucially, our method does not incur extra overhead in terms of generation, since the intermediate resampling step simply replicates higher-scoring prefixes to replace lower-scoring ones, rather than generating additional candidate steps (e.g., backtracking (Singh et al., 2025)). By avoiding redundant computation during generation, this streamlined design keeps the approach compatible with future acceleration techniques and preserves strong potential for real-time applications.

## B.2  DIFFERENCE FROM EXISTING PROBABILISTIC INFERENCE SCALING METHODS

Recent studies Feng et al. (2025); Puri et al. (2025) seek to enhance PTTS by incorporating probabilistic inference techniques. However, our approach fundamentally differs from these methods in several key aspects.

**1. First general framework for PTTS under the lens of the bias–variance tradeoff.** We propose the first general framework that recasts PTTS as probabilistic inference, unifying existing methods as special cases. This framework offers a new bias–variance tradeoff perspective to reveal the intrinsic limitations of existing methods and provides a principled foundation for developing new methods. In contrast, prior studies have not yet introduced or explored such a general and versatile conceptual framework for PTTS.

**2. First theoretical formulation of the bias-variance tradeoff in the context of PTTS.** We present the first theoretical formulation (Theorem 5.2) of the bias–variance tradeoff in the context of PTTS, which quantitatively characterizes how the relative variance upper bound is jointly governed by the generator and the verifier, as reflected by its decomposition into their respective gaps. We believe such a theoretical perspective offers valuable insights for the PTTS community largely driven by heuristics. In contrast, existing studies have not explored such a theoretical characterization of the bias–variance tradeoff in PTTS.

**3. Theory-driven generator-verifier optimization algorithm.** Our theoretical insight enables a principled and theory-driven verifier–generator optimization algorithm. Under our variance decomposition, which reveals both a verifier gap and a generator gap, prior work is provably suboptimal due

to its fundamental incompatibility with the conditions required for optimal variance reduction. In contrast, our EM-like optimization jointly updates the generator and verifier by directly targeting the variance bound, achieving monotonic variance reduction without compromising unbiasedness.

**4. Self-evolving TSMC mechanism.** We introduce a new self-evolving TSMC mechanism that uses a self-evolving target distribution proportional to $q(\mathbf{x}_{1:T})R(\mathbf{x}_{1:T})$. This design mitigates both the reward sparsity issue and the substantial computational overhead inherent in prior work, which stems from its reliance on a fixed base distribution $p_0(\mathbf{x}_{1:T})$. As a result, our approach significantly enhances the practical applicability of existing work.

Tab. 5 summarizes the distinctions between our method and prior probabilistic inference scaling approaches.

Table 5: Comprehensive comparison between our proposed TSMC-TTS and prior probabilistic inference scaling methods. Although Feng et al. (2025) and Puri et al. (2025) attempt to incorporate probabilistic inference to improve PTTS, they overlook the intrinsic bias–variance tradeoff inherent to such probabilistic inference techniques, leaving their framework design (i.e., the verifier and generator components) fundamentally vulnerable to this core limitation. In addition, within our general framework, these methods can be viewed as targeting a fixed distribution, whereas TSMC-TTS employs a self-evolving target distribution in which the generator $q^*$ is iteratively refined via Eq. (19), thereby alleviating the sparsity of high-reward samples under $p_0$ and simplifying the computation of step-wise scores. Discussions are provided in App. B.2.

| Method | Generator | Verifier | Target dist. $(\sigma \propto p \cdot \phi)$ | Addresses bias–variance tradeoff? |
|---|---|---|---|---|
| Feng et al. (2025) | Fixed (pre-trained backbone) | Learnable (trained from the generator backbone) | $p_0(\mathbf{x}_{1:T}) \cdot \mathcal{R}(\mathbf{x}_{1:T})$ (fixed) | ✗ |
| Puri et al. (2025) | Fixed (pre-trained backbone) | Fixed (pre-trained PRM, Qwen2.5-Math-PRM-7B) | $p_0(\mathbf{x}_{1:T}) \cdot \mathcal{R}(\mathbf{x}_{1:T})$ (fixed) | ✗ |
| **TSMC-TTS (Ours)** | **Learnable** (trained from the generator backbone) | **Learnable** (compatible with both training from the generator backbone and fine-tuning from a pre-trained PRM) | $q^*(\mathbf{x}_{1:T}) \cdot \mathcal{R}(\mathbf{x}_{1:T})$ (**self-evolving**) | ✓ |

## C  MORE THEOREMS

### C.1  VERIFIER-BASED METHODS AS IS ESTIMATORS (FULL VERSION OF PROPOSITION 4.1)

**Proposition C.1 (Verifier-based methods as IS estimators (Full version of Proposition 4.1)).**
*Let $q(\mathbf{x}_{1:T})$ be the generator and suppose the verifier assigns each candidate $\mathbf{x}_{1:T} \sim q(\mathbf{x}_{1:T})$ a sequence-level score $W(\mathbf{x}_{1:T}) = \mathrm{Aggr}\left[w(\mathbf{x}_{1:t})\right]_{t=1}^{T}$, where $\mathrm{Aggr}(\cdot)$ is a combination operator such as $\min$ or $\prod$ over step-wise scores $w(\mathbf{x}_{1:t})$. Verifier-based methods then select the answer with the highest total score as the final answer from $N$ candidates (i.e., score-weighted MV (Wu et al., 2025)):*

$$a^* = \arg\max_a \sum_{i=1}^{N} W(\mathbf{x}_{1:T}^i)\mathbb{I}\left(\mathrm{Ans}(\mathbf{x}_{1:T}^i) = a\right), \quad \mathbf{x}_{1:T}^i \sim q(\mathbf{x}_{1:T}). \tag{21}$$

*Consider a reward-aligned target distribution defined as a special case of Eq. (1): $\sigma(\mathbf{x}_{1:T}) = \frac{1}{Z_\sigma}q(\mathbf{x}_{1:T})\mathcal{R}(\mathbf{x}_{1:T}) = \frac{1}{Z_\sigma}q(\mathbf{x}_{1:T})W(\mathbf{x}_{1:T})$, where the second equality assumes $W(\mathbf{x}_{1:T})$ accurately*

approximates $\mathcal{R}(\mathbf{x}_{1:T})$ at inference time. By IS, the answer frequency $f_a$ in Eq. (2) can be estimated as

$$f_a = \mathbb{E}_{\sigma(\mathbf{x}_{1:T})}\left[\mathbb{I}\big(\mathrm{Ans}(\mathbf{x}_{1:T}) = a\big)\right] = \frac{1}{Z_\sigma}\mathbb{E}_{q(\mathbf{x}_{1:T})}\left[\frac{\tilde{\sigma}(\mathbf{x}_{1:T})}{q(\mathbf{x}_{1:T})}\mathbb{I}\left(\mathrm{Ans}(\mathbf{x}_{1:T}) = a\right)\right] = \frac{\mathbb{E}_{q(\mathbf{x}_{1:T})}\left[\frac{\tilde{\sigma}(\mathbf{x}_{1:T})}{q(\mathbf{x}_{1:T})}\mathbb{I}\left(\mathrm{Ans}(\mathbf{x}_{1:T}) = a\right)\right]}{\mathbb{E}_{q(\mathbf{x}_{1:T})}\left[\frac{\tilde{\sigma}(\mathbf{x}_{1:T})}{q(\mathbf{x}_{1:T})}\right]}$$

$$\approx \sum_{i=1}^{N}\frac{W(\mathbf{x}_{1:T}^i)}{\sum_{j=1}^N W(\mathbf{x}_{1:T}^j)}\mathbb{I}\big(\mathrm{Ans}(\mathbf{x}_{1:T}^i) = a\big) \triangleq \hat{f}_a^{\mathrm{IS}}, \quad \mathbf{x}_{1:T} \sim q(\mathbf{x}_{1:T}), \tag{22}$$

where the generator $q(\mathbf{x}_{1:T})$ plays the role of the proposal distribution and $\frac{\tilde{\sigma}(\mathbf{x}_{1:T})}{q(\mathbf{x}_{1:T})} = W(\mathbf{x}_{1:T})$ acts as the unnormalized importance weight in IS. $\hat{f}_a^{\mathrm{IS}}$ denotes the estimated answer frequency via IS. *The approximation is justified by the Monte Carlo estimation of expectations and its asymptotically vanishing error (Owen, 2013).*

Hence, verifier-based methods in Eq. (3) is equivalent to selecting the highest-frequency answer under $\sigma$ estimated by IS (i.e., Eq. (4), the denominator $\sum_{j=1}^N W(\mathbf{x}_{1:T}^j)$ is constant and thus irrelevant to maximization):

$$\arg\max_a \sum_{i=1}^{N} W(\mathbf{x}_{1:T}^i)\mathbb{I}\big(\mathrm{Ans}(\mathbf{x}_{1:T}^i) = a\big), \quad \mathbf{x}_{1:T} \sim q(\mathbf{x}_{1:T}) \iff \arg\max_a \hat{f}_a^{\mathrm{IS}}. \tag{23}$$

## C.2 GENERATOR-BASED METHODS AS VI ESTIMATORS (FULL VERSION OF PROPOSITION 4.2)

**Proposition C.2 (Generator-based methods as VI estimators (Full version of Proposition 4.2)).** *Let $q(\mathbf{x}_{1:T})$ denote the generator and suppose it is optimized via reinforcement fine-tuning (RFT) to produce high-reward candidates by typically minimizing the policy-gradient loss following (Jaques et al., 2020; Schulman et al., 2017; Zhang et al., 2025) (with $\beta > 0$ the KL-regularization coefficient):*

$$q_{\mathrm{RFT}} = \arg\min_q \mathcal{L}_{\mathrm{RFT}} = \arg\min_q \left[-\mathbb{E}_{\mathbf{x}_{1:T} \sim q}\big[\mathcal{R}(\mathbf{x}_{1:T})\big] + \beta\,\mathcal{D}_{\mathrm{KL}}\big(q(\mathbf{x}_{1:T}) \,\|\, p_0(\mathbf{x}_{1:T})\big)\right]. \tag{24}$$

*Generator-based methods then select the final answer from $N$ candidates generated the optimized generator $q_{\mathrm{RFT}}$ via MV (Wang et al., 2022):*

$$a^* = \arg\max_a \sum_{i=1}^{N}\mathbb{I}\big(\mathrm{Ans}(\mathbf{x}_{1:T}^i) = a\big), \quad \mathbf{x}_{1:T}^i \sim q_{\mathrm{RFT}}(\mathbf{x}_{1:T}). \tag{25}$$

*Consider a reward-aligned target distribution defined as a special case of Eq. (1): $\sigma(\mathbf{x}_{1:T}) = \frac{1}{Z_\sigma}p_0(\mathbf{x}_{1:T})\exp(\mathcal{R}(\mathbf{x}_{1:T})/\beta)$. By VI, the answer frequency $f_a$ in Eq. (2) can be estimated as*

$$f_a = \mathbb{E}_{\sigma(\mathbf{x}_{1:T})}\left[\mathbb{I}\big(\mathrm{Ans}(\mathbf{x}_{1:T}) = a\big)\right] \approx \mathbb{E}_{q_{\mathrm{VI}(\mathbf{x}_{1:T})}}\left[\mathbb{I}\big(\mathrm{Ans}(\mathbf{x}_{1:T}) = a\big)\right]$$

$$\approx \sum_{i=1}^{N}\mathbb{I}\big(\mathrm{Ans}(\mathbf{x}_{1:T}^i) = a\big) \triangleq \hat{f}_a^{\mathrm{VI}}, \quad \mathbf{x}_{1:T} \sim q_{\mathrm{VI}}(\mathbf{x}_{1:T}), \tag{26}$$

where $\hat{f}_a^{\mathrm{VI}}$ denotes the estimated answer frequency via VI. The first approximation holds since VI aims to optimize $q_{\mathrm{VI}}(\mathbf{x}_{1:T})$ as a surrogate for the intractable $\sigma(\mathbf{x}_{1:T})$ by minimizing the KL divergence:

$$q_{\mathrm{VI}} = \arg\min_q \mathcal{L}_{\mathrm{VI}} = \arg\min_q \left[\mathcal{D}_{\mathrm{KL}}\big(q(\mathbf{x}_{1:T}) \,\|\, \sigma(\mathbf{x}_{1:T})\big)\right] \tag{27}$$

Based on the equivalence between RFT and VI objective, i.e., $q_{\mathrm{RFT}} \iff q_{\mathrm{VI}}$ (see App. D.6 for proof), generator-based methods in Eq. (25) is equivalent to selecting the answer the highest-frequency answer under $\sigma$ estimated by VI (i.e., Eq. (26)):

$$\arg\max_a \sum_{i=1}^{N}\mathbb{I}\big(\mathrm{Ans}(\mathbf{x}_{1:T}^i) = a\big), \quad \mathbf{x}_{1:T} \sim q_{\mathrm{RFT}}(\mathbf{x}_{1:T}) \iff \arg\max_a \hat{f}_a^{\mathrm{VI}}. \tag{28}$$

### C.3 Bias Decomposition under Approximate RFT Optimization

The generator obtained by practical RFT fine-tuning does not exactly match the ideal minimizer of the RFT objective. This discrepancy indeed reflects an optimization bias. To address this concern, we extend our theoretical framework to formally characterize this bias and provide the following theorem in the updated version.

**Theorem C.3** (Bias decomposition under approximate RFT optimization). *Let the true target distribution be $\sigma\left(\mathbf{x}_{1:T}\right) \propto p_0\left(\mathbf{x}_{1:T}\right) R\left(\mathbf{x}_{1:T}\right)$ with normalizing constant $Z_\sigma = \int p_0\left(\mathbf{x}_{1:T}\right) R\left(\mathbf{x}_{1:T}\right) d\mathbf{x}_{1:T}$. Define the optimum regarding the RFT objective and VI objective within the parametric family induced by the generator network as $q_{\mathrm{RFT}} \iff q_{\mathrm{VI}} = \arg\min_q \mathrm{KL}(q\left(\mathbf{x}_{1:T}\right) \| \sigma\left(\mathbf{x}_{1:T}\right))$. Let $q'$ denote the generator obtained by a practical optimizer (e.g., RFT fine-tuning) and assume its suboptimality is bounded by $\mathrm{KL}(q'\|\sigma) \leq \mathrm{KL}\left(q_{\mathrm{VI}}\|\sigma\right) + \varepsilon_{\mathrm{opt}}$. Consider the VI normalizing-constant estimator $\hat{Z}_\sigma^{\mathrm{VI}}(q') = Z_\sigma \exp(-\mathrm{KL}(q'\|\sigma))$ (derivations are provided in the proof). Then the estimation bias satisfies*

$$\left|Z_\sigma - \hat{Z}_\sigma^{\mathrm{VI}}(q')\right| \leq Z_\sigma \left(1 - e^{-\mathrm{KL}(q_{\mathrm{VI}}\|\sigma)}\right) + Z_\sigma \left(1 - e^{-\varepsilon_{\mathrm{opt}}}\right) = O\left(Z_\sigma(1 - e^{-C_{\mathrm{KL}} \cdot T})\right) + O(\varepsilon_{\mathrm{opt}}),$$

*where the first term corresponds to the variational mode-seeking bias, and the second term corresponds to the optimization bias (due to not reaching the optimum exactly). Here $C_{\mathrm{KL}} > 0$ is a max per-step KL divergence such that the sequence-level KL divergence satisfies $\mathrm{KL}(q_{\mathrm{VI}}|\sigma') \leq C_{\mathrm{KL}} \cdot T$ under the standard autoregressive factorization over $T$ steps. The second term follows from $1 - e^{-x} = O(x)$ as $x \to 0$, which applies when the optimization error becomes near 0.*

**Remark.** The above theorem shows that the total bias under practical RFT optimization decomposes into a variational mode-seeking bias and an optimization bias. Crucially, only the variational term scales with the step number $T$, while the optimization term does not accumulate with the number of steps. Therefore, even though practical fine-tuning produces an approximate minimizer, its optimization bias remains lower order and is dominated by the variational term. Therefore, this theorem extends our theoretical framework to settings with approximate RFT optimization and shows that our original claim continues to hold under such approximations.

*Proof.* Let the true target be

$$\sigma\left(\mathbf{x}_{1:T}\right) \propto p_0\left(\mathbf{x}_{1:T}\right) R\left(\mathbf{x}_{1:T}\right), \quad Z_\sigma = \int p_0\left(\mathbf{x}_{1:T}\right) R\left(\mathbf{x}_{1:T}\right) d\mathbf{x}_{1:T}.$$

Next, consider the VI estimator $Z_\sigma$ under the target distribution $\sigma$. By the standard variational identity,

$$\log Z_\sigma = \mathrm{ELBO}(q) + \mathrm{KL}\left(q\|\sigma\right),$$

we have

$$\hat{Z}_\sigma^{\mathrm{VI}} = \exp(\mathrm{ELBO}(q)) = \exp\left(\log Z_\sigma - \mathrm{KL}\left(q\|\sigma\right)\right) = Z_\sigma \exp\left(-\mathrm{KL}\left(q\|\sigma\right)\right).$$

Thus,

$$\left|Z_\sigma - \hat{Z}_\sigma^{\mathrm{VI}}(q)\right| = Z_\sigma(1 - \exp(-\mathrm{KL}(q\|\sigma))).$$

Let

$$q_{\mathrm{VI}} = \arg\min_q \mathrm{KL}(q\|\sigma)$$

within the generator family and let the practical optimizer return $q'$ satisfying

$$\mathrm{KL}\left(q'\|\sigma\right) \leq \mathrm{KL}\left(q_{\mathrm{VI}}\|\sigma\right) + \varepsilon_{\mathrm{opt}}.$$

We obtain

$$1 - e^{-\mathrm{KL}\left(q'\|\sigma\right)} \leq 1 - e^{-(\mathrm{KL}(q_{\mathrm{VI}}\|\sigma)+\varepsilon_{\mathrm{opt}})} = 1 - e^{-\mathrm{KL}(q_{\mathrm{VI}}\|\sigma)} + e^{-\mathrm{KL}(q_{\mathrm{VI}}\|\sigma)}\left(1 - e^{-\varepsilon_{\mathrm{opt}}}\right).$$

Multiplying by $Z_\sigma$ yields the desired decomposition:

$$\left|Z_\sigma - \hat{Z}_\sigma^{\mathrm{VI}}(q')\right| \leq Z_\sigma \left(1 - e^{-\mathrm{KL}(q_{\mathrm{VI}}\|\sigma)}\right) + Z_\sigma \left(1 - e^{-\varepsilon_{\mathrm{opt}}}\right).$$

Using the standard autoregressive factorization, there exists the max per-step KL divergence $C_{\text{KL}} > 0$ such that $\text{KL}\left(q_{\text{VI}}\|\sigma\right) \leq C_{\text{KL}}T$. Thus,

$$Z_\sigma \left(1 - e^{-\text{KL}(q_{\text{VI}}\|\sigma)}\right) = O\left(Z_\sigma \left(1 - e^{-C_{\text{KL}}T}\right)\right).$$

Since $1 - e^{-x} = O(x)$ when $x \to 0$,

$$Z_\sigma \left(1 - e^{-\varepsilon_{\text{opt}}}\right) = O\left(\varepsilon_{\text{opt}}\right),$$

which does not depend on the step number $T$. Combine both to obtain:

$$\left|Z_\sigma - \hat{Z}_\sigma^{\text{VI}}\left(q'\right)\right| \leq O\left(Z_\sigma \left(1 - e^{-C_{\text{KL}}T}\right)\right) + O\left(\varepsilon_{\text{opt}}\right).$$

This completes the proof. $\qquad\square$

### C.4 ERROR BOUNDS FOR TSMC-TTS ESTIMATOR UNDER IMPERFECT VERIFIERS

The unbiasedness property of our method is based on the assumption of an oracle verifier (i.e., $\psi(\mathbf{x}_{1:T}) = R(\mathbf{x}_{1:T})$). In practice, however, $\psi$ is learned and inevitably imperfect, which introduces verifier-induced bias. We derive a formal error bound theorem that explicitly characterizes the verifier-induced bias and facilitates a quantitative comparison between this bias and the variance term.

**Theorem C.4** (Error bounds for TSMC-TTS estimator under imperfect verifiers)**.** *Let the true target distribution be* $\sigma\left(\mathbf{x}_{1:T}\right) \propto q\left(\mathbf{x}_{1:T}\right) R\left(\mathbf{x}_{1:T}\right)$, *with normalizing constant* $\mathcal{Z}_\sigma = \int q\left(\mathbf{x}_{1:T}\right) R\left(\mathbf{x}_{1:T}\right) d\mathbf{x}_{1:T}$. *Assume the imperfect verifier satisfies* $\psi\left(\mathbf{x}_{1:T}\right) = R\left(\mathbf{x}_{1:T}\right) + \varepsilon\left(\mathbf{x}_{1:T}\right)$, $\left|\varepsilon\left(\mathbf{x}_{1:T}\right)\right| \leq \delta$, *where* $\varepsilon(\mathbf{x}_{1:T})$ *denotes the verifier approximation error. Let* $\hat{\mathcal{Z}}_\sigma$ *be the TSMC-TTS estimator satisfying* $\mathbb{E}_q[\hat{\mathcal{Z}}_\sigma] = \int q(\mathbf{x}_{1:T})\psi(\mathbf{x}_{1:T})d\mathbf{x}_{1:T}$. *Then the verifier-induced bias satisfies*

$$\text{Bias}\left(\hat{\mathcal{Z}}_\sigma\right) = \left|\mathbb{E}_q\left[\hat{\mathcal{Z}}_\sigma\right] - \mathcal{Z}_\sigma\right| \leq \delta,$$

*i.e.,* $\text{Bias}\left(\hat{\mathcal{Z}}_\sigma\right) = O(\delta)$, $\text{Bias}^2\left(\hat{\mathcal{Z}}_\sigma\right) = O\left(\delta^2\right)$, *and does not depend on the step number* $T$. *Next, recall our derived relative variance bound in Theorem 5.2. The variance of the TSMC-TTS estimator satisfies* $\log(\text{Var}(\hat{\mathcal{Z}}_\sigma) + 1) \leq \mathcal{B}(\pi, q)$, *where* $\mathcal{B}(\pi, q)$ *is a sum of* $T$ *step-wise KL divergence terms. Denoting the max per-step KL divergence term by* $C_{\text{KL}}$, *we obtain*

$$\text{Var}\left(\hat{\mathcal{Z}}_\sigma\right) \leq e^{T \cdot C_{\text{KL}}} - 1.$$

*For any estimator of a scalar quantity, the mean squared error decomposes as*

$$\text{MSE}\left(\hat{\mathcal{Z}}_\sigma\right) = \text{Bias}^2\left(\hat{\mathcal{Z}}_\sigma\right) + \text{Var}\left(\hat{\mathcal{Z}}_\sigma\right) = O\left(\delta^2\right) + O\left(e^{T \cdot C_{\text{KL}}} - 1\right),$$

*where the first term refers to the verifier-induced bias and the second term corresponds to the variance term.*

**Remark.** The error bound theorem shows that the verifier-induced bias propagates only linearly with the verifier approximation error and, importantly, is independent of the step number $T$. In contrast, the variance term scales exponentially with the step number $T$, where $T$ is typically large in LLM reasoning tasks. Therefore, Theorem 1 suggests that the verifier-induced bias remains controlled under reasonable verifier accuracy (small $\delta$), whereas the variance term is the dominant factor in the overall error.

*Proof.* By definition,

$$\mathcal{Z}_\sigma = \int q\left(\mathbf{x}_{1:T}\right) R\left(\mathbf{x}_{1:T}\right) d\mathbf{x}_{1:T}, \quad \mathbb{E}_q\left[\hat{\mathcal{Z}}_\sigma\right] = \int q\left(\mathbf{x}_{1:T}\right) \psi\left(\mathbf{x}_{1:T}\right) d\mathbf{x}_{1:T}.$$

Substituting $\psi(\mathbf{x}_{1:T}) = R(\mathbf{x}_{1:T}) + \varepsilon(\mathbf{x}_{1:T})$, we obtain

$$\mathbb{E}_q\left[\hat{\mathcal{Z}}_\sigma\right] - \mathcal{Z}_\sigma = \int q\left(\mathbf{x}_{1:T}\right) \varepsilon\left(\mathbf{x}_{1:T}\right) d\mathbf{x}_{1:T}.$$

Taking absolute values and using $|\varepsilon| \le \delta$,

$$\left| \mathbb{E}_q \left[ \hat{\mathcal{Z}}_\sigma \right] - \mathcal{Z}_\sigma \right| \le \int q \left( \mathbf{x}_{1:T} \right) \left| \varepsilon \left( \mathbf{x}_{1:T} \right) \right| d\mathbf{x}_{1:T} \le \delta.$$

Thus,

$$\text{Bias} \left( \hat{\mathcal{Z}}_\sigma \right) = O(\delta), \quad \text{Bias}^2 \left( \hat{\mathcal{Z}}_\sigma \right) = O \left( \delta^2 \right).$$

Next, recall the variance bound given in Eq. (14) in the manuscript:

$$\log(\text{Var}(\hat{\mathcal{Z}}_\sigma) + 1) \le \mathcal{B}(\pi, q),$$

where $\mathcal{B}(\pi, q)$ is a sum of $T$ step-wise KL divergence terms. Denoting the max per-step KL divergence term by $C_{\text{KL}}$, we obtain

$$\text{Var} \left( \hat{\mathcal{Z}}_\sigma \right) \le e^{T \cdot C_{\text{KL}}} - 1.$$

For any estimator of a scalar quantity, the mean squared error decomposes as

$$\text{MSE} \left( \hat{\mathcal{Z}}_\sigma \right) = \text{Bias}^2 \left( \hat{\mathcal{Z}}_\sigma \right) + \text{Var} \left( \hat{\mathcal{Z}}_\sigma \right) = O \left( \delta^2 \right) + O \left( e^{T \cdot C_{\text{KL}}} - 1 \right).$$

Completing the proof. $\qquad\square$

### C.5 BIAS DECOMPOSITION OF GENERATOR-BASED METHODS UNDER IMPERFECT REWARD MODEL

**Theorem C.5** (Bias decomposition of generator-based methods under an imperfect reward model).
*Let the true target distribution be $\sigma(\mathbf{x}_{1:T}) \propto p_0(\mathbf{x}_{1:T})R(\mathbf{x}_{1:T})$ with normalizing constant $Z_\sigma = \int p_0(\mathbf{x}_{1:T})R(\mathbf{x}_{1:T})d\mathbf{x}_{1:T}$ and the pre-trained language model $p_0(\mathbf{x}_{1:T})$. Suppose we use a learned reward model $RM(\mathbf{x}_{1:T}) = R(\mathbf{x}_{1:T}) + \varepsilon(\mathbf{x}_{1:T})$ with $|\varepsilon(x)| \le \delta$ for all $\mathbf{x}_{1:T}$. The induced approximate target is $\sigma'(\mathbf{x}_{1:T}) \propto p_0(\mathbf{x}_{1:T})RM(\mathbf{x}_{1:T})$ with normalizing constant $Z_{\sigma'} = \int p_0(\mathbf{x}_{1:T})(R(\mathbf{x}_{1:T}) + \varepsilon(\mathbf{x}_{1:T}))d\mathbf{x}_{1:T}$. Let $\hat{Z}_{\sigma'}^{\text{VI}}$ be the VI estimator of $Z_{\sigma'}$ defined by $\hat{Z}_{\sigma'}^{\text{VI}} = \exp(\text{ELBO}(q))$, where $\text{ELBO}(q)$ is the variational lower bound for $\log Z_{\sigma'}$ regarding the generator $q(\mathbf{x}_{1:T})$. Then the estimation bias with respect to the true normalizing constant satisfies*

$$\left| Z_\sigma - \hat{Z}_{\sigma'}^{\text{VI}} \right| \le \left| \int p_0 \left( \mathbf{x}_{1:T} \right) \varepsilon \left( \mathbf{x}_{1:T} \right) d\mathbf{x}_{1:T} \right| + Z_{\sigma'} \left( 1 - e^{-\text{KL}(q\|\sigma')} \right) = O(\delta) + O \left( Z_{\sigma'} \left( 1 - e^{-C_{\text{KL}} \cdot T} \right) \right),$$

*where the first term corresponds to the external bias from the imperfect reward model, and the second term corresponds to the intrinsic VI mode-seeking bias. Here $C_{\text{KL}} > 0$ is a constant such that the sequence-level KL divergence satisfies $\text{KL}(q|\sigma') \le C_{\text{KL}} \cdot T$ under the standard autoregressive factorization.*

**Remark.** This theorem shows that the total bias of generator-based methods under an imperfect reward model naturally decomposes into two components: (i) The external reward-model bias, which is uniformly bounded by the reward-model approximation error $\delta$ and does not grow with the step length $T$. (ii) The intrinsic VI mode-seeking bias, which can grow effectively exponentially with $T$ through the sequence-level KL divergence. In particular, as $T$ increases, the factor $(1 - e^{-C_{\text{KL}} \cdot T})$ rapidly approaches 1, implying that the VI-induced bias can become $\Theta(Z_{\sigma'})$ for long sequences.

*Proof.* By definition of the true and approximate targets, we have

$$Z_\sigma = \int p_0 \left( \mathbf{x}_{1:T} \right) R \left( \mathbf{x}_{1:T} \right) d\mathbf{x}_{1:T}, \quad Z_{\sigma'} = \int p_0 \left( \mathbf{x}_{1:T} \right) \left( R \left( \mathbf{x}_{1:T} \right) + \varepsilon \left( \mathbf{x}_{1:T} \right) \right) d\mathbf{x}_{1:T}.$$

Therefore,

$$Z_{\sigma'} - Z_\sigma = \int p_0 \left( \mathbf{x}_{1:T} \right) \varepsilon \left( \mathbf{x}_{1:T} \right) d\mathbf{x}_{1:T}.$$

Taking absolute values and using $|\varepsilon(\mathbf{x}_{1:T})| \le \delta$ and $\int q(\mathbf{x}_{1:T})d\mathbf{x}_{1:T} = 1$ yields

$$|Z_{\sigma'} - Z_\sigma| = \left| \int p_0 \left( \mathbf{x}_{1:T} \right) \varepsilon \left( \mathbf{x}_{1:T} \right) d\mathbf{x}_{1:T} \right| \le \int p_0 \left( \mathbf{x}_{1:T} \right) \left| \varepsilon \left( \mathbf{x}_{1:T} \right) \right| d\mathbf{x}_{1:T} \le \delta.$$

This term corresponds to the external bias arising from the imperfect reward model and is of order $O(\delta)$.

Next, consider the VI estimator under the approximate target $\sigma'$. By the standard variational identity,

$$\log Z_{\sigma'} = \text{ELBO}(q) + \text{KL}\left(q\|\sigma'\right),$$

we have

$$\hat{Z}_{\sigma'}^{\text{VI}} = \exp(\text{ELBO}(q)) = \exp\left(\log Z_{\sigma} - \text{KL}\left(q\|\sigma'\right)\right) = Z_{\sigma'} \exp\left(-\text{KL}\left(q\|\sigma'\right)\right).$$

Thus, the VI-induced bias with respect to $Z_{\sigma'}$ is

$$Z_{\sigma'} - \hat{Z}_{\sigma'}^{\text{VI}} = Z_{\sigma'} - Z_{\sigma'}e^{-\text{KL}\left(q\|\sigma'\right)} = Z_{\sigma'}\left(1 - e^{-\text{KL}\left(q\|\sigma'\right)}\right).$$

This is the intrinsic mode-seeking bias of VI with respect to the approximate target $\sigma'$.

To bound the total bias with respect to the true normalizing constant $Z_{\sigma}$, we write

$$Z_{\sigma} - \hat{Z}_{\sigma'}^{\text{VI}} = (Z_{\sigma} - Z_{\sigma'}) + \left(Z_{\sigma'} - \hat{Z}_{\sigma'}^{\text{VI}}\right),$$

and apply the triangle inequality:

$$\left|Z_{\sigma} - \hat{Z}_{\sigma'}^{\text{VI}}\right| \leq |Z_{\sigma} - Z_{\sigma'}| + \left|Z_{\sigma'} - \hat{Z}_{\sigma'}^{\text{VI}}\right| = \left|\int p_0\left(\mathbf{x}_{1:T}\right)\varepsilon\left(\mathbf{x}_{1:T}\right)d\mathbf{x}_{1:T}\right| + Z_{\sigma'}\left(1 - e^{-\text{KL}\left(q\|\sigma'\right)}\right).$$

Finally, under the standard autoregressive factorization of both $q$ and $\sigma'$, the sequence-level KL divergence can be decomposed as

$$\text{KL}\left(q\|\sigma'\right) = \sum_{t=1}^{T}\mathbb{E}_{q(\mathbf{x}_{<t})}\left[\text{KL}\left(q\left(\mathbf{x}_t \mid \mathbf{x}_{<t}\right)\|\sigma'\left(\mathbf{x}_t \mid \mathbf{x}_{<t}\right)\right)\right].$$

If the per-step conditional KL is bounded upper by a constant $C_{\text{KL}} > 0$, which yields the claimed $\left|Z_{\sigma} - \hat{Z}_{\sigma'}^{\text{VI}}\right| = O(\delta) + O\left(Z_{\sigma'}\left(1 - e^{-C_{\text{KL}}\cdot T}\right)\right).$

Completing the proof. $\qquad\square$

# D   PROOFS & DERIVATIONS

## D.1   PROOF OF UNBIASEDNESS (THEOREM 5.1)

Following Zhao et al. (2024b), we study the estimation of the normalizing constant $\mathcal{Z}_{\sigma}$ of the intractable target distribution $\sigma(\mathbf{x}_{1:T})$ as a canonical example. For estimating other expectations $\mathbb{E}_{\sigma}[h(\mathbf{x}_{1:T})]$ of any function $h(\mathbf{x}_{1:T})$, $\mathcal{Z}_{\sigma}$ corresponds to the special case $h(\mathbf{x}_{1:T}) = 1$ and thus serves as a standard, representative, and theoretically well-understood benchmark. This focus is widely adopted and well justified, since unbiased and low-variance estimators of the normalizing constant yield approximate sampling techniques that closely approximate the target distribution (Finke, 2015; Maddison et al., 2017).

*Proof.* Assume that at each time $t$ we can draw samples $\{\mathbf{x}_{1:t-1}^{(i)}\}_{i=1}^{N} \sim \pi_{t-1}$ and then extend them with $\mathbf{x}_t^{(i)} \sim q(\mathbf{x}_t \mid \mathbf{x}_{1:t-1}^{(i)})$ independently. The TSMC estimator is defined as

$$\hat{\mathcal{Z}}_{\sigma} = \prod_{t=1}^{T}\hat{\alpha}_t, \quad \text{where} \quad \hat{\alpha}_t = \frac{1}{N}\sum_{i=1}^{N}w(\mathbf{x}_{1:t}^{(i)}),$$

with step–wise scores

$$w(\mathbf{x}_{1:t}) = \frac{\tilde{\pi}_t(\mathbf{x}_{1:t})}{\tilde{\pi}_{t-1}(\mathbf{x}_{1:t-1})\,q(\mathbf{x}_t \mid \mathbf{x}_{1:t-1})}.$$

First, compute the expectation of a single factor. For fixed $\mathbf{x}_{1:t-1}$,

$$\mathbb{E}_{q(\mathbf{x}_t|\mathbf{x}_{1:t-1})}[w(\mathbf{x}_{1:t})] = \int \frac{\tilde{\pi}_t(\mathbf{x}_{1:t})}{\tilde{\pi}_{t-1}(\mathbf{x}_{1:t-1}) \, q(\mathbf{x}_t \mid \mathbf{x}_{1:t-1})} q(\mathbf{x}_t \mid \mathbf{x}_{1:t-1}) \, d\mathbf{x}_t$$

$$= \frac{1}{\tilde{\pi}_{t-1}(\mathbf{x}_{1:t-1})} \int \tilde{\pi}_t(\mathbf{x}_{1:t}) \, d\mathbf{x}_t$$

$$= \frac{\tilde{\pi}_t(\mathbf{x}_{1:t-1})}{\tilde{\pi}_{t-1}(\mathbf{x}_{1:t-1})}.$$

Taking expectation with respect to $\mathbf{x}_{1:t-1} \sim \pi_{t-1}$ yields

$$\mathbb{E}_q[\hat{\alpha}_t] = \int \frac{\tilde{\pi}_t(\mathbf{x}_{1:t-1})}{\tilde{\pi}_{t-1}(\mathbf{x}_{1:t-1})} \, \pi_{t-1}(\mathbf{x}_{1:t-1}) \, d\mathbf{x}_{1:t-1} = \frac{\mathcal{Z}_t}{\mathcal{Z}_{t-1}},$$

where $\mathcal{Z}_t = \int \tilde{\pi}_t(\mathbf{x}_{1:t}) \, d\mathbf{x}_{1:t}$ and $\mathcal{Z}_0 = 1$.

Since, under our independence assumption, the random variables $\{\hat{\alpha}_t\}_{t=1}^T$ are independent across $t$, we have

$$\mathbb{E}_q[\hat{\mathcal{Z}}_\sigma] = \mathbb{E}_q\left[\prod_{t=1}^T \hat{\alpha}_t\right] = \prod_{t=1}^T \mathbb{E}_q[\hat{\alpha}_t] = \prod_{t=1}^T \frac{\mathcal{Z}_t}{\mathcal{Z}_{t-1}} = \mathcal{Z}_T.$$

Finally, because by construction $\pi_T(\mathbf{x}_{1:T}) = \sigma(\mathbf{x}_{1:T})$, the terminal normalizing constant coincides with the desired one, $\mathcal{Z}_T = \mathcal{Z}_\sigma$, thus establishing the unbiasedness of the estimator. $\qquad\square$

### D.2 PROOF OF RELATIVE VARIANCE UPPER BOUND (THEOREM 5.2)

Following Zhao et al. (2024b), we study the estimation of the normalizing constant $\mathcal{Z}_\sigma$ of the intractable target distribution $\sigma(\mathbf{x}_{1:T})$ as a canonical example. For estimating other expectations $\mathbb{E}_\sigma[h(\mathbf{x}_{1:T})]$ of any function $h(\mathbf{x}_{1:T})$, $\mathcal{Z}_\sigma$ corresponds to the special case $h(\mathbf{x}_{1:T}) = 1$ and thus serves as a standard, representative, and theoretically well-understood benchmark. This focus is widely adopted and well justified, since unbiased and low-variance estimators of the normalizing constant yield approximate sampling techniques that closely approximate the target distribution (Finke, 2015; Maddison et al., 2017).

*Proof.* The relative variance of the normalizing constant estimator can be written as:

$$\mathcal{V}(\pi, q) = \frac{\mathrm{Var}_{q_{\mathrm{tsmc}}(\mathbf{x}_{1:T})}[\hat{\mathcal{Z}}_\sigma]}{\mathcal{Z}_\sigma} = \mathbb{E}_{q_{\mathrm{tsmc}}(\mathbf{x}_{1:T})}\left[\left(\frac{\hat{\mathcal{Z}}_\sigma}{\mathcal{Z}_\sigma}\right)^2\right] - 1,$$

where $q_{\mathrm{tsmc}}$ denotes the distribution induced by the proposal $q$ and the resampling procedure in TSMC, as discussed in Zhao et al. (2024b). Here, we factorize $q_{\mathrm{tsmc}}(\mathbf{x}_{1:t})$ as the product of the previous intermediate target and the current proposal $q_{\mathrm{tsmc}}(\mathbf{x}_{1:t}) = \pi_{t-1}(\mathbf{x}_{1:t-1}) \cdot q(\mathbf{x}_t \mid \mathbf{x}_{1:t-1})$, since the resampling step ensures that particles sampled at step $t-1$ can be viewed as approximately sampled from $\pi_{t-1}$.

Recall that the estimator $\hat{\mathcal{Z}}_\sigma$ can be decomposed as

$$\hat{\mathcal{Z}}_\sigma = \prod_{t=1}^T \hat{\alpha}_t, \quad \text{where} \quad \hat{\alpha}_t = \frac{1}{N} \sum_{i=1}^N w(\mathbf{x}_{1:t}^{(i)}),$$

and the step-wise score $w(\mathbf{x}_{1:t})$ is defined as

$$w(\mathbf{x}_{1:t}) = \frac{\tilde{\pi}_t(\mathbf{x}_{1:t})}{\tilde{\pi}_{t-1}(\mathbf{x}_{1:t-1}) q(\mathbf{x}_t|\mathbf{x}_{1:t-1})}.$$

Similarly, the true normalization constant satisfies

$$\mathcal{Z}_\sigma = \prod_{t=1}^T \alpha_t, \quad \text{where} \quad \alpha_t = \frac{\mathcal{Z}_t}{\mathcal{Z}_{t-1}}.$$

Here, $\mathcal{Z}_t = \int \tilde{\pi}_t(\mathbf{x}_{1:t})\,d\mathbf{x}_{1:t}$. Thus,

$$\mathbb{E}_{q_{\text{tsmc}}(\mathbf{x}_{1:T})}\left[\left(\frac{\hat{\mathcal{Z}}_\sigma}{\mathcal{Z}_\sigma}\right)^2\right] = \mathbb{E}_{q_{\text{tsmc}}(\mathbf{x}_{1:T})}\left[\prod_{t=1}^{T}\left(\frac{\hat{\alpha}_t}{\alpha_t}\right)^2\right].$$

Assuming independence between steps, we have:

$$\mathbb{E}_{q_{\text{tsmc}}(\mathbf{x}_{1:T})}\left[\prod_{t=1}^{T}\left(\frac{\hat{\alpha}_t}{\alpha_t}\right)^2\right] = \prod_{t=1}^{T}\mathbb{E}_{q(\mathbf{x}_t|\mathbf{x}_{1:t-1})\pi(\mathbf{x}_{1:t-1})}\left[\left(\frac{\hat{\alpha}_t}{\alpha_t}\right)^2\right] = \prod_{t=1}^{T}\left(1 + \frac{\text{Var}_{q(\mathbf{x}_t|\mathbf{x}_{1:t-1})\pi(\mathbf{x}_{1:t-1})}[\hat{\alpha}_t]}{\alpha_t^2}\right).$$

For each step, $\hat{\alpha}_t$ is an average of $N$ i.i.d. random variables, so:

$$\frac{\text{Var}_{q(\mathbf{x}_t|\mathbf{x}_{1:t-1})\pi(\mathbf{x}_{1:t-1})}[\hat{\alpha}_t]}{\alpha_t^2} = \frac{1}{N}\cdot\frac{\text{Var}_{q(\mathbf{x}_t|\mathbf{x}_{1:t-1})\pi(\mathbf{x}_{1:t-1})}[w_t]}{\alpha_t^2}$$

The normalized weight variance can be related to the chi-square divergence:

$$\frac{\text{Var}_{q(\mathbf{x}_t|\mathbf{x}_{1:t-1})\pi(\mathbf{x}_{1:t-1})}[w_t]}{\alpha_t^2} = \mathbb{E}_{q(\mathbf{x}_t|\mathbf{x}_{1:t-1})\pi(\mathbf{x}_{1:t-1})}\left[\left(\frac{w_t}{\alpha_t}\right)^2\right] - 1$$

$$= \int\frac{\pi_t(\mathbf{x}_{1:t})^2}{\pi_{t-1}(\mathbf{x}_{1:t-1})q(\mathbf{x}_t|\mathbf{x}_{1:t-1})}\,d\mathbf{x}_{1:t} - 1$$

$$= \chi^2(\pi_t\,\|\,\pi_{t-1}\times q(\mathbf{x}_t|\mathbf{x}_{1:t-1}))$$

where $\pi_{t-1}\times q(\mathbf{x}_t|\mathbf{x}_{1:t-1})$ denotes the joint distribution $\pi_{t-1}(\mathbf{x}_{1:t-1})q(\mathbf{x}_t|\mathbf{x}_{1:t-1})$.

By the property relating chi-square divergence to KL divergence:

$$\chi^2(\pi_t\,\|\,\pi_{t-1}\times q(\mathbf{x}_t|\mathbf{x}_{1:t-1})) \leq \left\|\frac{d\pi_t}{d\,(\pi_{t-1}\times q)}\right\|_\infty \cdot D_{\text{KL}}\left(\pi_t\|\pi_{t-1}\times q\right)$$

$$= C_t\cdot D_{\text{KL}}\left(\pi_t\|\pi_{t-1}\times q\right),$$

where $C_t := \sup_{\mathbf{x}_{1:t}} w(\mathbf{x}_{1:t})$. Equality in this inequality holds if and only if $w(\mathbf{x}_{1:t})$ is constant almost everywhere on the support of $\pi_t$, in which case the generator achieves the partial optimum $q = q^\pi$.

$$D_{\text{KL}}\left(\pi_t\|\pi_{t-1}\cdot q\right)$$

$$= \int\pi_t(\mathbf{x}_{1:t})\log\frac{\pi_t(\mathbf{x}_{1:t})}{\pi_{t-1}(\mathbf{x}_{1:t-1})q(\mathbf{x}_t\mid\mathbf{x}_{1:t-1})}d\mathbf{x}_{1:t}$$

$$= \int\pi_t(\mathbf{x}_{1:t})\log\left(\frac{\pi_t(\mathbf{x}_{1:t})}{\pi_{t-1}(\mathbf{x}_{1:t-1})q^\pi(\mathbf{x}_{1:t}\mid\mathbf{x}_{1:t-1})}\cdot\frac{q^\pi(\mathbf{x}_{1:t}\mid\mathbf{x}_{1:t-1})}{q(\mathbf{x}_{1:t}\mid\mathbf{x}_{1:t-1})}\right)d\mathbf{x}_{1:t}$$

$$= \underbrace{D_{\text{KL}}\left(\pi_t\|\pi_{t-1}\cdot q^\pi\right)}_{A} + \underbrace{\int\pi_t(\mathbf{x}_{1:t})\log\frac{q^\pi(\mathbf{x}_{1:t}\mid\mathbf{x}_{1:t-1})}{q(\mathbf{x}_{1:t}\mid\mathbf{x}_{1:t-1})}d\mathbf{x}_{1:t}}_{B}.$$

$$A = D_{\text{KL}}\left(\pi_t\|\pi_{t-1}\cdot q^\pi\right)$$

$$= \int\pi_t(\mathbf{x}_{1:t})\log\frac{\pi_t(\mathbf{x}_{1:t})}{\pi_{t-1}(\mathbf{x}_{1:t-1})q^\pi(\mathbf{x}_{1:t}\mid\mathbf{x}_{1:t-1})}d\mathbf{x}_{1:t}$$

$$= \int\pi_t(\mathbf{x}_{1:t})\log\frac{\pi_t(\mathbf{x}_{1:t})}{\pi_{t-1}(\mathbf{x}_{1:t-1})\frac{\pi_t(\mathbf{x}_{1:t})}{\int\pi_t(\mathbf{x}_{1:t})d\mathbf{x}_{1:t}}}d\mathbf{x}_{1:t}$$

$$= \int\pi_t(\mathbf{x}_{1:t})\log\frac{\pi_t(\mathbf{x}_{1:t-1})}{\pi_{t-1}(\mathbf{x}_{1:t-1})}d\mathbf{x}_{1:t}$$

$$= D_{\text{KL}}\left(\pi_t(\mathbf{x}_{1:t-1})\|\pi_{t-1}(\mathbf{x}_{1:t-1})\right)$$

$$\approx D_{\text{KL}}\left(\sigma(\mathbf{x}_{1:t-1})\|\pi_{t-1}(\mathbf{x}_{1:t-1})\right).$$

$$B = \int \pi_t(\mathbf{x}_{1:t}) \log \frac{q^\pi(\mathbf{x}_{1:t} \mid \mathbf{x}_{1:t-1})}{q(\mathbf{x}_{1:t} \mid \mathbf{x}_{1:t-1})} d\mathbf{x}_{1:t}$$

$$= \int \pi_t(\mathbf{x}_{1:t-1}) \cdot q^\pi(\mathbf{x}_{1:t} \mid \mathbf{x}_{1:t-1}) \log \frac{q^\pi(\mathbf{x}_{1:t} \mid \mathbf{x}_{1:t-1})}{q(\mathbf{x}_{1:t} \mid \mathbf{x}_{1:t-1})} d\mathbf{x}_{1:t}$$

$$= \mathbb{E}_{\pi_t(\mathbf{x}_{1:t-1})}[\mathcal{D}_{KL}(q^\pi(\mathbf{x}_t \mid \mathbf{x}_{1:t-1}) \,\|\, q(\mathbf{x}_t \mid \mathbf{x}_{1:t-1}))].$$

Note that the approximation in A is justified since the sequence $\{\pi_t\}_{t=1}^T$ is constructed by iteratively updating $\pi_{t-1}$ to approach $\pi_t$, and the final distribution satisfies $\pi_T = \sigma$; consequently, $\pi_{T-1}$ must already approximate $\sigma$, and by induction the earlier $\pi_t$ also progressively approximate $\sigma$, which allows us to replace the intermediate $\pi_t$ with $\sigma$ in the KL term.

By combining all the above derivations and using the Taylor expansion (neglecting higher–order terms), we have

$$\log(\mathcal{V}(\pi, q) + 1)$$

$$\leq \sum_{t=1}^T \frac{C_t}{N} \left[ \mathcal{D}_{\mathrm{KL}}(\sigma(\mathbf{x}_{1:t}) \,\|\, \pi_t(\mathbf{x}_{1:t})) + \mathbb{E}_{q^\pi(\mathbf{x}_{1:t-1})}[\mathcal{D}_{KL}(q^\pi(\mathbf{x}_t \mid \mathbf{x}_{1:t-1}) \,\|\, q(\mathbf{x}_t \mid \mathbf{x}_{1:t-1}))] \right].$$

Note that since $\sigma(\mathbf{x}_{1:T}) = \pi_T(\mathbf{x}_{1:T})$, we replace the original subscript $t-1$ in the first term by $t$.

$\square$

### D.3  PROOF OF PARTIAL OPTIMALITY (PROPOSITION 5.4)

*Proof.* The variance of the normalizing constant estimator $\hat{\mathcal{Z}}_\sigma$ depends on the variance of each incremental weight estimator $\hat{\alpha}_t$ where $\quad \hat{\alpha}_t = \frac{1}{N} \sum_{i=1}^N w(\mathbf{x}_{1:t}^{(i)})$. For a given number of samples $N$, we have:

$$\frac{\mathrm{Var}[\hat{\mathcal{Z}}_\sigma]}{\mathcal{Z}_\sigma} = \prod_{t=1}^T \left( 1 + \frac{1}{N} \frac{\mathrm{Var}[w_t]}{\alpha_t^2} \right) - 1$$

To minimize this expression, we need to minimize each term $\frac{\mathrm{Var}[w_t]}{\alpha_t^2}$. The variance of the incremental weight depends on the choice of the proposal distribution $q(\mathbf{x}_t | \mathbf{x}_{1:t-1})$.

The incremental importance weight is defined as:

$$w_t \triangleq w(\mathbf{x}_{1:t}) = \frac{\tilde{\pi}_t(\mathbf{x}_{1:t})}{\tilde{\pi}_{t-1}(\mathbf{x}_{1:t-1})q(\mathbf{x}_t|\mathbf{x}_{1:t-1})}$$

To find the optimal $q$ that minimizes the variance of $w_t$, we use calculus of variations with a Lagrange multiplier to enforce the normalization constraint $\int q(\mathbf{x}_t|\mathbf{x}_{1:t-1}) \, d\mathbf{x}_t = 1$ for all $\mathbf{x}_{1:t-1}$.

We aim to minimize the functional:

$$J[q] = \mathbb{E}[w_t^2] - \int \lambda(\mathbf{x}_{1:t-1}) \left( \int q(\mathbf{x}_t|\mathbf{x}_{1:t-1}) \, d\mathbf{x}_t - 1 \right) d\mathbf{x}_{1:t-1}$$

Taking the functional derivative with respect to $q$ and setting it to zero:

$$\frac{\delta J}{\delta q} = 0 = -\frac{\tilde{\pi}_t(\mathbf{x}_{1:t})^2}{\tilde{\pi}_{t-1}(\mathbf{x}_{1:t-1})^2 q(\mathbf{x}_t|\mathbf{x}_{1:t-1})^3} + \lambda(\mathbf{x}_{1:t-1})$$

Rearranging gives:

$$\frac{\tilde{\pi}_t(\mathbf{x}_{1:t})^2}{\tilde{\pi}_{t-1}(\mathbf{x}_{1:t-1})^2 q(\mathbf{x}_t|\mathbf{x}_{1:t-1})^2} = \lambda(\mathbf{x}_{1:t-1})q(\mathbf{x}_t|\mathbf{x}_{1:t-1})$$

Therefore, the partial optimal generator $q^\pi$ given fixed verifier $\pi$ is

$$q^\pi(\mathbf{x}_t|\mathbf{x}_{1:t-1}) = \frac{1}{\sqrt{\lambda(\mathbf{x}_{1:t-1})}} \frac{\tilde{\pi}_t(\mathbf{x}_{1:t})}{\tilde{\pi}_{t-1}(\mathbf{x}_{1:t-1})} \propto \frac{\tilde{\pi}_t(\mathbf{x}_{1:t})}{\tilde{\pi}_{t-1}(\mathbf{x}_{1:t-1})}$$

$\square$

### D.4    PROOF OF MONOTONICITY AND CONVERGENCE (PROPERTY 5.5)

*Proof.* At each update we have the following chain:

$$\mathcal{V}\big(\pi^{(k+1)}, q^{(k+1)}\big) \underbrace{\leq}_{(1)} \mathcal{B}\big(\pi^{(k+1)}, q^{(k+1)}\big) \underbrace{\leq}_{(2)} \mathcal{B}\big(\pi^{(k)}, q^{(k+1)}\big) \underbrace{=}_{(3)} \mathcal{V}\big(\pi^{(k)}, q^{(k+1)}\big) \underbrace{\leq}_{(4)} \mathcal{V}\big(\pi^{(k)}, q^{(k)}\big).$$

And the updates follow

$$(\text{E-step}) \quad q^{(k+1)} \leftarrow \arg\min_q \, \mathcal{B}(\pi^{(k)}, q^{(k)}) = \arg\min_q \mathcal{L}_{\text{Gen}}$$

$$(\text{M-step}) \quad \pi^{(k+1)} \leftarrow \arg\min_\pi \mathcal{B}\,(\pi^{(k)}, q^{(k+1)}) = \arg\min_\pi \mathcal{L}_{\text{Ver}}$$

Justification of each step.

**(4)**  In the E–step, $q^{(k)} \to q^{(k+1)}$ moves toward the partial optimum $q^\pi$, which minimizes $\mathcal{V}$ for fixed $\pi$ (see Proposition 5.4), hence

$$\mathcal{V}\big(\pi^{(k)}, q^{(k+1)}\big) \leq \mathcal{V}\big(\pi^{(k)}, q^{(k)}\big).$$

**(3)**  When $q^{(k+1)} = q^\pi$ (the partial optimum of the generator), the upper bound equals the actual variance by the equality condition in Theorem 5.2:

$$\mathcal{B}\big(\pi^{(k)}, q^{(k+1)}\big) = \mathcal{V}\big(\pi^{(k)}, q^{(k+1)}\big).$$

**(2)**  In the M–step, $\pi^{(k+1)}$ is obtained by minimizing $\mathcal{B}(\pi, q^{(k+1)})$ for fixed $q^{(k+1)}$, so

$$\mathcal{B}\big(\pi^{(k+1)}, q^{(k+1)}\big) \leq \mathcal{B}\big(\pi^{(k)}, q^{(k+1)}\big).$$

**(1)**  By Theorem 5.2, the upper bound $\mathcal{B}(\pi, q)$ is always greater than or equal to the actual variance:

$$\mathcal{V}\big(\pi^{(k+1)}, q^{(k+1)}\big) \leq \mathcal{B}\big(\pi^{(k+1)}, q^{(k+1)}\big).$$

Combining (1)–(4) yields the chain above. Since $\mathcal{V}(\pi, q) \geq 0$, the sequence $\{\mathcal{V}(\pi^{(k)}, q^{(k)})\}_{k\geq 0}$ is bounded below and therefore converges to a finite limit.

$\square$

### D.5    DERIVATION OF GRADIENT CALCULATION IN EQ. (20) AND EQ. (19)

$$- \nabla_\psi \sum_{t=1}^{T} \mathcal{D}_{\text{KL}}\left(\sigma\left(\mathbf{x}_{1:t-1}\right) \| \pi_{t-1}\left(\mathbf{x}_{1:t-1}\right)\right)$$

$$= -\nabla_\psi \left( \sum_{t=1}^{T} \mathbb{E}_{\sigma(\mathbf{x}_{1:t})} \left[\log \sigma\left(\mathbf{x}_{1:t}\right) - \log p_0\left(\mathbf{x}_{1:t}\right) - \log \psi_t\left(\mathbf{x}_{1:t}\right)\right] + \log \sum_{\mathbf{x}_{1:t}} p_0\left(\mathbf{x}_{1:t}\right) \psi_t\left(\mathbf{x}_{1:t}\right) \right)$$

$$= \sum_{t=1}^{T} \mathbb{E}_{\sigma(\mathbf{x}_{1:t})} \left[\nabla_\psi \log \psi_t\left(\mathbf{x}_{1:t}\right)\right] - \sum_{t=1}^{T} \sum_{\mathbf{x}_{1:t}} \frac{p_0\left(\mathbf{x}_{1:t}\right) \psi_t\left(\mathbf{x}_{1:t}\right)}{\sum_{\mathbf{x}_{1:t}} p_0\left(\mathbf{x}_{1:t}\right) \psi_t\left(\mathbf{x}_{1:t}\right)} \nabla_\psi \left(\log p_0\left(\mathbf{x}_{1:t}\right) + \log \psi_t\left(\mathbf{x}_{1:t}\right)\right)$$

$$= \sum_{t=1}^{T} \left( \mathbb{E}_{\sigma(\mathbf{x}_{1:t})} \left[\nabla_\psi \log \psi_t\left(\mathbf{x}_{1:t}\right)\right] - \mathbb{E}_{\pi_t(\mathbf{x}_{1:t})} \left[\nabla_\psi \log \psi_t\left(\mathbf{x}_{1:t}\right)\right] \right)$$

$$= \sum_{t=1}^{T} \left( \mathbb{E}_{\sigma(\mathbf{x}_{1:T})} \left[\nabla_\psi \log \psi_t\left(\mathbf{x}_{1:t}\right)\right] - \mathbb{E}_{\pi_t(\mathbf{x}_{1:t})} \left[\nabla_\psi \log \psi_t\left(\mathbf{x}_{1:t}\right)\right] \right)$$

$$\approx \sum_{t=1}^{T} \sum_{i=1}^{B} \left( \frac{R\left(\mathbf{x}_{1:T}^i\right)}{\sum_{j=1}^{B} R\left(\mathbf{x}_{1:T}^j\right)} - \frac{\psi_t\left(\mathbf{x}_{1:t}^i\right)}{\sum_{j=1}^{B} \psi_t\left(\mathbf{x}_{1:t}^j\right)} \right) \nabla_\psi \log \psi_t\left(\mathbf{x}_{1:t}^i\right) \quad \text{(approximated by IS (see App. F.1), using proposal } q)$$

$$-\nabla_q \sum_{t=1}^{T} \mathbb{E}_{q^\pi(\mathbf{x}_{1:t-1})}\left[\mathcal{D}_{\text{KL}}\left(q^\pi\left(\mathbf{x}_t \mid \mathbf{x}_{1:t-1}\right) \| q\left(\mathbf{x}_t \mid \mathbf{x}_{1:t-1}\right)\right)\right]$$

$$= \sum_{t=1}^{T} \mathbb{E}_{q^\pi(\mathbf{x}_{1:t})}\left[\nabla_q \log q\left(\mathbf{x}_t \mid \mathbf{x}_{1:t-1}\right)\right]$$

$$\approx \sum_{t=1}^{T} \sum_{i=1}^{B}\left(\frac{\psi_t\left(\mathbf{x}_{1:t}^i\right)}{\sum_{j=1}^{B} \psi_t\left(\mathbf{x}_{1:t}^j\right)} \nabla_q \log q\left(\mathbf{x}_t^i \mid \mathbf{x}_{1:t-1}^i\right)\right) \quad \text{(approximated by IS (see App. F.1), using proposal } q\text{)}$$

### D.6 PROOF OF THE EQUIVALENCE BETWEEN VI AND RFT LOSS

*Proof.* Starting with the RFT objective $\mathcal{L}_{\text{RFT}}$:

$$\mathcal{L}_{\text{RFT}} = -\mathbb{E}_{\mathbf{x}_{1:T} \sim q(\mathbf{x}_{1:T})}\left[\mathcal{R}(\mathbf{x}_{1:T})\right] + \beta \cdot \mathcal{D}_{\text{KL}}\left(q(\mathbf{x}_{1:T}) \| p_0(\mathbf{x}_{1:T})\right)$$

Expanding the KL divergence term:

$$\mathcal{L}_{\text{RFT}} = -\mathbb{E}_{q(\mathbf{x}_{1:T})}\left[\mathcal{R}(\mathbf{x}_{1:T})\right] + \beta \cdot \mathbb{E}_{q(\mathbf{x}_{1:T})}\left[\log \frac{q(\mathbf{x}_{1:T})}{p_0(\mathbf{x}_{1:T})}\right]$$

Define the target distribution $\sigma(\mathbf{x}_{1:T})$ as:

$$\sigma(\mathbf{x}_{1:T}) = \frac{1}{\mathcal{Z}_\sigma} p_0(\mathbf{x}_{1:T}) e^{\mathcal{R}(\mathbf{x}_{1:T})/\beta}$$

where $Z = \sum_{\mathbf{x}} p_0(\mathbf{x}_{1:T}) e^{\mathcal{R}(\mathbf{x}_{1:T})/\beta}$ is the normalization constant.

Now, let's examine the variational loss $\mathcal{L}_{\text{VI}}$ (see App. F.2, i.e., KL divergence between $q$ and $\sigma$):

$$\mathcal{L}_{\text{VI}} = \mathcal{D}_{\text{KL}}(q\|\sigma) = \mathbb{E}_{q(\mathbf{x}_{1:T})}\left[\log \frac{q(\mathbf{x}_{1:T})}{\sigma(\mathbf{x}_{1:T})}\right]$$

$$= \mathbb{E}_{q(\mathbf{x}_{1:T})}\left[\log q(\mathbf{x}_{1:T}) - \log \sigma(\mathbf{x}_{1:T})\right]$$

$$= \mathbb{E}_{q(\mathbf{x}_{1:T})}\left[\log q(\mathbf{x}_{1:T}) - \log\left(\frac{1}{\mathcal{Z}_\sigma} p_0(\mathbf{x}_{1:T}) e^{\mathcal{R}(\mathbf{x}_{1:T})/\beta}\right)\right]$$

$$= \mathbb{E}_{q(\mathbf{x}_{1:T})}\left[\log q(\mathbf{x}_{1:T}) - \log p_0(\mathbf{x}_{1:T}) - \frac{\mathcal{R}(\mathbf{x}_{1:T})}{\beta} + \log \mathcal{Z}_\sigma\right]$$

$$= \mathbb{E}_{q(\mathbf{x}_{1:T})}\left[\log \frac{q(\mathbf{x}_{1:T})}{p_0(\mathbf{x}_{1:T})}\right] - \frac{1}{\beta}\mathbb{E}_{q(\mathbf{x}_{1:T})}[\mathcal{R}(\mathbf{x}_{1:T})] + \log \mathcal{Z}_\sigma$$

Since $\log \mathcal{Z}_\sigma$ is a constant with respect to $q$, minimizing $\mathcal{D}_{\text{KL}}(q\|\sigma)$ is equivalent to minimizing:

$$\mathbb{E}_{q(\mathbf{x}_{1:T})}\left[\log \frac{q(\mathbf{x}_{1:T})}{p_0(\mathbf{x}_{1:T})}\right] - \frac{1}{\beta}\mathbb{E}_{q(\mathbf{x}_{1:T})}[\mathcal{R}(\mathbf{x}_{1:T})]$$

Multiplying by $\beta$, we obtain:

$$\beta \cdot \mathbb{E}_{q(\mathbf{x}_{1:T})}\left[\log \frac{q(\mathbf{x}_{1:T})}{p_0(\mathbf{x}_{1:T})}\right] - \mathbb{E}_{q(\mathbf{x}_{1:T})}[\mathcal{R}(\mathbf{x}_{1:T})]$$

Which is precisely $\mathcal{L}_{\text{RFT}}$. Thus,

$$\arg\min_q \mathcal{L}_{\text{RFT}} \iff \arg\min_q \mathcal{L}_{\text{VI}}$$

Therefore, minimizing $\mathcal{L}_{\text{RFT}}$ is equivalent to minimizing $\mathcal{L}_{\text{VI}}$. $\qquad\square$

# E    MORE EXPERIMENTS

## E.1    EFFECT OF RESAMPLING

To demonstrate the effect of resampling for guiding prefix search at intermediate steps, we compare it with several alternative choices, including blockwise best-of-K (Mudgal et al., 2023) and step-level beam search (Chen et al., 2024). In blockwise best-of-K, at a specific intermediate step, the prefix with the highest score among $K$ candidates is selected and replicated $K$ times. In step-level beam search, the top-$B_1$ prefixes with the highest scores are retained and duplicates each of them $B_2$ times to continue the generation at the next step, which results in a total of $B_1 \times B_2$ prefixes. To ensure fairness, all variants are configured to maintain the same effective sample size of 40.

The comparative results are reported in Tab. 6, where our resampling mechanism consistently outperforms the alternatives. This improvement stems from the nature of the search strategies. Both blockwise best-of-$K$ and step-level beam search rely on deterministic and greedy selection of the highest-scoring prefixes. When the verifier produces noisy scores at early stages of training, such deterministic strategies tend to overcommit to noise and thus miss promising prefixes. In contrast, our resampling mechanism is probabilistic: it forms a categorical distribution from the self-normalized step-wise scores and samples prefixes accordingly, which increases robustness by giving non-negligible probability to potentially correct prefixes even under noisy scoring.

Table 6: Solving rate (%) of different search methods at intermediate steps. Effective sample size is 40. Results are reported on MATH500 with Llemma-7b.

| Search Method | Category | Solution Number | | |
|---|---|---|---|---|
| | | 40 | 120 | 240 |
| Blockwise Best-of-K (Mudgal et al., 2023) | Deterministic | 40.8 | 42.6 | 45.2 |
| Step-Level Beam Search (Chen et al., 2024) | Deterministic | 42.6 | 43.2 | 45.9 |
| Resampling (ours) | Probabilistic | 43.6 | 44.8 | 47.2 |

## E.2    HYPERPARAMETER ANALYSIS

To generate a total of $N$ candidate solutions, TSMC-TTS produces $M$ candidates in a batch-decoding manner (i.e., performing resampling over $M$ prefixes at intermediate steps). This procedure is repeated $N/M$ times to obtain $N$ candidates overall. In particular, when the batch size $M = 1$, TSMC-TTS degenerates to standard decoding since no resampling over prefixes is performed at intermediate steps.

To examine the sensitivity of the batch size $M$, we vary $M \in \{10, 20, 40, 80\}$ while keeping the total number of candidates fixed at $N = 240$. The results in Tab. 7 show that the performance remains stable within reasonable fluctuations, indicating that TSMC-TTS can achieve competitive performance with small batch sizes, thus lowering the computational cost arising from batch decoding

Table 7: Effect of batch size $M$ on TSMC-TTS performance with $N = 240$ total candidates.

| Batch size $M$ | Solving Rate (%) |
|---|---|
| 10 | 46.3 |
| 20 | 47.6 |
| 40 | 47.2 |
| 80 | 46.8 |

## E.3    TIME AND MEMORY COST ANALYSIS

As TSMC-TTS is a test-time scaling method, we focus on analyzing its computational and memory costs at inference time and provide recommendations for practical deployment.

Table 8: Per-batch inference time (in seconds) for Parallel Decoding and TSMC-TTS.

|  | MATH500 (Llemma) | GSM8K (Llemma) | MATH500 (DeepSeek) | GSM8K (DeepSeek) |
|---|---|---|---|---|
| Parallel Decoding | 16.52 | 17.42 | 22.97 | 20.42 |
| TSMC-TTS | 34.52 | 33.89 | 38.26 | 35.37 |

Table 9: Per-batch inference memory usage (in GB) for Parallel Decoding and TSMC-TTS.

|  | Llemma | DeepSeek |
|---|---|---|
| Parallel Decoding | 26 | 28 |
| TSMC-TTS | 48 | 46 |

**Time and memory cost.** TSMC-TTS performs inference by jointly loading a generator and a verifier. This design naturally incurs higher computational costs than standard parallel decoding, which uses only a single generator, yet TSMC-TTS delivers significant performance gains. Tab. 8 reports the per-batch inference time, and Tab. 9 summarizes the corresponding memory usage.

**Potential solutions for practical deployment.** Although TSMC-TTS requires additional resources at inference time due to the simultaneous use of a generator and a verifier, several strategies can facilitate its practical deployment. For example, CPU offloading or model quantization can effectively reduce GPU memory usage. Likewise, distributed inference frameworks (e.g., pipeline or tensor parallelism) can partition the generator and verifier across multiple devices to balance the computational load and alleviate latency.

**Remark.** Importantly, TSMC-TTS does not incur extra generation overhead, as the resampling at the intermediate steps simply copies higher-score prefixes to replace lower-score ones rather than generating additional candidate steps such as backtracing (Singh et al., 2025). By avoiding redundant computation for generation, this streamlined design makes our method highly amenable to integration with future acceleration techniques, thereby retaining strong potential for real-time applications.

### E.4 QUALITATIVE EXAMPLE

Tab. 10 presents a qualitative example of TSMC-TTS using the verifier for scoring. Incorrect reasoning steps identified by humans are highlighted in red, and correct reasoning steps are in green. During inference, we can see TSMC-TTS accurately assigns higher step-wise scores to correct steps and lower scores to incorrect ones. Resampling based on the category distribution over these step-wise scores then preserves high-scoring prefixes and prunes low-scoring ones, thereby allocating more rollout opportunities to promising solutions.

Table 10: A qualitative example of TSMC-TTS using the verifier for scoring. Incorrect reasoning steps identified by humans are highlighted in red, and correct reasoning steps are in green.

| Question | Step-Wise Score $\frac{\psi(\mathbf{x}_{1:t})}{\psi(\mathbf{x}_{1:t-1})}$ | One Reasoning Step |
|---|---|---|
| The superfactorial $n\$$ is defined as $n\$ = n!^{n!^{\cdots^{n!}}}$ with $n!$ copies. What is the units digit of $4\$$? | 0.32 | I notice that the exponent of 4! is also a multiple of 4!, so I can use the property that $a^b \equiv a^{b \bmod \varphi(b)} \pmod{b}$, where $\varphi(b)$ is Euler's totient function. |
|  | 1.18 | The units digit of a factorial depends only on the units digits of the numbers from 1 to $n$. |

# F  MORE PRELIMINARIES

## F.1  PRELIMINARIES OF IMPORTANCE SAMPLING ESTIMATOR

Let $\sigma(\mathbf{x}_{1:T})$ denote the target distribution, i.e., $\sigma(\mathbf{x}_{1:T}) = \tilde{\sigma}(\mathbf{x}_{1:T})/Z_\sigma$. $\tilde{\sigma}(\mathbf{x}_{1:T}) \geq 0$ denotes the density and $Z_\sigma = \int \tilde{\sigma}(\mathbf{x}_{1:T})\,d\mathbf{x}_{1:T}$ denotes the normalization constant, which is typically intractable. For any given function $h(\mathbf{x}_{1:T})$, direct calculation of its expectations $\mathbb{E}_\sigma[h(\mathbf{x}_{1:T})]$ via direct sampling is infeasible. To address this, importance sampling introduces an auxiliary proposal $q(\mathbf{x}_{1:T})$ from which samples are easily drawn and provides an estimator of the expectation as

$$\mathbb{E}_{\sigma(\mathbf{x}_{1:T})}[h(\mathbf{x}_{1:T})] = \frac{1}{Z_\sigma}\mathbb{E}_{q(\mathbf{x}_{1:T})}\left[\frac{\tilde{\sigma}(\mathbf{x}_{1:T})}{q(\mathbf{x}_{1:T})}h(\mathbf{x}_{1:T})\right] = \frac{\mathbb{E}_{q(\mathbf{x}_{1:T})}\left[\frac{\tilde{\sigma}(\mathbf{x}_{1:T})}{q(\mathbf{x}_{1:T})}h(\mathbf{x}_{1:T})\right]}{\mathbb{E}_{q(\mathbf{x}_{1:T})}\left[\frac{\tilde{\sigma}(\mathbf{x}_{1:T})}{q(\mathbf{x}_{1:T})}\right]}, \tag{29}$$

where $\frac{\tilde{\sigma}(\mathbf{x}_{1:T})}{q(\mathbf{x}_{1:T})}$ is known as the importance weight $W(\mathbf{x}_{1:T})$:

$$W(\mathbf{x}_{1:T}) := \frac{\tilde{\sigma}(\mathbf{x}_{1:T})}{q(\mathbf{x}_{1:T})}. \tag{30}$$

Using independent draws $\mathbf{x}_{1:T}^i \sim q(\mathbf{x}_{1:T})$ for $i = 1, \ldots, N$, we can estimate Eq. (29) via Monte Carlo approximation:

$$\mathbb{E}_{\sigma(\mathbf{x}_{1:T})}[h(\mathbf{x}_{1:T})] \overset{\text{IS}}{\approx} \sum_{i=1}^{N} \frac{W(\mathbf{x}_{1:T}^i)}{\sum_{j=1}^{N} W(\mathbf{x}_{1:T}^j)} h(\mathbf{x}_{1:T}^i), \quad \mathbf{x}_{1:T}^i \sim q(\mathbf{x}_{1:T}). \tag{31}$$

**Bias-variance analysis.** It is well known that importance sampling provide an unbiased estimator (Liu & Lee, 2017; Elvira & Martino, 2021; Zhao et al., 2024b). When $q(\mathbf{x}_{1:T})$ exactly matches $\sigma(\mathbf{x}_{1:T})$, the estimator attains zero variance, but such a perfect proposal is rarely available. In high-dimensional problems, even moderate discrepancies between $q$ and $\sigma$ can accumulate with $T$ steps, dramatically inflating the variance of the weights and necessitating large sample sizes $N$ for accurate estimates (Robert et al., 1999; Doucet & Johansen, 2009).

## F.2  PRELIMINARIES OF VARIATIONAL INFERENCE ESTIMATOR

Let $\sigma(\mathbf{x}_{1:T})$ denote the target distribution, i.e., $\sigma(\mathbf{x}_{1:T}) = \tilde{\sigma}(\mathbf{x}_{1:T})/Z_\sigma$. $\tilde{\sigma}(\mathbf{x}_{1:T}) \geq 0$ denotes the density and $Z_\sigma = \int \tilde{\sigma}(\mathbf{x}_{1:T})\,d\mathbf{x}_{1:T}$ denotes the normalization constant, which is typically intractable. For any given function $h(\mathbf{x}_{1:T})$, direct calculation of its expectations $\mathbb{E}_\sigma[h(\mathbf{x}_{1:T})]$ via direct sampling is infeasible. To address this, variational inference turns inference into an optimization problem by introducing a parametric family of distributions $\mathcal{Q} = \{q(\mathbf{x}_{1:T})\}$. The goal is to select $q \in \mathcal{Q}$ that is closest to $\sigma$ according to a divergence measure, most commonly the KL divergence:

$$\mathcal{L}_{\text{VI}}(q) \triangleq \mathcal{D}_{\text{KL}}\big(q(\mathbf{x}_{1:T}) \,\|\, \sigma(\mathbf{x}_{1:T})\big) = \underbrace{\mathbb{E}_{q(\mathbf{x}_{1:T})}\big[\log q(\mathbf{x}_{1:T}) - \log \tilde{\sigma}(\mathbf{x}_{1:T})\big]}_{-\,\text{ELBO}(q)} + \log \mathcal{Z}_\sigma. \tag{32}$$

Since $\log \mathcal{Z}_\sigma$ does not depend on $q$, minimizing $\mathcal{L}_{\text{VI}}(q)$ is equivalent to maximizing the evidence lower bound (ELBO):

$$q^* = \arg\min_q \mathcal{L}_{\text{VI}}(q) = \arg\max_q (\text{ELBO}(q)). \tag{33}$$

Maximizing the ELBO yields a tractable approximation $q^\star(\mathbf{x}_{1:T}) \in \mathcal{Q}$ to the true target $\sigma(\mathbf{x}_{1:T})$, which can then be used to approximate expectations of interest:

$$\mathbb{E}_{\sigma(\mathbf{x}_{1:T})}[h(\mathbf{x}_{1:T})] \overset{\text{VI}}{\approx} \mathbb{E}_{q^\star(\mathbf{x}_{1:T})}[h(\mathbf{x}_{1:T})] \approx \frac{1}{N}\sum_{i=1}^{N} h(\mathbf{x}_{1:T}^i), \quad \mathbf{x}_{1:T}^i \sim q^\star(\mathbf{x}_{1:T}). \tag{34}$$

**Bias–variance analysis.** In contrast to importance sampling, VI produces a deterministic approximation from which estimation can have low variance (Yao et al., 2018), but at the expense of introducing systematic bias whenever the true $\sigma$ does not belong to the chosen family $\mathcal{Q}$ (Blei et al., 2017; Zhang et al., 2018).

## G  MORE DETAILS

### G.1  DETAILS OF THE ORM BASELINE

To ensure a fair comparison, the ORM baseline is trained on the same data used for training our TSMC-TTS method. Basically, we follow the training protocol of Cobbe et al. (2021) to train the ORM. A binary cross-entropy loss is applied to each token during training, whereas only the final token is used for scoring at inference time.

### G.2  DETAILS OF THE PRM BASELINE

For the PRM baseline, we train models on the PRM800K (Lightman et al., 2024) and MATH-SHEPHERD (Wang et al., 2023) datasets. We basically follow the setting in Feng et al. (2025). The PRM trained on PRM800K is applied to both GSM8K (Cobbe et al., 2021) and MATH (Hendrycks et al., 2021). For MATH-SHEPHERD, which consists of samples from GSM8K and MATH, we train two separate PRMs on the respective portions of the data. During training, a binary cross-entropy loss is applied to the second newline token within each step and to the final token of every sample.

### G.3  DETAILS OF TSMC-TTS IMPLEMENTATION

All experiments run on 8 NVIDIA A100 80GB GPUs. For generator training, we set the learning rate to $2 \times 10^{-5}$, the batch size to 128, and train for 3 epochs. For verifier training, the learning rate is $1 \times 10^{-5}$ for Llemma-7B and $5 \times 10^{-5}$ for DeepSeekMath-7B, with batch sizes of 40 and 80 respectively, and we train for 2 epochs. We set the maximum generation length to 768 tokens for all models and use BF16 precision. During inference, we generate solutions using top-$K$ sampling with $K = 20$ and set the temperature to 0.7. The maximum solution length for inference is fixed at 768 tokens. We define each step as producing a variable-length string $\mathbf{x}_t$ corresponding to one reasoning segment, taken as the text between two double-newline tokens.

### G.4  DETAILS OF TAB. 1

Table 11: A general probabilistic inference framework for PTTS. Existing methods are unified as answer-frequency estimators under a reward-aligned target distribution $\sigma(\mathbf{x}_{1:T})$ via different inference algorithms. For verifier-based methods ▪, the generator is initialized as the pretrained language model $p_0$, while an external verifier (e.g., ORM or PRM) assigns step-wise scores $w(\mathbf{x}_{1:T})$. For generator-based methods ▪, the generator is optimized by $\mathcal{L}_{\mathrm{RFT}}$ in Eq. (24) without using an external verifier. The bias–variance tradeoff inherent in their corresponding inference algorithms is analyzed in Sec. 4.2. Our method ▪ adopts TSMC as the inference algorithm (see Sec. 5.1), jointly optimizing the generator $q^*$ and verifier $\psi^*$ for variance reduction without compromising unbiasednes (see Sec. 5.3).

| Method | Generation (generator) | Verification (score $w(\mathbf{x}_{1:t})$) $t < T$ | $t = T$ | Aggregation | Mapping to Probabilistic Inference: estimator under $\sigma(\mathbf{x}_{1:T})$ Target distribution | Inference Algorithm | Bias | Variance |
|---|---|---|---|---|---|---|---|---|
| MV | $p_0\left(\mathbf{x}_t \mid \mathbf{x}_{1:t-1}\right)$ | 1 | 1 | MV | $p_0(\mathbf{x}_{1:T})$ | – | – | – |
| ORM | $p_0\left(\mathbf{x}_t \mid \mathbf{x}_{1:t-1}\right)$ | 1 | $\mathrm{ORM}(\mathbf{x}_{1:T})$ | WMV | $p_0(\mathbf{x}_{1:T})\mathcal{R}(\mathbf{x}_{1:T})$ | IS | unbiased | high |
| PRM | $p_0\left(\mathbf{x}_t \mid \mathbf{x}_{1:t-1}\right)$ | $\mathrm{PRM}(\mathbf{x}_{1:t})$ | $\mathrm{PRM}(\mathbf{x}_{1:T})$ | WMV | $p_0(\mathbf{x}_{1:T})\mathcal{R}(\mathbf{x}_{1:T})$ | IS | unbiased | high |
| RFT | $\arg\min_q \mathcal{L}_{\mathrm{RFT}}$ | 1 | 1 | MV | $p_0(\mathbf{x}_{1:T})\exp\left(\frac{\mathcal{R}(\mathbf{x}_{1:T})}{\beta}\right)$ | VI | biased | low |
| TSMC-TTS | $q^*\left(\mathbf{x}_t \mid \mathbf{x}_{1:t-1}\right)$ | $\frac{\psi^*(\mathbf{x}_{1:t})}{\psi^*(\mathbf{x}_{1:t-1})}$ | $\frac{\psi^*(\mathbf{x}_{1:T})}{\psi^*(\mathbf{x}_{1:T-1})}$ | WMV | $q^*(\mathbf{x}_{1:T})\mathcal{R}(\mathbf{x}_{1:T})$ | TSMC | unbiased | low |

## H  USE OF LARGE LANGUAGE MODELS (LLMS)

We use LLMs solely to check grammatical correctness and refine the phrasing of certain sentences. All research ideas, methodology, analyses, experiments, and substantive writing are conceived and conducted entirely by the authors.

