# OpenReview forum: "A Bias–Variance Tradeoff Perspective for Improving Test-Time Scaling"
_ICLR.cc/2026/Conference — Submitted to ICLR 2026_

### Official Review · Reviewer_UupB · 2025-10-25

**Soundness:** 1
**Presentation:** 2
**Contribution:** 2
**Rating:** 2
**Confidence:** 4

**Summary:**

The paper proposes to view test-time scaling (TTS) of language models through the lens of bias-variance tradeoff, by viewing "verifier-based methods" (i.e. TTS using outcome reward models (ORMs) or process reward models (PRMs) to score and choose from multiple proposed outputs, without training) as unbiased, high-variance; and "generator-based methods" (i.e. finetuning a model in some way to generate better responses, usually guided by a reward function/model) as biased, low-variance. Then, the authors propose a new method for TTS they call TSMC-TTS (twisted sequential Monte Carlo TTS). They provide experiments on GSM8K and MATH500.

As outlined in "Weaknesses" below, I believe the papers suffers from multiple weaknesses: Large parts of the theory are either trivial or not proven, notation and assumptions are unclear, and the experiments should include TSMC as a baseline and are somewhat limited (only two datasets). Hence, **I recommend rejection of the paper in its current form, and encourage the authors to work on improving on these points.**

**Strengths:**

The bias-variance viewpoint of training-free TTS (viewed as importance sampling) as low bias (unbiased) and high variance, vs finetuning methods as biased, low variance is interesting (yet not new, as the authors themselves say this is "well established", line 191).
The idea of combining TSMC with finetuning could be interesting, if it was better supported and evaluated.
The experimental results seem to show that TSMC-TTS outperforms some baselines, but the results are limited (see "Weaknesses").

**Weaknesses:**

In my view, the paper suffers from many extensive weaknesses.

### Notation
The notation is confusing and seems at least partially wrong. For example, the authors define the reward function as $R(x_{1:T})$; however, reward functions also take as input the prompt $x_0$, which is omitted in this notation; same for $\phi$. It is not clear what is meant with notations like $q(x_{1:T})$ (is the prompt fixed here?). The authors switch between $q$, $p_0$, and $p$, without making it clear how these relate and whether these are the same distributions (I assume $p$ is supposed to be the base distribution and $q$ then one we sample from, possibly finetuned from $p$, but this is never made explicit). E.g., in Equation (1) they use $p$, but when referring to Equation (1) in Proposition 4.1, they replace it by $q$. Various other confusions in the notation arise, e.g.: Is $R$ the same as $W$? How do they relate? Is $R$ supposed to be an unknown ground truth reward? Much of the paper -- e.g. Equation (4) -- only seems to make sense if $R=W$ or they are at least assumed to be close, but this is never made specific. The authors omit clear definitions of various concepts used throughout the paper, for example "verifier", which is used e.g. in Proposition 4.1 without a clear definition; from the Proposition, it seems like "verifier" refers to a PRM, but this is never made explicit. If this is the case, then similar to $R$ and $\phi$, the verifier should also depend on $x_0$.

### Section 4
All claims in Section 4 seems largely trivial or unsupported. Neither Proposition 4.1 nor 4.2 have any real proofs, only "extended versions" in the appendix, that don't contain proofs of any claims either.

In particular, the extended version of Proposition 4.1, i.e. Proposition C.1, seems to rely on an assumption that $R$ and $W$ are close in some sense without making it explicit. Equation (23)/(5) is trivial -- it states two terms that only differ by a constant normalizing factor have the same maximizer. The only interesting part in Proposition 4.1 in my opinion is the "$\approx$" identity, which relates how well the true answer frequency $f_a$ can be approximated by $\hat{f}^{IS}_a$. However, the authors simply state this is approximately equal, without giving any justification for why, or any quantification of the actual approximation (e.g., what would be nice to see is an asymptotic result that shows that as $N$ increases, the approximation becomes arbitrarily good).

Proposition 4.2 has no proof either. Furthermore, much of the notation and claims are unclear to me: How is $q_{VI}$ defined? I don't see how this has anything to do with VI. In VI, a distance/divergence is minimized over a certain class of distributions (e.g. factorized distributions). However, it seems here the authors simply minimize $KL(q ||\sigma)$ without any restrictions, which is trivially minimized by $\sigma$. In particular, this is not VI in the usual sense.
The equivalence $q_{RFT}\Leftrightarrow q_{VI}$ which the authors prove in Appendix D.6 is the well-known result that the optimizer of the KL-regularized objective is the tilted distribution $\sigma\propto p_0 e^{R/\beta}$, which is not novel and does not require a proof. Consequently, Equation (8) (presumably the main proposed contribution of Proposition 4.2) also trivially follows from this relationship. Furthermore, Proposition 4.2 defines $q_{RFT}$ as "the generator trained via RFT [...] by minimizing [the KL regularized objective". However, consequently the authors treat $q_{RFT}$ as the *minimizer* of this objective, which is clearly not the same as the policy one would get from fine-tuning in practice.

To summarize, I do not see any novelty in either Proposition 4.1 or 4.2.

### Experiments
The proposed method combines TSMC with a fine-tuning objective, where the policy and a verifier (parametrized as a head on top of the base policy) are fine-tuned, before running TSMC at inference using the fine-tuned model and verifier (if I understand the algorithm correctly). TSMC is a fairly new concept I was not familiar with, but I had to look it up (https://arxiv.org/abs/2404.17546) since it's not well-explained in the paper. In particular, the authors do not compare vs. vanilla TSMC as a baseline, which would be the most obvious and straightforward baseline to compare against. As a consequence, it remains unclear if the proposed method achieves any improvement over TSMC.

Furthermore, the experimental results are somewhat limited (only two datasets), and do not contain any error bars. The paper would benefit from more extensive evaluation.

**Questions:**

To the best of my knowledge, I have carefully read and evaluated the paper's weaknesses as stated above; however, if the authors believe I gravely misunderstood certain parts of the paper, please point them out to me.
Some questions that I have already stated in the previous section include:

- could you clarify the notation? E.g. why does $R$ not depend on the prompt (same for $\phi$ etc); what's the relationship between $q$, $p_0$, and $p$?
- how do $R$ and $W$ relate?
- could you clarify the contributions of Propositions 4.1 and 4.2? In particular, Equation (5) seems trivial, and the relationship $q_{RFT}\Leftrightarrow q_{VI}$, in your notation, is already well-known, which means Equation (8) does not provide any novel contribution either as far as I can see.
- in what way do you believe VI plays a role? To me it seems you are simply minimizing the objective $KL(q||\sigma)$ without restrictions which is trivially minimized by $\sigma$.
- how does your proposed method compare to TSMC, both conceptually (inference time algorithm) and empirically (experimental accuracy)?

---

> ### Author Response · Authors · 2025-11-20
> **Author Response (1/6)**
>
> We sincerely appreciate your valuable time and constructive feedback. We address each of your concerns carefully below. Points 1–4 are about concerns regarding notation, points 5–9 are about concerns regarding Section 4, and points 10–11 are about concerns regarding the experiments.
>
> ------
>
>
>
> > 1. Omittion of the prompt $\mathbf{x}_{0}$.
>
> We acknowledge that the notation in our paper, such as the reward signal $R(\mathbf{x} _ {1:T})$, the potential function $\psi(\mathbf{x} _ {1:T})$, and distributions like the generator $q(\mathbf{x} _ {1:T})$, should formally depend on the prompt $\mathbf{x}_0$. **To address this concern, we have updated the notation in the Problem Setup (Sec. 3) and Eq. (1) in the updated version**, replacing $R(\mathbf{x} _ {1:T})$, $\psi(\mathbf{x} _ {1:T})$, and $q(\mathbf{x} _ {1:T})$ with the more precise forms $R(\mathbf{x} _ {1:T}\mid \mathbf{x}_0)$, $\psi(\mathbf{x} _ {1:T}\mid \mathbf{x}_0)$, and $q(\mathbf{x} _ {1:T}\mid \mathbf{x}_0)$.
>
> However, immediately after Eq. (1) **in our original manuscript, we explicitly stated that "*for notational simplicity, we omit the conditioning on $\mathbf{x}_{0}$ throughout the remainder of this paper***." Consequently, we do not modify the subsequent notation (e.g., $R(\mathbf{x} _ {1:T})$, $q(\mathbf{x} _ {1:T})$), and all such expressions should be interpreted as implicitly conditioned on the fixed prompt $\mathbf{x}_0$. Since all candidate responses $\mathbf{x} _ {1:T}$ for a given query share the same prompt $\mathbf{x}_0$, omitting the conditioning does not affect any derivations and is solely for notational clarity.
>
> ------
>
> > 2. Relation between $p(\mathbf{x} _ {1:T})$, $q(\mathbf{x} _ {1:T})$ and $p _ 0(\mathbf{x} _ {1:T})$.
>
> We would like to clarify that **the relation between $q(\mathbf{x} _ {1:T})$ and $p _ 0(\mathbf{x} _ {1:T})$ is explicitly stated in the Problem Setup (Sec. 3) in our original manuscript (now Line 119 in the updated version) , where we state that *"$q(\mathbf{x} _ {1:T})$ denote a generator initialized by a pre-trained language model $p _ 0(\mathbf{x} _ {1:T})$"***.
>
> **The role of $p(\mathbf{x} _ {1:T})$ in Eq. (1) is explicitly described in our original manuscript (now Line 130 in the updated version): "*the target distribution $\sigma(\mathbf{x} _ {1:T}\mid \mathbf{x}_0)$ is defined by modulating a base distribution $p$ with a potential function $\phi$."***  That is  $\sigma(\mathbf{x} _ {1:T}) := p(\mathbf{x} _ {1:T}) \cdot \phi(\mathbf{x} _ {1:T}).$ Here, $p(\mathbf{x} _ {1:T})$ is not an additional distribution but a **generic placeholder** to define the general form of the target distribution. Depending on different TTS methods, this placeholder $p(\mathbf{x} _ {1:T})$ is instantiated as either the $q(\mathbf{x} _ {1:T})$ or $p _ 0(\mathbf{x} _ {1:T})$, and this mapping is explicitly shown in the “target distribution’’ column of Table 1 in our manuscript.  This design makes it clear that all methods fit into the same general target-distribution form, differing in how they instantiate the base distribution $p(\mathbf{x} _ {1:T})$.
>
> **To address this concern, we have emphasized the placeholder role of  $p(\mathbf{x} _ {1:T})$ more clearly on Line 136-138 in our updated version**, where we add descriptions like "The base distribution $p$ and potential function $\phi$ serve as flexible placeholders that can be instantiated by different TTS methods summarized in Table 1."
>
> ------
>
>
>
> > 3. Relation between the score $W(\mathbf{x} _ {1:T})$ and reward $R(\mathbf{x} _ {1:T}).$
>
> We would llike to clarify that **the relation between $W(\mathbf{x} _ {1:T})$ and $R(\mathbf{x} _ {1:T})$ is explicitly stated in Appendix C.1 in our original manuscript (now Line 971 in the updated version), where we state that "*the second equality assumes  $W(\mathbf{x} _ {1:T})$ accurately approximates $R(\mathbf{x} _ {1:T})$ at inference time."***
>
> **To address this concern, we have moved this statement into Proposition 4.1 in our updated version.** This assumption is well justified, since the verifier is trained to ensure that the predicted score $W(\mathbf{x} _ {1:T})$ closely matches the reward $R(\mathbf{x} _ {1:T})$ during training.
>
> **To further address this concern from a theoretical perspective, we extend our theoretical framework to the setting with imperfect verifiers and include Theorem C.4  in the updated version.** This theorem quantifies the error bounds under imperfect verifier models. It shows that the bias induced by the imperfect verifier scales only as $O(\delta)$ w.r.t. the verifier approximation error $\delta$, whereas the variance term grows as $O\left(e^{T\cdot C _ {\mathrm{KL}}}-1\right)$ with the step number $T$. Thus, under reasonable verifier accuracy, the verifier-induced bias remains small relative to the exponentially increasing variance. This result extends our theoretical framework to imperfect verifiers and confirms that our original conclusions hold even in the presence of verifier noise.
>
> ------

---

> ### Author Response · Authors · 2025-11-20
> **Author Response (2/6)**
>
> > 4. Concept of "verifier".
>
> We would like to clarify that **the notion of a verifier is explicitly described in Sec. 2 in our original manuscript (now Line 099-101 in the updated version), where we state that "*Verifier-based methods introduce an external verifier to score each candidate for guiding aggregation. Such verifiers are divided into Outcome Reward Models (ORMs) (Cobbe et al., 2021) and Process Reward Models (PRMs) (Uesato et al., 2022).***"
>
> **To address this concern more explicitly, our updated version adds an explicit description of verifiers into Proposition 4.1**, where we restate that verifiers typically fall into ORMs and PRMs. This addition ensures that readers unfamiliar with this terminology can clearly understand what specific forms the verifier can take.
>
> ------
>
>
>
> > 5. All claims in Section 4 seems largely trivial or unsupported. Neither Proposition 4.1 nor 4.2 have any real proofs, only "extended versions".
>
> We would like to clarify that all derivations required for Proposition 4.1 and Proposition 4.2 are fully provided in the “extended version” in the appendix, with explicit pointers from the main paper. **Although these derivations are not placed under a separate “Proof” heading, the "extended versions" contain complete step-by-step justifications.** For example, the derivation of Eq. (4) in Proposition 4.1 is provided in Eq. (22) of the "extended version" and the derivation of Eq. (7) in Proposition 4.2 is provided in Eq. (26) of the "extended version".
>
> Thus, all claims in Section 4 are rigorously supported by explicit derivations in the "extended version", even though these derivations are not labeled with a “Proof” title.
>
> ------
>
>
>
> > 6. Why $f_a\approx\hat{f}_a^{\rm IS}$ in Eq. (4)?
>
> We would like to clarify that this approximation follows directly from the standard Monte Carlo estimation of expectations [1], i.e. $\mathbb{E}(X)\approx\frac{1}{N}\sum_{i=1}^{N}{X}_i$ [1].
>
> Specifically, recall that
>
> $f_a=\frac{\mathbb{E} _ {q\left(\mathbf{x} _ {1: T}\right)}\left[\frac{\tilde{\sigma}\left(\mathbf{x} _ {1: T}\right)}{q\left(\mathbf{x} _ {1: T}\right)} \mathbb{I}\left(\operatorname{Ans}\left(\mathbf{x} _ {1: T}\right)=a\right)\right]}{\mathbb{E} _ {q\left(\mathbf{x} _ {1: T}\right)}\left[\frac{\tilde{\sigma}\left(\mathbf{x} _ {1: T}\right)}{q\left(\mathbf{x} _ {1: T}\right)}\right]}$ (derivation is provided in Eq. (22)).
>
> We approximate the expectations in both the numerator and denominator via Monte Carlo samples $\mathbf{x} _ {1:T}^i\sim q(\mathbf{x} _ {1:T})$ for $i=1,...,N$, yielding
>
> $\hat{f} _ a^{\mathrm{IS}}=\frac{\sum _ {i=1}^N W\left(\mathbf{x} _ {1: T}^i\right) \mathbb{I}\left(\operatorname{Ans}\left(\mathbf{x} _ {1: T}^i\right)=a\right)}{\sum _ {j=1}^N W\left(\mathbf{x} _ {1: T}^j\right)}, \quad W\left(\mathbf{x} _ {1: T}\right):=\frac{\tilde{\sigma}\left(\mathbf{x} _ {1: T}\right)}{q\left(\mathbf{x} _ {1: T}\right)} .$
>
> Thus, the relation $f_a\approx\hat{f}_a^{\rm IS}$ is simply the standard Monte Carlo approximation of expectations.
>
> By the law of large numbers, we have $\hat f_a^{\mathrm{IS}}\xrightarrow{a.s.}f_a$ as $N\to\infty$, and the asymptotic error is $O(N^{-1/2})$ (see chapter 2 in [1]).
>
> **To address this concern, our updated version now adds in Proposition 4.1 (Line 168) the sentence:  "The approximation is justified by the Monte Carlo estimation of expectations and its asymptotically vanishing error [1]."**
>
> [1] Monte Carlo theory, methods and examples, 2013.
>
> ------
>
>
>
> > 7. Definition and role of $q_{\rm VI}$. In what way do you believe VI plays a role? To me it seems you are simply minimizing the objective $KL(q||\sigma)$ without restrictions which is trivially minimized by $\sigma$​.
>
> Our use of VI is not an unconstrained minimization over all distributions. Rather, the optimization $q_{\rm VI}=\arg \min _q\left[\mathcal{D} _ {\mathrm{KL}}\left(q\left(\mathbf{x} _ {1: T}\right) \| \sigma\left(\mathbf{x} _ {1: T}\right)\right)\right]$ is optimized **within the parametric family induced by the generator network**. That is, $q$ is restricted to the expressivity of the neural generator, exactly following the standard amortized variational inference paradigm [2,3],  where the variational distribution is amortized through a shared neural network.
>
> **To address this concern, we have added statement "$q _ {\rm VI}$ is constrained to lie within the parametric family induced by the generator network" in Proposition 4.2 (Line 195) in the updated version.**
>
> [2] Auto-encoding variational bayes, 2013.
>
> [3] Amortized Bayesian Meta-Learning, 2019.
>
> ------

---

> ### Author Response · Authors · 2025-11-20
> **Author Response (3/6)**
>
> > 8. The minimizer of this RFT objective is clearly not the same as the policy one would get from fine-tuning in practice.
>
> We agree with the reviewer that the generator obtained by practical RFT fine-tuning does not exactly match the ideal minimizer of the RFT objective. This discrepancy indeed reflects an optimization bias. **To address this concern, we extend our theoretical framework to formally characterize this bias and provide the following theorem in the updated version (see Appendix C.3).**
>
> **Theorem C.3 (Bias decomposition under approximate RFT optimization).** Let the true target distribution be $\sigma\left(\mathbf{x} _ {1: T}\right) \propto p _ 0\left(\mathbf{x} _ {1: T}\right) R\left(\mathbf{x} _ {1: T}\right)$ with normalizing constant $Z _ \sigma=\int p _ 0\left(\mathbf{x} _ {1: T}\right) R\left(\mathbf{x} _ {1: T}\right) d \mathbf{x} _ {1: T}$. Define the optimum regarding the RFT objective and VI objective within the parametric family induced by the generator network as $q _ {\rm RFT} \iff q _ {\rm VI}=\arg\min _ {q}{\rm KL}(q\left(\mathbf{x} _ {1: T}\right) \| \sigma\left(\mathbf{x} _ {1: T}\right))$. Let $q'$ denote the  generator obtained by a practical optimizer (e.g., RFT fine-tuning) and assume its suboptimality is bounded by $\mathrm{KL}({q}' \| \sigma) \leq \mathrm{KL}\left(q _ {\mathrm{VI}} \| \sigma\right)+\varepsilon_{\mathrm{opt}}$. Consider the VI normalizing-constant estimator $\hat{Z} _ \sigma^{\mathrm{VI}}(q')=Z _ \sigma \exp (-\mathrm{KL}(q' \| \sigma))$ (derivations are provided in the proof). Then the estimation bias satisfies
>
> $\left|Z _ \sigma-\hat{Z} _ \sigma^{\mathrm{VI}}(q')\right| \leq Z _ \sigma\left(1-e^{-\mathrm{KL}\left(q _ {\mathrm{VI}} \| \sigma\right)}\right)+Z _ \sigma\left(1-e^{-\varepsilon_{\mathrm{opt}}}\right)=O\left(Z _ \sigma(1-e^{-C _ {\mathrm{KL}} \cdot T} )\right)+O(Z _ \sigma\varepsilon _ {\mathrm{opt}}),$
>
> where the first term corresponds to the variational mode-seeking bias, and the second term corresponds to the optimization bias (due to not reaching the optimum exactly). Here $C _ {\mathrm{KL}} > 0$ is a max per-step KL divergence such that the sequence-level KL divergence satisfies $\mathrm{KL}(q_{\rm VI} | \sigma) \le C _ {\mathrm{KL}} \cdot T$ under the standard autoregressive factorization over $T$ steps. The second term follows from $1-e^{-x}=O(x)$ as $x\rightarrow 0$​, which applies when the optimization error becomes near 0.
>
>
>
> **Remark.** The above theorem shows that the total bias under practical RFT optimization decomposes into a variational mode-seeking bias and an optimization bias. Crucially, only the variational term scales with the step number $T$, while the optimization term does not accumulate with the number of steps. **Consequently, when $\varepsilon_{\mathrm{opt}}$ is small, the optimization bias remains lower order and is dominated by the variational term, particularly in long-sequence LLM reasoning settings (i.e., large $T$).** This confirms that our main conclusions continue to hold in this setting.

---

> ### Author Response · Authors · 2025-11-20
> **Author Response (4/6)**
>
> **Proof of Theorem C.3.** Let the true target be
>
> $\sigma\left(\mathbf{x} _ {1: T}\right) \propto p_0\left(\mathbf{x} _ {1: T}\right) R\left(\mathbf{x} _ {1: T}\right), \quad Z_\sigma=\int p_0\left(\mathbf{x} _ {1: T}\right) R\left(\mathbf{x} _ {1: T}\right) d \mathbf{x} _ {1: T}.$
>
> Next, consider the VI estimator $Z _ {\sigma}$ under the target distribution $\sigma$.
>
> By the standard variational identity,
>
> $\log Z _ {\sigma}=\operatorname{ELBO}(q)+\operatorname{KL}\left(q || \sigma\right),$
>
> we have
>
> $\hat{Z} _ {\sigma}^{\mathrm{VI}}=\exp (\operatorname{ELBO}(q))=\exp \left(\log Z _ \sigma -\mathrm{KL}\left(q || \sigma\right)\right)=Z _ {\sigma} \exp \left(-\mathrm{KL}\left(q || \sigma\right)\right).$
>
> Thus,
>
> $\left|Z _ \sigma-\hat{Z} _ \sigma^{\mathrm{VI}}(q)\right|=Z _ \sigma(1-\exp (-\mathrm{KL}(q || \sigma))).$
>
> Let
>
> $q _ {\mathrm{VI}}=\arg \min _q \mathrm{KL}(q || \sigma)$
>
> within the generator family and let the practical optimizer return $q'$ satisfying
>
> $\mathrm{KL}\left(q^{\prime} \| \sigma\right) \leq \mathrm{KL}\left(q _ {\mathrm{VI}} || \sigma\right)+\varepsilon_{\mathrm{opt}} .$
>
> We obtain
>
> $1-e^{-\mathrm{KL}\left(q^{\prime} \| \sigma\right)} \leq 1-e^{-\left(\mathrm{KL}\left(q _ {\mathrm{VI}} || \sigma\right)+\varepsilon_{\mathrm{opt}}\right)} =1-e^{-\mathrm{KL}\left(q_{\mathrm{VI}} || \sigma\right)}+e^{-\mathrm{KL}\left(q_{\mathrm{VI}} || \sigma\right)}\left(1-e^{-\varepsilon_{\mathrm{opt}}}\right) .$
>
> Multiplying by $Z _ {\sigma}$ yields the desired decomposition:
>
> $\left|Z _ \sigma-\hat{Z} _ \sigma^{\mathrm{VI}}\left(q^{\prime}\right)\right| \leq Z _ \sigma\left(1-e^{-\mathrm{KL}\left(q _ {\mathrm{VI}} || \sigma\right)}\right)+Z _ \sigma\left(1-e^{-\varepsilon _ {\mathrm{opt}}}\right) .$
>
>  Suppose there exists the max per-step KL divergence $C _ {\mathrm{KL}}>0$ such that $\mathrm{KL}\left(q _ {\mathrm{VI}} || \sigma\right) \leq C _ {\mathrm{KL}} T$ under the standard autoregressive factorization. Thus,
>
> $Z_\sigma\left(1-e^{-\mathrm{KL}\left(q _ {\mathrm{VI}} || \sigma\right)}\right)=O\left(Z_\sigma\left(1-e^{-C _ {\mathrm{KL}} T}\right)\right).$
>
> Since $1-e^{-x}=O(x)$ when $x\rightarrow 0$,
>
> $Z_\sigma\left(1-e^{-\varepsilon_{\mathrm{opt}}}\right)=O\left(\varepsilon_{\mathrm{opt}}\right),$
>
> which does not depend on the step number $T$.
>
> Combine both to obtain:
>
> $\left|Z _ \sigma-\hat{Z} _ \sigma^{\mathrm{VI}}\left(q^{\prime}\right)\right| \leq O\left(Z _ \sigma\left(1-e^{-C _ {\mathrm{KL}} T}\right)\right)+O\left(\varepsilon _ {\mathrm{opt}}\right) .$
>
> This completes the proof.
>
> ------

---

> ### Author Response · Authors · 2025-11-20
> **Author Response (5/6)**
>
> > 9. Contributions of Propositions 4.1 and 4.2.
>
> - Proposition 4.1 establishes the formal connection between verifier-based TTS methods and importance sampling (IS) estimators of answer frequency. **However, existing works treat verifier-based TTS methods as heuristic reranking or selection, but do not connect them to a principled IS estimator of answer frequency.** This IS view is essential because it allows us to analyze the verifier-based TTS methods under a new and principled bias-variance tradeoff perspective, which previous papers did not explore.
>
>
>
> - For Proposition 4.2, we acknowledge that prior work has noted that the minimizer of the RFT objective takes the form of a tilted distribution $p_0 \cdot \exp(R/\beta)$. **However, these earlier works do not articulate how this expression connects RFT-based methods to the variance-inference (VI) estimator of answer frequency in the context of TTS.** In contrast, Proposition 4.2 repurposes this known expression as a bridge between RFT-based TTS methods and VI estimators in the context of TTS. This connection enables a bias–variance tradeoff analysis of RFT-based approaches, which has not been explored in prior literature.
>
>
>
>
> - **Clarifying our core contributions.** We emphasize that Propositions 4.1 and 4.2 are not the core contributions of our paper. They are just instantiations of our proposed broader conceptual framework for TTS, and are included to illustrate how existing TTS methods can be instantiated within this general framework. In contrast, the conceptual framework itself is one of our core contributions, rather than the individual propositions.
>
>   To avoid confusion, we have revised the Introduction part in the updated version to state our core contributions more clearly. Below, our core contributions are briefly summarized as
>
>   1. **The first general framework for TTS through a unifying probabilistic inference formulation**, seamlessly encompassing previously disparate methods as special cases. This framework offers **a new bias–variance tradeoff perspective** to reveal the intrinsic limitations of existing TTS methods and provides **a principled foundation for developing new TTS method**.
>   2. **The first theoretical formulation of bias–variance tradeoff in the context of TTS (Theorem 5.2)**, which quantitatively characterizes how the relative variance upper bound is jointly governed by the generator and the verifier.  **We believe such a theoretical perspective offers valuable insights for the TTS community largely driven by heuristics.**
>   3. **Theory-driven verifier–generator optimization algorithm and self-evolving TSMC mechanism.** Our proposed TSMC-TTS is a principled instantiation of our general framework, which performs EM-like optimization of the generator and verifier guided by the theoretical variance bound, achieving monotonic variance reduction  without compromising unbiasedness. Furthermore, we introduce a self-evolving TSMC mechanism that alleviates both the reward sparsity issue and the substantial computational overhead inherent with vanilla TSMC.
>
> We kindly refer the reviewer to the updated introduction section for a better understanding of our core contributions.
>
> ------

---

> ### Author Response · Authors · 2025-11-20
> **Author Response (6/6)**
>
> > 10. Comparison with Vanilla TSMC.
>
> - **Conceptual comparison.** Vanilla TSMC is an approximate inference method that performs sequential sampling and estimation with respect to an unnormalized target distribution [4]. However, when applied to the TTS context, its inherent probabilistic bias–variance tradeoff has not been analyzed, and there is no principled way to address it. **Our work provides fundamental improvements over Vanilla TSMC both theoretically and algorithmically along three aspects.**
>
>   1. First, we give the **first theoretical formulation of the bias–variance tradeoff of TSMC in the TTS context**. In particular, Theorem 5.2 quantitatively characterizes how the relative variance upper bound of TSMC is governed jointly by the generator and verifier, revealing a theoretical framework that has not been identified in prior TSMC work.
>   2. Second, our theoretical insight enables a **theory-driven verifier–generator optimization algorithm for TSMC**. Under our variance decomposition, which reveals both a verifier gap and a generator gap, Vanilla TSMC is provably suboptimal due to its fundamental incompatibility with the conditions required for optimal variance reduction. In contrast, our EM-like optimization jointly updates the generator and verifier by directly targeting the variance bound, achieving monotonic variance reduction without compromising unbiasedness.
>   3. Third, we introduce a new **self-evolving TSMC mechanism** that uses a self-evolving target distribution proportional to $q(\mathbf{x} _ {1:T})R(\mathbf{x} _ {1:T})$, where $q$ is iteratively updated via our EM-like optimization.. This elegant plug-and-playd design mitigates both the reward sparsity issue and the substantial computational overhead inherent in vanilla TSMC. As a result, our approach significantly enhances the practical applicability of TSMC.
>
>   Together, these advances extend Vanilla TSMC both theoretically and algorithmically.
>
>
>
> - **Empirical comparison.** As shown in the table below, our proposed method consistently and significantly outperforms the Vanilla TSMC across different model architectures and datasets. This empirical improvement is fully aligned with our theoretical insights: Vanilla TSMC is provably suboptimal due to its incompatibility with the conditions required for optimal variance reduction, whereas our method explicitly targets the derived variance bound for effective variance reduction. The observed gains therefore validate the necessity and effectiveness of our theory-guided improvements.
>
>   | Method           | Llemma-7B (GSM8K) | Llemma-7B (MATH500) | DeepSeekMath-7B (GSM8K) | DeepSeekMath-7B (MATH500) |
>   | ---------------- | ----------------- | ------------------- | ----------------------- | ------------------------- |
>   | Vanilla TSMC [4] | 78.4              | 43.8                | 88.4                    | 57.4                      |
>   | TSMC-TTS (ours)  | 82.2              | 47.2                | 92.6                    | 61.8                      |
>
>   [4] Probabilistic Inference in Language Models via Twisted Sequential Monte Carlo, 2024.
>
> ------
>
>
>
> > 11. Experiments on more datasets.
>
> To further demonstrate the effectiveness of our method, we extend our experiments across **a wider range of reasoning datasets and model architectures**. Specifically, we add evaluations on several **generalized reasoning** datasets, including FinanceBench [5] (financial domain) and NumGLUE Task 2 Chemistry [6] (scientific domain). We also evaluate our approach on additional model of Llama-3.1-8B-Instruct.
>
> As shown in the table below, our method consistently and significantly outperforms all baselines. These results indicate that our approach is not only highly effective for mathematical reasoning but also generalizes well to broader domain-specific reasoning scenarios. To address this concern, we have added this experiment to the main paper of the updated version (see Table 4 on LLine 494).
>
> | Method          | Llama 3.1-8B-Instruct (FinanceBench) | Llama 3.1-8B-Instruct (NumGLUE Task 2 Chemistry) |
> | --------------- | ------------------------------------ | ------------------------------------------------ |
> | ORM             | 67.67                                | 79.69                                            |
> | PRM             | 68.33                                | 80.92                                            |
> | TSMC-TTS (ours) | 72.67                                | 86.92                                            |
>
> [5] Financebench: A new benchmark for financial question answering. 2023.
>
> [6] NumGLUE: A suite of fundamental yet challenging mathematical reasoning tasks. 2022.

---

### Official Review · Reviewer_CfpZ · 2025-10-31

**Soundness:** 3
**Presentation:** 2
**Contribution:** 2
**Rating:** 4
**Confidence:** 3

**Summary:**

This paper proposes an improvement to Test-time Scaling (TTS) from a bias-variance perspective. The authors first categorize existing TTS improvement methods into two groups: verifier-based approaches, which aggregate answers using external verifiers, and generator-based approaches, which enhance the generation model to produce better candidates.

The authors then introduce a probabilistic inference framework that unifies these two TTS-augmentation paradigms from the perspective of bias and variance. Through theoretical derivation, they show that the verifier-based method corresponds to importance sampling, which has low bias but high variance, while the generator-based method corresponds to variational inference, which has high bias but low variance.

Building on this perspective, the authors propose TSMC-TTS, a method that employs twisted sequential Monte Carlo as the inference algorithm to reduce variance while maintaining unbiasedness. Furthermore, they use a theoretically derived upper bound to jointly optimize the generator and the verifier, thereby reducing variance without sacrificing unbiasedness.

The experimental results on gsm8k and MATH show that the proposed method outperform the verifier-based method and the generator-based method.

**Strengths:**

1. The bias-variance perspective proposed by the authors is highly novel, offering a new way to understand and analyze inference-time scaling methods.
2. The authors conducted extensive theoretical derivations to support their conclusions, such as establishing the correspondence between their framework and methods like importance sampling and variational inference.
3. The authors designed a targeted optimization method based on their analysis and supported it with rigorous theoretical justification.

**Weaknesses:**

1. The excessive use of derivations and mathematical notations makes the paper difficult to read, especially since even simple concepts (like majority voting) and methods are expressed with an overabundance of symbols and formulas.
2. The experimental evaluation is insufficient, as experiments were only conducted on Llemma-7B and DeepSeek-7B, neither of which are state-of-the-art models. Moreover, the evaluation was limited to just two datasets, GSM8K and MATH, lacking validation across a broader range of domains and benchmarks.

**Questions:**

Is the experimental comparison fair? TSMC-TTS is a method that combines both verifier-based and generator-based approaches, yet it is only compared against individual methods. It remains unclear what the results would be if a naïve combination of the verifier-based and generator-based approaches were implemented for comparison.

Why is the Generator-based approach categorized as a method for improving TTS? For example, methods like RFT are primarily training-time enhancement techniques rather than inference-time ones.

---

> ### Author Response · Authors · 2025-11-20
> **Author Response**
>
> We sincerely appreciate your valuable time and constructive comments. We also appreciate your acknowledgment of our novel perspective and solid theoretical justifications. Below, we carefully address your concerns in detail.
>
> ------
>
>
>
> > (Weakness 1): The excessive use of derivations and mathematical notations makes the paper difficult to read, especially since even simple concepts (like majority voting) and methods are expressed with an overabundance of symbols and formulas.
>
> We appreciate the reviewer’s feedback regarding readability and the use of mathematical notation. We would like to clarify that our intention was not to overcomplicate simple concepts, but to provide a unified and rigorous formulation that can encompass diverse existing TTS methods (both verifier-based and generator-based) within a general probabilistic inference framework. Achieving this unification requires expressing even familiar procedures (such as majority voting) using additional notation, for example, by formulating them in the same style as importance-sampling estimators with explicit proposal distributions and importance weights, so that all methods can be analyzed within a single coherent mathematical framework.
>
> ------
>
>
>
> > (Weakness 2): Lacking validation across a broader range of domains and benchmarks.
>
> To further demonstrate the effectiveness of our method, we extend our experiments across **a wider range of reasoning datasets and model architectures**. Specifically, we add evaluations on several **generalized reasoning** datasets, including FinanceBench [1] (financial domain) and NumGLUE Task 2 Chemistry [2] (scientific domain). We also evaluate our approach on an additional model of Llama-3.1-8B-Instruct.
>
> As shown in the table below, our method consistently and significantly outperforms all baselines. These results indicate that our approach is not only highly effective for mathematical reasoning but also generalizes well to broader domain-specific reasoning scenarios. To address this concern, we have added this experiment on more datasets in our updated version.
>
> | Method          | Llama 3.1-8B-Instruct (FinanceBench) | Llama 3.1-8B-Instruct (NumGLUE Task 2 Chemistry) |
> | --------------- | ------------------------------------ | ------------------------------------------------ |
> | ORM             | 67.67                                | 79.69                                            |
> | PRM             | 68.33                                | 80.92                                            |
> | TSMC-TTS (ours) | 72.67                                | 86.92                                            |
>
> [1] Financebench: A new benchmark for financial question answering. 2023.
>
> [2] NumGLUE: A suite of fundamental yet challenging mathematical reasoning tasks. 2022.
>
> ------
>
>
>
> > (Question 1): Is the experimental comparison fair? It remains unclear what the results would be if a naive combination of the verifier-based and generator-based approaches were implemented for comparison.
>
> We would like to clarify that Table 2 in our original manuscript includes the "naive combinations of generator-based and verifier-based approaches". Specifically, the baselines **RFT+ORM and RFT+PRM** pair an RFT-trained generator with an off-the-shelf ORM/PRM verifier, which reflects the comparison the reviewer asked about. Our method outperforms these naive combinations by about 8%, which aligns with our bias–variance tradeoff analysis: such direct combinations still inherit the generator’s training bias and therefore scale poorly with larger solution budgets. In contrast, our proposed TSMC-TTS jointly optimizes the generator and verifier under our derived theoretical variance bound, enabling monotonic variance reduction without compromising unbiasedness.
>
> To address this concern, we have added a dedicated discussion of these hybrid baselines in the updated version (Lines 414–416 and 430–453).
>
> ------
>
>
>
> >(Question 2): Why is the Generator-based approach categorized as a method for improving TTS? For example, methods like RFT are primarily training-time enhancement techniques rather than inference-time ones.
>
> Our categorization of generator-based versus verifier-based methods follows **the structural role** they play in the TTS framework, rather than whether they operate during training or inference. In our proposed general TTS framework, both the generator and the verifier explicitly affect the final TTS performance (see Theorem 5.2). **This theorem directly implies that improving the generator component (e.g., via RFT) is a valid and principled mechanism for enhancing TTS performance, which motivates and justifies our use of the term generator-based methods.**

---

### Official Review · Reviewer_MrsS · 2025-10-31

**Soundness:** 3
**Presentation:** 2
**Contribution:** 2
**Rating:** 4
**Confidence:** 3

**Summary:**

This paper introduces a probabilistic inference framework for Parallel Test-Time Scaling (PTTS), casting existing methods into a bias-variance tradeoff perspective. It frames verifier-based methods as high-variance Importance Sampling (IS) estimators and generator-based methods as biased Variational Inference (VI) estimators. To resolve this tradeoff, the authors propose TSMC-TTS, a novel algorithm based on Twisted Sequential Monte Carlo designed to reduce variance while aiming to preserve unbiasedness. The method is validated empirically, achieving state-of-the-art results on mathematical reasoning benchmarks.

**Strengths:**

- Elegant Theoretical Framework: The paper's primary strength is its novel and insightful unification of verifier-based (IS) and generator-based (VI) methods under the bias-variance tradeoff. This provides a much-needed theoretical lens for a field largely driven by heuristics.

- Principled and Effective Algorithm: The proposed TSMC-TTS is a principled instantiation of the framework. The joint generator-verifier optimization, guided by a variance bound, is a sophisticated and well-motivated approach that directly addresses the identified limitations of prior work.

- Clever Handling of Practical Challenges: The adoption of a self-evolving target distribution σ(x) ∝ q(x)R(x) is a significant strength. It effectively mitigates the reward sparsity and high computational costs associated with relying on a fixed base model p0.

- Strong Empirical Validation: The experiments are thorough and the results are compelling. The method achieves state-of-the-art performance, and the scaling curves (Fig 1, 2) provide strong empirical evidence for the theoretical claims regarding the limitations of IS- and VI-based approaches.

**Weaknesses:**

- Fragile "Unbiasedness" Claim: The paper's central claim of "preserving unbiasedness" is misleading as it relies on the unrealistic assumption that the learned verifier is perfect (ψ = R). In practice, any learned verifier has its own estimation error, which introduces a verifier-induced bias. The paper's analysis completely overlooks this crucial factor and lacks a discussion of how this bias propagates and potentially dominates the variance term, thereby weakening the main argument for adopting a low-variance estimator like TSMC.

- Theory-Practice Gap in Reward Signal: The theoretical analysis of generator-based methods (VI) assumes access to a true reward signal R(x). However, all practical RFT/RLHF implementations rely on a learned, and therefore biased, reward model RM(x). This disconnect means the theory analyzes an ideal scenario, while the experiments operate in a practical (biased) one. Consequently, the paper conflates two different sources of error: the intrinsic mode-seeking bias of VI and the external bias from the imperfect reward model. The framework, in its current form, cannot disentangle these two effects.

**Questions:**

1. The claim of unbiasedness for TSMC-TTS hinges on ψ = R. Since this is never true in practice, how does the verifier error, say ||ψ - R||, propagate through the TSMC estimator? Can you provide any analysis on how this verifier-induced bias compares in magnitude to the variance reduction gained from using TSMC?

2. Your VI analysis is based on a true reward R(x), but practical RFT uses a learned reward model RM(x). What does the theoretical equivalence between RFT and VI hold when considering a biased RM(x)? Is it possible that the observed "VI bias" discussed in the paper is largely dominated by this external "reward model bias"?

3. For practical applications, the trade-off between performance and computational cost is critical. Could you provide a more detailed cost-benefit analysis (e.g., final accuracy vs. total FLOPs or wall-clock time) comparing TSMC-TTS against simpler but strong baselines, such as a well-fine-tuned generator using only majority voting?

---

> ### Author Response · Authors · 2025-11-20
> **Author Response (1/4)**
>
> We sincerely appreciate your valuable time and constructive comments. We are encouraged by your recognition of the theoretical elegance of our framework and the principled nature of our method. Below, we carefully address your concerns point by point.
>
> ------
>
> > (Weakness 1 & Question 1) Concerns about the fragile "unbiasedness" claim.
>
> - **Regarding the concern "The paper's analysis completely overlooks verifier-induced bias and lacks a discussion of how this bias propagates and potentially dominates the variance term.**
>
>   We acknowledge that the unbiasedness property of our method is based on the assumption of an oracle verifier (i.e., $\psi(\mathbf{x} _ {1:T})=R(\mathbf{x} _ {1:T})$). In practice, however, the verifier $\psi$​​ is learned and inevitably imperfect, which introduces verifier-induced bias.
>
>   **We address this concern from both theoretical and empirical perspectives.** We derive a formal error bound theorem that explicitly characterizes the verifier-induced bias and facilitates a quantitative comparison between this bias and the variance term. We have included this theorem in our updated version (see Theorem C.4 in Appendix C.4).
>
>   **Theorem 1 (Error bounds for TSMC-TTS estimator under imperfect verifiers).** Let the true target distribution be $\sigma\left(\mathbf{x} _ {1: T}\right) \propto q\left(\mathbf{x} _ {1: T}\right) R\left(\mathbf{x} _ {1: T}\right)$, with normalizing constant $\mathcal{Z} _ \sigma=\int q\left(\mathbf{x} _ {1: T}\right) R\left(\mathbf{x} _ {1: T}\right) d \mathbf{x} _ {1: T}$. Assume the imperfect verifier satisfies $\psi\left(\mathbf{x} _ {1: T}\right)=R\left(\mathbf{x} _ {1: T}\right)+\varepsilon\left(\mathbf{x} _ {1: T}\right), \quad\left|\varepsilon\left(\mathbf{x} _ {1: T}\right)\right| \leq \delta$, where $\varepsilon(\mathbf{x} _ {1:T})$ denotes the verifier approximation error. Let $\hat{\mathcal{Z}} _ {\sigma}$ be the TSMC-TTS estimator satisfying $\mathbb{E} _ {q}[\hat{\mathcal{Z}} _ {\sigma}]=\int q(\mathbf{x} _ {1:T}) \psi(\mathbf{x} _ {1:T}) d\mathbf{x} _ {1:T}$ (see derivations in Appendix D.1). Then the verifier-induced bias satisfies
>
>   $\operatorname{Bias}\left(\hat{\mathcal{Z}} _ \sigma\right)=\left|\mathbb{E} _ q\left[\hat{\mathcal{Z}} _ \sigma\right]-\mathcal{Z} _ \sigma\right| \leq \delta,$
>
>   i.e., $\operatorname{Bias}\left(\hat{\mathcal{Z}} _ \sigma\right)=O(\delta), \quad \operatorname{Bias}^2\left(\hat{\mathcal{Z}} _ \sigma\right)=O\left(\delta^2\right)$, and does not depend on the sequential step number $T$.
>
>   From Theorem 5.2, the variance of the TSMC-TTS estimator satisfies $\log (\operatorname{Var}(\hat{\mathcal{Z}} _ {\sigma})+1) \leq \mathcal{B}(\pi, q)$, where $\mathcal{B}(\pi,q)$ is a sum of $T$ step-wise KL divergence terms. Denoting the max per-step KL divergence term by $C _ {\rm KL}$, we obtain
>
>   $\operatorname{Var}\left(\hat{\mathcal{Z}} _ \sigma\right) \leq e^{T \cdot C _ {\mathrm{KL}}}-1.$
>
>   The mean-squared error therefore decomposes as [1]
>    $\operatorname{MSE}\left(\hat{\mathcal{Z}} _ \sigma\right)=\operatorname{Bias}^2\left(\hat{\mathcal{Z}} _ \sigma\right)+\operatorname{Var}\left(\hat{\mathcal{Z}} _ \sigma\right)=O\left(\delta^2\right)+O\left(e^{T \cdot C _ {\mathrm{KL}}}-1\right),$
>
>   where the first term refers to the verifier-induced bias and the second term corresponds to  the variance term.
>
>   **Remark.** The error bound theorem shows that the verifier-induced bias propagates only linearly with the verifier approximation error and, importantly, is independent of the step number $T$. In contrast, the variance term scales exponentially with the step number $T$, where $T$ is typically large in LLM reasoning tasks. Therefore, **Theorem 1 suggests that the verifier-induced bias remains controlled under reasonable verifier accuracy (small $\delta$), whereas the variance term is the dominant factor in the overall error.**
>
> - **Regarding the question "how does the verifier error propagate through the TSMC estimator?"**
>
>   As shown in the theorem above, $\operatorname{Bias}\left(\hat{\mathcal{Z}} _ \sigma\right)=O\left(\delta\right)$ implies that  the verifier-induced bias propagates only linearly with the verifier approximation error $\delta$. Moreover, the bound is independent of the step number $T$, meaning that the verifier-induced bias does not accumulate across the sequential steps.
>
>
> - **Regarding the question "how this verifier-induced bias compares in magnitude to the variance reduction gained from using TSMC?"**
>
>   The theorem above quantifies this comparison explicitly. The magnitude of verifier-induced bias $O(\delta)$ is inherently much smaller than that of the variance term $O\left(e^{T \cdot C _ {\mathrm{KL}}}-1\right)$ because: (i) the bias grows only linearly in $\delta$, whereas the variance grows exponentially with $T \cdot C _ {\rm KL}$; (ii) the bias does not accumulate over the step number $T$, while the variance increases exponentially with $T$​, especially in long-sequence LLM reasoning tasks.

---

> ### Author Response · Authors · 2025-11-20
> **Author Response (2/4)**
>
> - **Empirical support for our claim of adopting a low-variance estimator even under the imperfect verifiers.**
>
>   To support our argument that adopting a low-variance estimator is crucial even in the presence of verifier-induced bias, we conduct experiments comparing a low-variance estimator (our method) with a high-variance estimator (ORM) under different levels of verifier-induced bias. We simulate varying degrees of verifier-induced bias by using verifiers at different training stages: a normally trained verifier represents a low-bias regime, while an early-stage verifier represents a high-bias regime.
>
>   Efficacy of our method under low/normal verifier-induced bias.
>
>   | Method          | Variance | Llemma-7B (GSM8K) | Llemma-7B (MATH500) | DeepSeekMath-7B (GSM8K) | DeepSeekMath-7B (MATH500) |
>   | --------------- | -------- | ----------------- | ------------------- | ----------------------- | ------------------------- |
>   | ORM             | High     | 77.6              | 42.6                | 88.2                    | 56.2                      |
>   | TSMC-TTS (ours) | Low      | **82.2**          | **47.2**            | **92.6**                | **61.8**                  |
>
>   Efficacy of our method under higher verifier-induced bias.
>
>   | Method          | Variance | Llemma-7B (GSM8K) | Llemma-7B (MATH500) | DeepSeekMath-7B (GSM8K) | DeepSeekMath-7B (MATH500) |
>   | --------------- | -------- | ----------------- | ------------------- | ----------------------- | ------------------------- |
>   | ORM             | High     | 74.8              | 39.2                | 85.6                    | 53.6                      |
>   | TSMC-TTS (ours) | Low      | **80.6**          | **45.4**            | **91.2**                | **59.8**                  |
>
>   As shown in the two tables above, our method (using a low-variance estimator) consistently and significantly outperforms ORM (using a higher-variance estimator) across different levels of verifier-induced bias. These empirical findings are fully consistent with our theoretical analysis, which shows that the magnitude of verifier-induced bias is smaller in realistic settings. **Consequently, variance reduction remains the key factor for achieving better performance in practice, even when the verifier is imperfect.**
>
>
>
> **Proof of Theorem 1.** By definition,
>
> $\mathcal{Z} _ \sigma=\int q\left(\mathbf{x} _ {1: T}\right) R\left(\mathbf{x} _ {1: T}\right) d \mathbf{x} _ {1: T}, \quad \mathbb{E} _ q\left[\hat{\mathcal{Z}} _ \sigma\right]=\int q\left(\mathbf{x} _ {1: T}\right) \psi\left(\mathbf{x} _ {1: T}\right) d \mathbf{x} _ {1: T}.$
>
> Substituting $\psi(\mathbf{x} _ {1:T})=R(\mathbf{x} _ {1:T})+\varepsilon(\mathbf{x} _ {1:T})$, we obtain
>
> $\mathbb{E} _ q\left[\hat{\mathcal{Z}} _ \sigma\right]-\mathcal{Z} _ \sigma=\int q\left(\mathbf{x} _ {1: T}\right) \varepsilon\left(\mathbf{x} _ {1: T}\right) d \mathbf{x} _ {1: T}.$
>
> Taking absolute values and using $|\varepsilon| \leq \delta$,
>
> $\left|\mathbb{E} _ q\left[\hat{\mathcal{Z}} _ \sigma\right]-\mathcal{Z} _ \sigma\right| \leq \int q\left(\mathbf{x} _ {1: T}\right)\left|\varepsilon\left(\mathbf{x} _ {1: T}\right)\right| d \mathbf{x} _ {1: T} \leq \delta.$
>
> Thus,
>
> $\operatorname{Bias}\left(\hat{\mathcal{Z}} _ \sigma\right)=O(\delta), \quad \operatorname{Bias}^2\left(\hat{\mathcal{Z}} _ \sigma\right)=O\left(\delta^2\right).$
>
> Next, recall the variance bound given in Theorem 5.2 in the manuscript:
>
> $\log (\operatorname{Var}(\hat{\mathcal{Z}} _ {\sigma})+1) \leq \mathcal{B}(\pi, q)$,
>
> where $\mathcal{B}(\pi,q)$ is a sum of $T$ step-wise KL divergence terms. Denoting the max per-step KL divergence term by $C _ {\rm KL}$​, we obtain
>
> $\operatorname{Var}\left(\hat{\mathcal{Z}} _ \sigma\right) \leq e^{T \cdot C _ {\mathrm{KL}}}-1.$
>
> For any estimator of a scalar quantity, the mean squared error decomposes as [1]
> $\operatorname{MSE}\left(\hat{\mathcal{Z}} _ \sigma\right)=\operatorname{Bias}^2\left(\hat{\mathcal{Z}} _ \sigma\right)+\operatorname{Var}\left(\hat{\mathcal{Z}} _ \sigma\right)=O\left(\delta^2\right)+O\left(e^{T \cdot C _ {\mathrm{KL}}}-1\right).$
>
> Completing the proof.
>
>
>
> [1] Mean squared error, deconstructed. 2021.
>
> ------
>
> >

---

> ### Author Response · Authors · 2025-11-20
> **Author Response (3/4)**
>
> >(Weakness 2 & Question 2) Concerns on theory-practice gap in reward signal (Weakness 2 and Question 2).
>
> - **Regarding the concern "the framework cannot disentangle the mode-seeking bias and bias from the imperfect reward model".**
>
>   We acknowledge that our original theoretical framework focuses on the intrinsic mode-seeking bias of generator-based methods and does not explicitly model the external bias introduced by an imperfect reward model.
>
>   To address this concern, we provide a formal bias decomposition theorem below. We have included this theorem below in our updated version (see Theorem C.5 in Appendix C.5).
>
>   **Theorem 2 (Bias decomposition of generator-based methods under an imperfect reward model).** Let the true target distribution be $\sigma(\mathbf{x} _ {1:T}) \propto p _ 0(\mathbf{x} _ {1:T}) R(\mathbf{x} _ {1:T})$ with normalizing constant $Z _ {\sigma} = \int p _ 0(\mathbf{x} _ {1:T}) R(\mathbf{x} _ {1:T}) d\mathbf{x} _ {1:T}$ and the pre-trained language model $p _ 0(\mathbf{x} _ {1:T})$. Suppose we use a learned reward model $RM(\mathbf{x} _ {1:T}) = R(\mathbf{x} _ {1:T}) + \varepsilon(\mathbf{x} _ {1:T})$ with $|\varepsilon(x)| \le \delta$ for all $\mathbf{x} _ {1:T}$. The induced approximate target is $\sigma'(\mathbf{x} _ {1:T}) \propto p _ 0(\mathbf{x} _ {1:T}) RM(\mathbf{x} _ {1:T})$ with normalizing constant $Z _ {\sigma'} = \int p _ 0(\mathbf{x} _ {1:T}) (R(\mathbf{x} _ {1:T}) + \varepsilon(\mathbf{x} _ {1:T})) d\mathbf{x} _ {1:T}$. Let $\hat{Z}^{\mathrm{VI}} _ {\sigma'}$ be the VI estimator of $Z _ {\sigma'}$ defined by $\hat{Z}^{\mathrm{VI}} _ {\sigma'} = \exp(\mathrm{ELBO}(q))$, where $\mathrm{ELBO}(q)$ is the variational lower bound for $\log Z _ {\sigma'}$ regarding the generator $q(\mathbf{x} _ {1:T})$.  Then the estimation bias with respect to the true normalizing constant satisfies
>
>   $\left|Z_\sigma-\hat{Z} _ {\sigma^{\prime}}^{\mathrm{VI}}\right| \leq\left|\int p_0\left(\mathbf{x} _ {1: T}\right) \varepsilon\left(\mathbf{x} _ {1: T}\right) d \mathbf{x} _ {1: T}\right|+Z _ {\sigma^{\prime}}\left(1-e^{-\mathrm{KL}\left(q || \sigma^{\prime}\right)}\right)=O(\delta)+O\left(Z _ {\sigma^{\prime}}\left(1-e^{-C _ {\mathrm{KL}} \cdot T}\right)\right),$
>
>   where the first term corresponds to the external bias from the imperfect reward model, and the second term corresponds to the intrinsic VI mode-seeking bias. Here $C _ {\mathrm{KL}} > 0$ is a constant where we suppose the sequence-level KL divergence admits an upper bound $\mathrm{KL}(q || \sigma') \le C _ {\mathrm{KL}} \cdot T$​ under the standard autoregressive factorization over $T$ steps.
>
>
>
>   **Remark.** This theorem shows that the total estimation bias decomposes naturally into two components: (i) The **external reward-model bias**, which is uniformly bounded by the reward-model approximation error $\delta$ and does not grow with the step number $T$. Thus the impact of imperfect reward modeling remains controlled. (ii) The **intrinsic VI mode-seeking bias**, which can grow exponentially with $T$ through the sequence-level KL divergence. In particular, whenever the sequence-level KL does not vanish, the factor $(1 - e^{-C _ {\mathrm{KL}} \cdot T})$ rapidly approaches $1$ as $T$ increases, implying that the VI-induced bias can become $\Theta(Z _ {\sigma'})$ for long sequences.
>
>
>
> - **Regarding the question “Is the observed VI bias dominated by the reward-model bias?”**
>
>   No. The reward-model bias is at most $O(\delta)$ and independent of the step number $T$, while the intrinsic VI bias scales as $O\left(Z _ {\sigma'}(1 - e^{-C _ {\mathrm{KL}}T})\right)$ and typically dominates for sufficiently long sequences. Moreover, in our experiments of mathatical reasoning, the reward is rule-based and exact ($\delta = 0$), so the observed bias arises solely from the intrinsic VI mode-seeking bias.
>
>
>
> - **Regarding the question “Does the VI–RFT equivalence still hold with a biased reward model?”**
>
>   Yes. A biased reward model $RM(\mathbf{x} _ {1:T})=R(\mathbf{x} _ {1:T})+\varepsilon(\mathbf{x} _ {1:T})$ simply induces an approximate target $\sigma'(\mathbf{x} _ {1:T})\propto q(\mathbf{x} _ {1:T})RM(\mathbf{x} _ {1:T})$, and both VI and RFT optimize the same ELBO with respect to $\sigma'$. The structural mode-seeking behavior of minimizing $\mathrm{KL}(q|\sigma')$ remains unchanged, so the theoretical equivalence continues to hold.

---

> ### Author Response · Authors · 2025-11-20
> **Author Response (4/4)**
>
> ***Proof of Theorem 2*.** By definition of the true and approximate targets, we have
>
> $Z_\sigma=\int p_0\left(\mathbf{x} _ {1: T}\right) R\left(\mathbf{x} _ {1: T}\right) d \mathbf{x} _ {1: T}, \quad Z _ {\sigma^{\prime}}=\int p_0\left(\mathbf{x} _ {1: T}\right)\left(R\left(\mathbf{x} _ {1: T}\right)+\varepsilon\left(\mathbf{x} _ {1: T}\right)\right) d \mathbf{x} _ {1: T}.$
>
> Therefore,
>
> $Z _ {\sigma^{\prime}}-Z_\sigma=\int p_0\left(\mathbf{x} _ {1: T}\right) \varepsilon\left(\mathbf{x} _ {1: T}\right) d \mathbf{x} _ {1: T}.$
>
> Taking absolute values and using $|\varepsilon(\mathbf{x} _ {1:T})| \le \delta$ and $\int q(\mathbf{x} _ {1:T}) d\mathbf{x} _ {1:T} = 1$​ yields
>
> $\left|Z _ {\sigma^{\prime}}-Z_\sigma\right|=\left|\int p_0\left(\mathbf{x} _ {1: T}\right) \varepsilon\left(\mathbf{x} _ {1: T}\right) d \mathbf{x} _ {1: T}\right| \leq \int p_0\left(\mathbf{x} _ {1: T}\right)\left|\varepsilon\left(\mathbf{x} _ {1: T}\right)\right| d \mathbf{x} _ {1: T} \leq \delta.$
>
> This term corresponds to the external bias arising from the imperfect reward model and is of order $O(\delta)$.
>
> Next, consider the VI estimator under the approximate target $\sigma'$. By the standard variational identity,
>
> $\log Z _ {\sigma^{\prime}}=\operatorname{ELBO}(q)+\operatorname{KL}\left(q \| \sigma^{\prime}\right),$
>
> we have
>
> $\hat{Z} _ {\sigma^{\prime}}^{\mathrm{VI}}=\exp (\operatorname{ELBO}(q))=\exp \left(\log Z_\sigma -\mathrm{KL}\left(q \| \sigma^{\prime}\right)\right)=Z _ {\sigma^{\prime}} \exp \left(-\mathrm{KL}\left(q \| \sigma^{\prime}\right)\right).$
>
> Thus, the VI-induced bias with respect to $Z _ {\sigma'}$ is
>
> $Z _ {\sigma^{\prime}}-\hat{Z} _ {\sigma^{\prime}}^{\mathrm{VI}}=Z _ {\sigma^{\prime}}-Z _ {\sigma^{\prime}} e^{-\mathrm{KL}\left(q \| \sigma^{\prime}\right)}=Z _ {\sigma^{\prime}}\left(1-e^{-\mathrm{KL}\left(q \| \sigma^{\prime}\right)}\right).$
>
> This is the intrinsic mode-seeking bias of VI with respect to the approximate target $\sigma'$.
>
> To bound the total bias with respect to the true normalizing constant $Z _ {\sigma}$, we write
>
> $Z_\sigma-\hat{Z} _ {\sigma^{\prime}}^{\mathrm{VI}}=\left(Z_\sigma-Z _ {\sigma^{\prime}}\right)+\left(Z _ {\sigma^{\prime}}-\hat{Z} _ {\sigma^{\prime}}^{\mathrm{VI}}\right),$
>
> and apply the triangle inequality:
>
> $\left|Z_\sigma-\hat{Z} _ {\sigma^{\prime}}^{\mathrm{VI}}\right| \leq\left|Z_\sigma-Z _ {\sigma^{\prime}}\right|+\left|Z _ {\sigma^{\prime}}-\hat{Z} _ {\sigma^{\prime}}^{\mathrm{VI}}\right|=\left|\int p_0\left(\mathbf{x} _ {1: T}\right) \varepsilon\left(\mathbf{x} _ {1: T}\right) d \mathbf{x} _ {1: T}\right|+Z _ {\sigma^{\prime}}\left(1-e^{-\mathrm{KL}\left(q \| \sigma^{\prime}\right)}\right).$
>
> Finally, under the standard autoregressive factorization of both $q$ and $\sigma'$, the sequence-level KL divergence can be decomposed as
>
> $\mathrm{KL}\left(q \| \sigma^{\prime}\right)=\sum _ {t=1}^T \mathbb{E} _ {q\left(\mathbf{x} _ {<t}\right)}\left[\mathrm{KL}\left(q\left(\mathbf{x}_t \mid \mathbf{x} _ {<t}\right) \| \sigma^{\prime}\left(\mathbf{x}_t \mid \mathbf{x} _ {<t}\right)\right)\right] .$​
>
> If the per-step conditional KL is bounded upper by a constant $C _ {\mathrm{KL}} > 0$, which yields the claimed $\left|Z_\sigma-\hat{Z} _ {\sigma^{\prime}}^{\mathrm{VI}}\right|=O(\delta)+O\left(Z _ {\sigma^{\prime}}\left(1-e^{-C _ {\mathrm{KL}} \cdot T}\right)\right).$
>
>  Completing the proof.
>
> ------
>
> > (Question 3) Concerns on trade-off between performance and computational cost.
>
> A detailed analysis of wall-clock time and memory usage is provided in Appendix E.3. As summarized in the table below, TSMC-TTS achieves a **+11% improvement** in solving rate while incurring higher inference cost due to jointly loading a generator and a verifier. While the two-model setup does introduce additional cost, the resulting accuracy improvements offer a favorable trade-off for applications that prioritize correctness. In practice, the overhead can be further reduced through techniques such as CPU offloading, quantization, or distributed inference (e.g., pipeline/tensor parallelism), which mitigate memory overhead and latency.
>
> | Method          | Solving Rate (Llemma-7B on GSM8K) | Time Cost Per Batch (Seconds) | Memory Cost Per Batch (GB) |
> | --------------- | --------------------------------- | ----------------------------- | -------------------------- |
> | Majority Voting | 71.2                              | 17.42                         | 26                         |
> | TSMC-TTS (ours) | 82.2                              | 33.89                         | 48                         |

---

### Official Review · Reviewer_5mgV · 2025-11-01

**Soundness:** 3
**Presentation:** 3
**Contribution:** 2
**Rating:** 4
**Confidence:** 4

**Summary:**

This paper addresses the problem of improving Parallel Test-time Scaling (PTTS) for mathematical reasoning in Large Language Models (LLMs). The authors propose a novel and general probabilistic inference framework that recasts existing PTTS methods through the lens of the bias-variance tradeoff. Within this framework, the paper unifies two common approaches: Verifier-based methods are identified as Importance Sampling (IS) estimators, which are unbiased but suffer from high variance. Generator-based methods (using reinforcement fine-tuning) are identified as Variational Inference (VI) estimators, which achieve low variance but introduce systematic bias. To mitigate this tradeoff and achieve an estimator that is both unbiased and has low variance, the paper proposes TSMC-TTS. This method uses Twisted Sequential Monte Carlo (TSMC) as the inference algorithm. The key technical novelty is a principled joint optimization of the generator and the verifier (i.e., the twist function)5. This optimization is guided by a theoretically derived upper bound on the relative variance and employs an EM-like algorithm to ensure monotonic variance reduction. This approach also uses a "self-evolving" target distribution, which iteratively refines the generator. Experiments on GSM8K and MATH500 show that TSMC-TTS outperforms the verifier-based and generator-based baselines included in the main tables.

**Strengths:**

- The paper is well-written, and the core idea is presented clearly.

- The primary strength is the novel conceptual framework that unifies PTTS approaches under the bias-variance tradeoff. Casting verifier-based methods as IS and generator-based methods as VI is an elegant and insightful contribution. This perspective provides a principled and systematic way to understand the intrinsic limitations (high variance vs. systematic bias) of these popular methods.

- The proposed TSMC-TTS method is theoretically well-grounded. The derivation of the relative variance upper bound (Theorem 5.2) and the formulation of the EM-like joint optimization algorithm (Property 5.5) provide a solid theoretical foundation for the proposed approach.

- Empirically, the method achieves strong results, outperforming the baseline methods (Base+MV, Base+ORM, RFT+MV) presented in the main comparison tables.

**Weaknesses:**

- The main weakness of this paper lies in its comparison to very closely related prior work, **specifically Feng et al. (2025)**. The paper's core technical contribution over Feng et al. is the *joint optimization* of the generator and verifier (using a "self-evolving" target distribution), whereas Feng et al. used a *fixed* generator and a fixed target distribution.

- However, the empirical gains from this new joint optimization appear to be **marginal**. For example, on DeepSeek-7B/MATH500, TSMC-TTS achieves 61.8%, while the fixed-generator TSMC from Feng et al. (2025) reported 60.8%. On Llemma-7B/GSM8K, the gap is similarly small at 82.2% vs. 80.4%. This minimal improvement (0.8-1.8%) calls into question the practical value of the paper's main technical novelty, especially given the significant additional training compute required to jointly optimize the generator.

- For transparency, the results from Feng et al. (2025) should be included directly in the main results (Table 2), as this is the most relevant baseline for the TSMC-based approach. Relegating this comparison to a discussion in the appendix makes the paper's contribution seem larger than it is.

- Additionally, the proposed EM-like optimization (Alg. 1) relies on estimating gradients from samples. The stability of this "online updating" process is a potential concern. If the batch size $B$ is small, the gradient estimates (Eq. 19, 20) could have high variance, potentially leading to unstable training.

**Questions:**

Related to the stability concern in the weakness section: How sensitive is the joint optimization algorithm (Alg. 1) to the batch size $B$ used for gradient estimation? Given that the algorithm's design already takes steps to avoid "noisy gradients", could a small $B$ lead to high-variance loss estimates that make the training unstable or prevent the joint optimization from converging effectively?

---

> ### Author Response · Authors · 2025-11-20
> **Author Response (1/2)**
>
> We sincerely appreciate your time and constructive suggestions. We are also encouraged by your recognition of the novelty and theoretical soundness of our framework and methodology. Below, we carefully address your concerns in detail.
>
> ------
>
>
>
> > (Weakness 1) Concerns about comparison with Feng et al. (2025).
>
> To address this concern, we would like to highlight the key distinctions and novel contributions of our work compared with Feng et al. (2025).
>
> 1. **First general framework for PTTS under the lens of the bias–variance tradeoff.** We present the first general framework for PTTS through a unifying probabilistic inference formulation, seamlessly encompassing previously disparate methods as special cases. This framework offers a new bias–variance tradeoff perspective to reveal the intrinsic limitations of existing methods and provides a principled foundation for developing new methods. **In contrast, prior studies, including Feng et al. (2025), have not yet introduced or explored such a general and versatile conceptual framework for PTTS.**
> 2. **First theoretical formulation of the bias-variance tradeoff in the context of PTTS.** We present the first theoretical formulation (Theorem 5.2) of the bias–variance tradeoff in the context of PTTS, which quantitatively characterizes how the relative variance upper bound is jointly governed by the generator and the verifier via their respective optimality gaps. We believe such a theoretical perspective offers valuable insights for the PTTS community largely driven by heuristics. **In contrast, Feng et al. (2025) overlook the bias–variance tradeoff in PTTS, not to mention providing any theoretical characterization of it.**
> 3. **Theory-driven generator-verifier optimization algorithm.** Our theoretical insight enables a principled and theory-driven verifier–generator optimization. Under our variance decomposition, which reveals both a verifier gap and a generator gap, **Feng et al. (2025) is provably suboptimal due to its fundamental incompatibility with the optimal conditions revealed by our theorem**. In contrast, our EM-like optimization jointly updates the generator and verifier by directly targeting the variance bound, achieving monotonic variance reduction without compromising unbiasedness.
> 4. **New TSMC mechanism: self-evolving TSMC.** We introduce a new self-evolving TSMC mechanism that uses a self-evolving target distribution $\sigma(\mathbf{x}_{1:T})$ proportional to $q(\mathbf{x} _ {1:T})R(\mathbf{x} _ {1:T})$, where $q$ is iteratively updated via our EM-like optimization. **This elegant plug-and-play design alleviates both the reward sparsity and the computational overhead issues inherent in vanilla TSMC adopted by Feng et al. (2025).**
>
> To more clearly emphasize our contributions, we have updated the introduction section with a revised contribution summary and included these additional discussions comparing our work with Feng et al. (2025) in Appendix B.2.
>
> ------
>
> >(Weakness 2) Concerns about the empirical gains over Feng et al. (2025).
>
> The table below provides a direct empirical comparison with Feng et al. (2025) under the same training conditions. As shown, our proposed TSMC-TTS consistently achieves approximately +2.5% higher accuracy across datasets and model architectures. Since Feng et al. (2025) did not release their model checkpoints, we reproduce their method by training it under exactly the same conditions as ours. In particular, we use **the same fine-tuned base model checkpoints and the corresponding generated rollouts for both their method and ours**, ensuring that the comparison is fair and isolates the effect of the learning algorithms rather than differences in data or initialization. This setup therefore provides a more accurate reflection of the performance gains.
>
> | Method             | Llemma-7B (GSM8K) | Llemma-7B (MATH500) | DeepSeekMath-7B (GSM8K) | DeepSeekMath-7B (MATH500) |
> | -- | -- | -- | -- | --|
> | Feng et al. (2025) | 79.6  | 45.2   | 90.4  | 58.8  |
> | TSMC-TTS (ours)    | 82.2| 47.2 | 92.6  | 61.8 |
>
> ------
>
> > (Weakness 3) The results from Feng et al. (2025) should be included directly in the main results (Table 2).
>
> Thanks for the suggestion. We have updated Table 2 in the revised manuscript to directly include the results of Feng et al. (2025).
>
> ------

---

> ### Author Response · Authors · 2025-11-20
> **Author Response (2/2)**
>
> ------
>
> >  (Weakness 4 & Question 1) Concerns about the stability of the gradient estimation when the batch size (B) is small.
>
> We address this concern from both analytical and empirical perspectives.
>
> 1. **Analytical perspective:  self-normalization stabilizes gradient estimates.**
>
>    As shown in Eq. (19) and Eq. (20), the derived gradient updates rely on self-normalized importance weights: $\frac{\psi\left(\mathbf{x} _ {1: t}^i\right)}{\sum _ {j=1}^B \psi\left(\mathbf{x} _ {1: T}^j\right)}$ and $\frac{\mathcal{R}\left(\mathbf{x} _ {1: T}^i\right)}{\sum _ {j=1}^B \mathcal{R}\left(\mathbf{x} _ {1: T}^j\right)}$. This structure corresponds to self-normalized importance sampling (SNIS) [1]. A property is that it bounds the influence of each sample, since each weight lies in [0,1] and all weights sum to 1 [1]. Hence, although the gradient variance may theoretically increase with smaller batch size $B$, the SNIS normalization prevents weight collapse and ensures that no single sample significantly dominates the gradient. In practice, we did not observe severe instability during optimization.
>
> 2. **Empirical support.** As shown in Appendix E.2, we have conducted a sensitivity study by varying the batch size $M \in \{10, 20, 40, 80\}$​. The results (Table 7 in Appendix E.2, also the table below) show that the performance remains stable within reasonable fluctuations (46.3–47.6% on MATH500), indicating that the optimization remains stable across a wide range of batch sizes.
>
>    | Batch Size | Solving Rate (%) on Llemma-7B (MATH500) |
>    | ---------- | --------------------------------------- |
>    | 10         | 46.3                                    |
>    | 20         | 47.6                                    |
>    | 40         | 47.2                                    |
>    | 80         | 46.8                                    |
>
>    [1] Monte Carlo Theory, Methods and Examples, 2013. Chapter 9.

---

### Author Response · Authors · 2025-11-22
**Summary of Paper Revision**

We thank all reviewers for their constructive and insightful feedback. We have provided individual responses to each reviewer. We have also uploaded a **revised version** of the paper that includes additional results and clarifications. The main updates are as follows:

- **Abstract and Introduction:** Refined contribution summary to better highlight our core insights.

- **Section 4:** Improved notation for better clarity.

- **Table 2:** Performance comparison with vanilla TSMC in Feng et al., 2025.

- **Table 2:** Additional discussion on hybrid baselines (i.e., naive combinations of generator-based and verifier-based methods).

- **Table 3:** Experiments on generalized domain-specific reasoning.

- **Appendix B.2:** Discussion of our contributions compared with existing work.

- **Theorem C.3:** Bias decomposition under approximate RFT optimization.

- **Theorem C.4:** Error bounds under imperfect verifiers.

- **Theorem C.5:** Bias decomposition under imperfect reward models.

---

### Meta-Review · Area_Chair_MqmZ · 2025-12-29

**Summary:**

This paper proposes a framework for casting Parallel Test Time Scaling (PTTS) as an inference problem for estimating expected statistics such as answer frequency. The authors establish some theoretical results connecting importance sampling and variational inference to specific test time methods. They also propose an algorithm TSMC-TTS that does joint EM-style optimization of the generator and the verifier guided by the theoretical variance bound.

**Reviewer Concerns:**

The reviewers noted that the proposed framework is elegant and novel, the proposed TSMC-TTS algorithm is theoretically grounded, and the empirical results are encouraging. One of the reviewers also commented on lack of adequate comparison with a recent related work. There were also questions about the clarity of notations. Finally, there were expressed concerns about some of the idealized assumptions the authors made (e.g.., knowing  the true reward function) and limited evaluation on only two datasets.

**Reviewer Scores:**

5mgV - increase to 6;
MrsS - unchanged/increase to 6
CfpZ - unchanged/increase to 6
UupB - unchanged/increase to 4

AC concerns: The authors provided extensive responses to address those comments. However, those responses do not fully mitigate the issues.  For instance, the authors provided new theoretical results by relaxing the assumption of the known reward function, however, their conclusions still hinge on the assumption that the reward function is known with sufficiently high accuracy (within small delta), which is not necessarily feasible in practice. Also, while the authors added new experiments results two more datasets, the evaluation still seems inadequate, as it does not include stronger baselines (such as search-based methods) and/or stronger base models. It’s not clear whether TSMC-TTS will have advantage over those methods, especially given its significant computational overhead (Tables 8, 9 in the appendix). Furthermore, the Math dataset is rather saturated, and would be much more convincing if the authors experiments with more challenging math/reasoning datasets such as AIME 24/25, GPQA Diamond, etc. Finally, although the authors claim that their framework unifies all different PTTS methods, this is not necessarily the case, and the framework seems to cover only majority voting (Eq. 7) and weighted majority voting (Eq. 3). Even the canonical best-of-n method is captured under this framework only in the special case where all the sampled solutions are unique, so that Eq. 3 picks up the solution with the best score. In the revised submission the author should comment on this limitation of their framework to analyze other popular classes of TTS techniques.

---

### Decision · Program_Chairs · 2026-01-26

Reject